# Linear Convergence in Federated Learning: Tackling Client Heterogeneity and Sparse Gradients

**Aritra Mitra**     **Rayana Jaafar**     **George J. Pappas**     **Hamed Hassani**
Department of Electrical and Systems Engineering
{amitra20,rayanaj,pappasg,hassani}@seas.upenn.edu

## Abstract

We consider a standard federated learning (FL) setup where a group of clients periodically coordinate with a central server to train a statistical model. We develop a general algorithmic framework called FedLin to tackle some of the key challenges intrinsic to FL, namely objective heterogeneity, systems heterogeneity, and infrequent and imprecise communication. Our framework is motivated by the observation that under these challenges, various existing FL algorithms suffer from a fundamental speed-accuracy conflict: they either guarantee linear convergence but to an incorrect point, or convergence to the global minimum but at a sub-linear rate, i.e., fast convergence comes at the expense of accuracy. In contrast, when the clients' local loss functions are smooth and strongly convex, we show that FedLin guarantees linear convergence to the global minimum, despite arbitrary objective and systems heterogeneity. We then establish matching upper and lower bounds on the convergence rate of FedLin that highlight the effects of infrequent, periodic communication. Finally, we show that FedLin preserves linear convergence rates under aggressive gradient sparsification, and quantify the effect of the compression level on the convergence rate. Notably, our work is the first to provide tight linear convergence rate guarantees, and constitutes the first comprehensive analysis of gradient sparsification in FL.

## 1   Introduction

In a canonical federated learning (FL) architecture, a set $\mathcal{S}$ of clients periodically communicate with a central server to find a global statistical model that solves the following problem [1–5]:

$$\min_{x \in \mathbb{R}^d} f(x), \quad \text{where } f(x) = \frac{1}{m} \sum_{i=1}^{m} f_i(x). \tag{1}$$

Here, $m$ is the number of clients, $f_i : \mathbb{R}^d \to \mathbb{R}$ is the local objective (loss) function of client $i$, and $f(x)$ is the global objective function. Some of the core distinguishing tenets of the FL paradigm are as follows [1–5]. First, due to privacy considerations, clients cannot directly share their local training data with the server. Second, differences in the clients' data-sets may cause the clients to have non-identical loss functions with different minima - this is known as *statistical* or *objective* heterogeneity. Third, due to variability in hardware (CPU, memory) and power (battery level), i.e., due to *systems* or *device* heterogeneity, the client devices may have different computation speeds; in particular, this may lead to slow and straggling devices that affect convergence guarantees. Finally, *communication-efficiency* is a major concern, dictating the need to reduce the number of communication rounds, and also the size of the messages transmitted in each round. The above considerations pose unique technical challenges that we aim to address in this paper.

In a typical FL setting, to reduce the number of communication rounds, clients perform multiple local training steps in isolation before communicating with the server. Due to such local steps, the popular

35th Conference on Neural Information Processing Systems (NeurIPS 2021).

`FedAvg` algorithm suffers from a "client-drift phenomenon" under objective heterogeneity [6–11]: the local iterates of each client drift-off towards the minimum of their own local loss function, leading to slow convergence rates. For analysis on `FedAvg`, we refer the reader to [6, 8, 12–21]. Recently, several new algorithms such as `FedProx` [22], `SCAFFOLD` [11], `FedSplit` [10], and `FedNova` [23] have been proposed as improvements to `FedAvg`. Despite these advances, there remain gaps in our understanding of the extent to which these algorithms match the guarantees of a centralized baseline.[1]

For instance, even for simple, deterministic settings, `FedProx` [22] and `FedNova` [23] exhibit a fundamental speed-accuracy conflict under objective heterogeneity; see [8, 9] and Section 2. Specifically, with constant step-sizes, these algorithms converge linearly, but potentially to an incorrect point. Thus, convergence to the minimum of the global loss function necessitates diminishing step-sizes, which, in turn, leads to sub-linear convergence. Thus, fast convergence comes at the expense of accuracy. Although `SCAFFOLD` [11] and `FedSplit` [10] employ variance-reduction and operator-splitting techniques, respectively, to tackle objective heterogeneity, it is not known whether the rates in these papers are tight. More importantly, neither `SCAFFOLD` nor `FedSplit` account for the effects of systems heterogeneity or compression, both of which are key challenges in FL. Indeed, due to systems heterogeneity, the number of local steps may vary across clients, causing some clients to make much less progress than others in each round [23]. Moreover, while empirical studies [24, 25] have revealed significant benefits of biased sparsification, theoretical guarantees for such methods in a federated setting have remained elusive. In this context, our **contributions** are as follows.

● **A New Algorithm:** Motivated by the above concerns, we develop a general algorithmic framework called `FedLin` that simultaneously accounts for objective heterogeneity, systems heterogeneity, and gradient sparsification. The key components of `FedLin` include a gradient correction term in the local update rule that exploits memory; the use of client-specific learning rates; and error-feedback mechanisms at the clients and the server.

● **Matching Centralized Rates:** For smooth and strongly convex losses, we show that `FedLin` converges to the global minimum linearly in the deterministic setting, and with a $O(1/T)$ rate for a general stochastic oracle model, thereby matching centralized rates (up to constants). We then present matching rates for smooth, convex and non-convex settings as well. Importantly, our results hold under *arbitrary* objective *and* systems heterogeneity. In contrast, the only other work in FL (as far as we are aware) that investigates both objective and systems heterogeneity [23] provides results only for the non-convex setting, under a bounded dissimilarity assumption. Moreover, the `FedNova` algorithm in [23] suffers from the speed-accuracy conflict, while `FedLin` does not.

● **Quantifying the Price of Multiple Local Steps:** We establish a lower bound for `FedLin` that matches the upper-bound we obtain for smooth, strongly convex losses. In doing so, we provide the first (as far as we are aware) *tight* linear convergence rate analysis. Our lower bound highlights the price paid for performing multiple local steps, i.e., the effect of infrequent communication on the convergence rate. In particular, our analysis reveals, perhaps surprisingly, that there exist simple instances (involving quadratic losses) for which performing multiple local steps does not improve the rate of convergence, indicating that *even mild statistical heterogeneity can hurt.* Our analysis also provides valuable insights into the limitations of gradient-tracking/variance-reduction techniques.

● **Analyzing the Impacts of Gradient Sparsification at Server and at Clients:** While several works explore the effect of unbiased random quantization in distributed settings [26–31], there are only a handful of papers [15, 32] that also consider the effect of local steps in FL. Different from all these works, we explore the impacts of sparsifying gradients using a *biased* `TOP-k` operator, both at the server side and at the clients. Our results in this context (i) constitute the *first formal study of gradient sparsification in a federated setting*; (ii) reveal key differences between up-link and down-link compression; and (iii) quantify the effect of the compression level on the convergence rate. Notably, `FedLin` preserves linear convergence rates despite aggressive gradient sparsification.

**Basic Notation and Terminology:** Referring to (1), let $x^* \in \arg\min_{x \in \mathbb{R}^d} f(x)$, and $x_i^* \in \arg\min_{x \in \mathbb{R}^d} f_i(x)$. Every FL algorithm mentioned in this paper operates in rounds $t \in \{1, \ldots, T\}$. In each round $t$, every client performs a certain number of local steps in isolation, starting from a common global model $\bar{x}_t$. We will denote by $x_{i,\ell}^{(t)}$ client $i$'s estimate of the model at the $\ell$-th local step of round $t$. In particular, $x_{i,0}^{(t)} = \bar{x}_t, \forall i \in \mathcal{S}$.

---

[1]By a centralized baseline, we refer to a setup where each client can communicate with every other client at all time steps via the server.

| Method | Linear Convergence to $x^*$ | Lower Bounds | Variable Client Speeds | Sparsification/ Compression |
|---|---|---|---|---|
| FedAvg [2] | ✗ | Thm. II in [11] | ✗ | ✗ |
| FedProx [22] | ✗ | — | ✗ | ✗ |
| FedNova [23] | ✗ | — | ✓ | ✗ |
| FedSplit [10] | ✗ | — | ✗ | ✗ |
| SCAFFOLD [11] | ✓ | — | ✗ | ✗ |
| FedLin (Sec. 3) | ✓ | Thm. 5 | ✓ | ✓ |

Table 1: Comparison of our proposed algorithm FedLin with popular FL algorithms. We indicate whether or not each algorithm (i) guarantees linear convergence to $x^*$ for smooth, strongly convex losses in a deterministic setting under objective heterogeneity; (ii) comes with lower bounds; (iii) accounts for variable local steps across clients (systems heterogeneity); and (iv) performs compression.

## 2 Motivation: Speed-Accuracy Trade-Off

To motivate our work, we first show how some recently proposed FL algorithms, namely FedProx [22] and FedNova [23], exhibit a fundamental speed-accuracy trade-off even in simple, deterministic settings. Specifically, we show that these schemes do not, in general, guarantee convergence to the global minimum with constant step-sizes. This, in turn, necessitates diminishing step-sizes, leading to sub-linear convergence rates. Our analysis here is inspired by that in [8] for FedAvg. We consider a deterministic quadratic model where the local loss function of client $i$ is given by

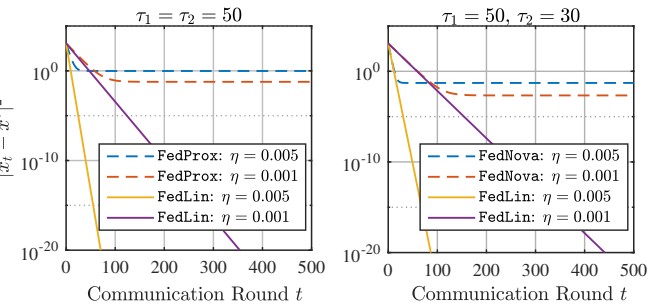

Figure 1: Simulations comparing FedProx, FedNova, and FedLin for two clients with $f_1(x) = (1/2)(x - 3)^2$ and $f_2(x) = (x - 50)^2$. **Left**: Clients perform the same number of local steps, $H = 50$. For FedProx, we set $\beta = 5$. **Right**: Clients 1 and 2 perform 50 and 30 local steps, respectively.

$f_i(x) = 1/2\|A_i^{1/2}(x - c_i)\|^2$, where $A_i$ is a symmetric positive-definite matrix. We begin by assuming that all clients perform the same number of local steps $H$. The following is the FedProx update rule where a proximal term is added to mitigate client-drift:

$$x_{i,\ell+1}^{(t)} = x_{i,\ell}^{(t)} - \eta\left(\nabla f_i(x_{i,\ell}^{(t)}) + \beta(x_{i,\ell}^{(t)} - \bar{x}_t)\right), \ell = 0, \ldots, H - 1; \quad \bar{x}_{t+1} = \frac{1}{m}\sum_{i\in\mathcal{S}} x_{i,H}^{(t)}. \quad (2)$$

**Proposition 1.** *For any step-size $\eta > 0$, $T$ rounds of FedProx amount to performing $T$ rounds of parallel GD on the* surrogate *optimization problem given by*

$$\min_x \frac{1}{m}\sum_{i\in\mathcal{S}} \frac{1}{2}\left\|\left(\sum_{\ell=0}^{H-1}[I - \eta(A_i + \beta I)]^\ell A_i\right)^{1/2}(x - c_i)\right\|^2. \quad (3)$$

Proposition 1 shows that even when clients perform the same number of local updates, FedProx minimizes a surrogate objective function (3) whose minimum may not, in general, coincide with the minimum of the original problem. When $\beta = 0$, FedProx reduces to FedAvg, and our observations continue to hold. To capture systems heterogeneity as in [23], suppose now that client $i$ performs $\tau_i$ local steps. Define $\tau_{eff} \triangleq 1/m \sum_{i\in\mathcal{S}} \tau_i$ and $\alpha_i \triangleq \tau_{eff}/\tau_i, \forall i \in \mathcal{S}$. The update rule of FedNova relies on normalized aggregation of cumulative local gradients, and is given by

$$x_{i,\ell+1}^{(t)} = x_{i,\ell}^{(t)} - \eta\nabla f_i(x_{i,\ell}^{(t)}); \quad \bar{x}_{t+1} = \bar{x}_t - \frac{\eta}{m}\sum_{i\in\mathcal{S}}\alpha_i\sum_{\ell=0}^{\tau_i-1}\nabla f_i(x_{i,\ell}^{(t)}), \quad (4)$$

where $\ell = 0, \ldots, \tau_i - 1, \ i \in \mathcal{S}$. Although FedNova can accommodate any local solver whose accumulated gradients are expressible as a linear combination of local gradients, we choose gradient descent, a simple solver, to isolate the impact of *normalized aggregation* - the essence of FedNova.

**Algorithm 1** FedLin

---

1: **Input:** Client step-sizes $\eta_i, i \in \mathcal{S}$, compression levels $\delta_c$ and $\delta_s$, initial iterate $\bar{x}_1 \in \mathbb{R}^d$,
   $g_1 = \nabla f(\bar{x}_1)$, initial compression errors $\rho_{i,1} = 0, \forall i \in \mathcal{S}$ and $e_1 = 0$
2: **for** $t = 1, \ldots, T$ **do**
3:      **for** $i = 1, \ldots, m$ **do**
4:          **for** $\ell = 0, \ldots, \tau_i - 1$ **do**
5:              $x_{i,\ell+1}^{(t)} \leftarrow x_{i,\ell}^{(t)} - \eta_i(\nabla f_i(x_{i,\ell}^{(t)}) - \nabla f_i(\bar{x}_t) + g_t); \quad x_{i,0}^{(t)} = \bar{x}_t$
6:          **end for**
7:          Transmit $x_{i,\tau_i}^{(t)}$ to server
8:      **end for**
9:      Server transmits $\bar{x}_{t+1} = 1/m \sum_{i \in \mathcal{S}} x_{i,\tau_i}^{(t)}$
10:      **for** $i = 1, \ldots, m$ **do**
11:          Transmit $h_{i,t+1} = \mathcal{C}_{\delta_c}(\rho_{i,t} + \nabla f_i(\bar{x}_{t+1}))$ to server
12:          $\rho_{i,t+1} \leftarrow \rho_{i,t} + \nabla f_i(\bar{x}_{t+1}) - h_{i,t+1}$
13:      **end for**
14:      Server transmits $g_{t+1} = \mathcal{C}_{\delta_s}(e_t + 1/m \sum_{i \in \mathcal{S}} h_{i,t+1})$
15:      $e_{t+1} \leftarrow e_t + 1/m \sum_{i \in \mathcal{S}} h_{i,t+1} - g_{t+1}$
16: **end for**

---

**Proposition 2.** *For any step-size $\eta > 0$, $T$ rounds of* FedNova *amount to performing $T$ rounds of parallel GD on the* surrogate *optimization problem given by*

$$\min_x \frac{1}{m} \sum_{i \in \mathcal{S}} \frac{1}{2} \left\| \left( \sum_{\ell=0}^{\tau_i - 1} [I - \eta A_i]^\ell \alpha_i A_i \right)^{1/2} (x - c_i) \right\|^2. \tag{5}$$

For the proofs of Propositions 1 and 2, see Appendix B. Proposition 2 shows that in the presence of both objective and systems heterogeneity, FedNova minimizes a surrogate loss function whose minimum may not coincide with $x^*$.[2] Observe from (3) and (5) that using a larger learning rate $\eta$ introduces more *distortion* to the original problem. In Figure 1, we see how FedProx and FedNova both converge to incorrect minimizers, even for simple instances with two clients and deterministic, quadratic losses. In contrast, FedLin, our proposed approach that we develop in the next section, guarantees linear convergence to the global minimum.

**Main Takeaway:** The main message we want to convey here is that even for deterministic settings, there are non-trivial challenges posed by objective and systems heterogeneity that only get amplified when one additionally considers biased compression. For such scenarios, it is not at all apparent whether (and to what extent) one can match even the basic centralized benchmark of achieving linear convergence for smooth, strongly convex loss functions. To focus on the above unresolved issues, we will primarily consider a deterministic model in this paper. *Nonetheless, the general approach we develop applies to the stochastic setting as well, as aptly demonstrated by Theorem 4 in Section 4.*

## 3 Proposed Algorithm: FedLin

In this section, we develop our proposed algorithm FedLin, formally described in Algorithm 1. FedLin is initialized from a common global iterate $\bar{x}_1 \in \mathbb{R}^d$. For simplicity, we assume that $g_1 = \nabla f(\bar{x}_1)$, i.e., every client has access to the true gradient of $f(\cdot)$ initially; we can allow $g_1$ to be arbitrary as well without affecting the convergence guarantees. FedLin proceeds in rounds: in each round $t$, starting from a common global model $\bar{x}_t$, each client $i$ performs $\tau_i$ local training steps in parallel, as per line 5 of Algorithm 1. The key features of our local update rule are as follows: *exploiting past gradients* to account for objective heterogeneity, using *client-specific step-sizes* to tackle systems heterogeneity, and employing *error-feedback* to account for gradient sparsification. We now discuss each of these features in detail.

---

[2] In a follow-up work to [8], the authors in [9] generalize their framework to encompass proximal methods such as FedProx as well. As such, Propositions 1 and 2 in this section turn out to be special cases of the results in [9]. We were not aware of [9] at the time of submission of this paper.

To gain intuition regarding the local step in line 5, note that the ideal local update at client $i$ is $x_{i,\ell+1}^{(t)} = x_{i,\ell}^{(t)} - \eta_i \nabla f(x_{i,\ell}^{(t)})$. However, this requires client $i$ to have access to the gradients of all other clients - which it does not, since clients do not communicate between rounds. To get around this, client $i$ *exploits memory*, and uses the gradient of the global function $\nabla f(\bar{x}_t)$ from the beginning of round $t$ (when the clients last communicated) as a guiding direction in its update rule. However, since $\nabla f(\bar{x}_t)$ is evaluated at a stale point $x_{i,0}^{(t)} = \bar{x}_t$, client $i$ subtracts off $\nabla f_i(\bar{x}_t)$ from $\nabla f(\bar{x}_t)$, and adds in the most recently evaluated gradient $\nabla f_i(x_{i,\ell}^{(t)})$. This results in the update rule: $x_{i,\ell+1}^{(t)} = x_{i,\ell}^{(t)} - \eta_i(\nabla f_i(x_{i,\ell}^{(t)}) - \nabla f_i(\bar{x}_t) + \nabla f(\bar{x}_t))$. Our local update rule in line 5 is precisely of the above form, where $g_t$ is an inexact version of $\nabla f(\bar{x}_t)$ to account for gradient sparsification.

When each client $i$ performs $\tau_i$ local-steps, our analysis reveals that the bound on the drift-term $\|x_{i,\ell} - \bar{x}_t\|$ scales linearly in $\tau_i$ (see Lemma 9 in Appendix F). Accordingly, to compensate for such drift at client $i$, the step-size $\eta_i$ needs to be chosen to vary inversely with the number of local steps $\tau_i$. In fact, the requirement that $\eta_i \propto 1/\tau_i$ also turns out to be necessary (see Theorem 5), providing further motivation for the choice of client-specific learning rates in `FedLin`.

To explain the gradient sparsification module, let us denote by $\mathcal{C}_\delta : \mathbb{R}^d \to \mathbb{R}^d$ the `TOP-k` operator, where $\delta = d/k$, and $k \in \{1, \ldots, d\}$. Given any $x \in \mathbb{R}^d$, let $\mathcal{E}_\delta(x)$ be a set containing the indices of the $k$ largest-magnitude components of $x$. Then, the `TOP-k` operator we consider is given by $(\mathcal{C}_\delta(x))_j = (x)_j$ if $j \in \mathcal{E}_\delta(x)$, and $(\mathcal{C}_\delta(x))_j = 0$ otherwise. Here, we use $(x)_j$ to denote the $j$-th component of a vector $x$. Clearly, a larger $\delta$ implies more aggressive compression. We employ a standard error-feedback mechanism [33–35] at both the server and the clients to account for gradient sparsification. At client $i$, $\rho_{i,t}$ represents the accumulated error due to gradient sparsification. At the end of round $t$, instead of just compressing $\nabla f_i(\bar{x}_{t+1})$, client $i$ instead compresses $\nabla f_i(\bar{x}_{t+1}) + \rho_{i,t}$, to account for gradient coordinates not transmitted in the past. It then updates the aggregate error via line 12. An analogous description applies to the error-feedback scheme at the server, where $e_t$ is the aggregate error at the beginning of round $t$. The parameters of `FedLin` are the client step-sizes $\{\eta_i\}_{i \in \mathcal{S}}$, and the compression levels $\delta_c$ and $\delta_s$ at the clients and at the server, respectively. We now comment on some related algorithmic ideas.

**Related Algorithmic Approaches:** In the related but different setting of distributed optimization, we note that the idea of exploiting past gradients has been used to design *gradient-tracking* algorithms [36–40]. In the context of FL, this idea is also related to the variance-reduction technique employed in `SCAFFOLD` [11]. A major difference of `FedLin` with the above works is that none of them consider the effect of systems heterogeneity or biased compression. In particular, accounting for the inexact gradient term $g_t$ in our update rule introduces new technical challenges that we address in this paper.

There are some additional basic differences between `FedLin` and `SCAFFOLD`. To see this, consider the update rule of `FedLin` without sparsification: $x_{i,\ell+1}^{(t)} = x_{i,\ell}^{(t)} - \eta_i(\nabla f_i(x_{i,\ell}^{(t)}) - \nabla f_i(\bar{x}_t) + \nabla f(\bar{x}_t))$. Now suppose the global model $\bar{x}_t$ at the beginning of round $t$ has already converged to $x^*$. Since $x_{i,0}^{(t)} = \bar{x}_t, \forall i \in \mathcal{S}$, and $\nabla f(x^*) = 0$, it is easy to see that the iterates of the clients do not evolve any further, as one would ideally want. *Thus, the global optimum $x^*$ can be viewed as a fixed-point of the FedLin update rule.* Adapting to our notation, and considering the case when there is no noise in the gradients, the update rule of `SCAFFOLD` takes the form $x_{i,\ell+1}^{(t)} = x_{i,\ell}^{(t)} - \eta(\nabla f_i(x_{i,\ell}^{(t)}) - c_i + c)$, where $c_i$ is a 'control-variate' maintained by client $i$, and $c$ is the average of the $c_i$'s. Importantly, the control variates $\{c_i\}_{i \in \mathcal{S}}$ used in round $t$ of `SCAFFOLD` contain stale terms from round $t-1$. As a result, even if $\bar{x}_t = x^*$, it may very well be that $(\nabla f_i(\bar{x}_t) - c_i + c) \neq 0$, causing the iterates of the clients to move away from $x^*$, and requiring further rounds of communication to average out the imbalance. Thus, the fixed-point property we discussed for `FedLin` does not hold in general for `SCAFFOLD`. Our simulations in Section 7 reveal that `FedLin` converges much faster relative to `SCAFFOLD` on a simple linear regression model; we conjecture it is precisely due to the reason described above.

Keeping aside the differences due to systems heterogeneity and compression, the `FedSVRG` algorithm in [1] includes a similar gradient correction term as in `FedLin`, but makes use of certain additional diagonal scaling and pre-conditioning matrices. Although promising empirical results are reported for `FedSVRG` in [1], these results come with no supporting theoretical guarantees of convergence. In contrast, we will develop rigorous complexity guarantees for `FedLin` in the following sections. Specifically, we will show that `FedLin` guarantees linear convergence rates despite the challenges of objective heterogeneity, systems heterogeneity, and aggressive gradient sparsification.

# 4 Matching Centralized Rates under Objective and Systems Heterogeneity

In this section, we will analyze the performance of `FedLin` in the face of both objective and systems heterogeneity. To focus solely on the effects of client heterogeneity, we will assume throughout this section that there is no gradient sparsification, i.e., $\delta_c = \delta_s = 1$. Accordingly, observe that $\rho_{i,t} = 0, e_t = 0, \forall i \in \mathcal{S}, \forall t \in \{1, \ldots, T\}$. Thus, the local update rule for `FedLin` simplifies to

$$x_{i,\ell+1}^{(t)} = x_{i,\ell}^{(t)} - \eta_i (\nabla f_i(x_{i,\ell}^{(t)}) - \nabla f_i(\bar{x}_t) + \nabla f(\bar{x}_t)). \tag{6}$$

Let us denote by $\kappa = L/\mu$ the condition number of an $L$-smooth and $\mu$-strongly convex function. Also, let $\eta_i = \bar{\eta}/\tau_i, \forall i \in \mathcal{S}$, where $\bar{\eta} \in (0, 1)$ is a flexible parameter that we will specify based on context. We are now ready to state the main results of this section.

**Theorem 1.** *(**Strongly convex case**) Suppose each $f_i(x)$ is $L$-smooth and $\mu$-strongly convex. Moreover, suppose $\tau_i \geq 1, \forall i \in \mathcal{S}$, and $\delta_c = \delta_s = 1$. Then, with $\eta_i = \frac{1}{6L\tau_i}, \forall i \in \mathcal{S}$, `FedLin` guarantees:*

$$f(\bar{x}_{T+1}) - f(x^*) \leq \left(1 - \frac{1}{6\kappa}\right)^T (f(\bar{x}_1) - f(x^*)).$$

**Theorem 2.** *(**Convex case**) Suppose each $f_i(x)$ is $L$-smooth and convex. Moreover, suppose $\tau_i \geq 1, \forall i \in \mathcal{S}$, and $\delta_c = \delta_s = 1$. Then, with $\eta_i = \frac{1}{10L\tau_i}, \forall i \in \mathcal{S}$, `FedLin` guarantees:*

$$f\left(\frac{1}{T}\sum_{t=1}^{T} \bar{x}_t\right) - f(x^*) \leq \frac{10L}{T} \left(\|\bar{x}_1 - x^*\|^2 - \|\bar{x}_{T+1} - x^*\|^2\right).$$

**Theorem 3.** *(**Non-convex case**) Suppose each $f_i(x)$ is $L$-smooth. Moreover, suppose $\tau_i \geq 1, \forall i \in \mathcal{S}$, and $\delta_c = \delta_s = 1$. Then, with $\eta_i = \frac{1}{26L\tau_i}, \forall i \in \mathcal{S}$, `FedLin` guarantees:*

$$\min_{t \in [T]} \|\nabla f(\bar{x}_t)\|^2 \leq \frac{52L}{T}(f(\bar{x}_1) - f(\bar{x}_{T+1})). \tag{7}$$

**Noisy Case Analysis:** We now analyze the performance of `FedLin` under a general stochastic oracle model. For each $i \in \mathcal{S}$ and $x \in \mathbb{R}^d$, let $q_i(x)$ be an unbiased estimate of the gradient $\nabla f_i(x)$ with variance bounded above by $\sigma^2$. We consider the update rule: $x_{i,\ell+1}^{(t)} = x_{i,\ell}^{(t)} - \eta_i(q_i(x_{i,\ell}^{(t)}) - q_i(\bar{x}_t) + q(\bar{x}_t))$, where $q(x) \triangleq 1/m \sum_{i \in \mathcal{S}} q_i(x), \forall x \in \mathbb{R}^d$. We then have the following result.

**Theorem 4.** *(**Strongly convex case with noise**) Consider the above stochastic oracle model. Suppose each $f_i(x)$ is $L$-smooth and $\mu$-strongly convex. Moreover, suppose $\tau_i \geq 1, \forall i \in \mathcal{S}$, and $\delta_c = \delta_s = 1$. For each $i \in \mathcal{S}$, let $\eta_i = \frac{\bar{\eta}}{\tau_i}$, where $\bar{\eta} \in (0, 1)$ satisfies $\bar{\eta} < \frac{1}{6L}$. Then, $\forall t \in [T]$, `FedLin` guarantees:*

$$\mathbb{E}[\|\bar{x}_{t+1} - x^*\|^2] \leq \left(1 - \frac{\bar{\eta}\mu}{2}\right) \mathbb{E}[\|\bar{x}_t - x^*\|^2] + 25\bar{\eta}^2\sigma^2. \tag{8}$$

The proofs of Theorems 1, 2, 3, and 4 are provided in Appendix F.

**Main Takeaways:** From Theorems 1, 2, and 3, we note that `FedLin` matches the convergence guarantees of centralized gradient descent (up to constants) for smooth, strongly convex, convex, and non-convex settings, respectively. *As far as we are aware, this is the first work to provide such comprehensive guarantees under arbitrary objective and systems heterogeneity.* In fact, all our results continue to hold even when the operating speeds of the client machines vary across rounds, i.e., $\tau_i$ is allowed to be a function of $t$. Each client $i$ can simply adjust its learning rate $\eta_i \propto 1/\tau_i(t)$ *locally* to account for such variations. The bound for the noisy case in Theorem 4 resembles that of centralized SGD [41]: with a time-varying parameter $\bar{\eta}_t = O(1/t)$, we get the standard $O(1/T)$ rate after $T$ rounds (using the exact same arguments as in [41]). The key thing to note here is that despite arbitrary heterogeneity, the assumptions we make on the stochastic gradients are the same as those made in the analysis of centralized SGD: unbiased gradients with bounded variance, nothing more.

**Comparison with Related Work**: In the recent paper [10], the authors propose `FedSplit`, and analyze it in a deterministic setting. For strongly-convex and smooth loss functions, `FedSplit` guarantees linear convergence, but only to a *non-vanishing neighborhood* of $x^*$. Thus, like `FedAvg` [2], `FedProx` [22], and `FedNova` [23], `FedSplit` fails to guarantee *exact* linear convergence to $x^*$. Empirically, we observe that `FedSplit` diverges on certain instances; see Appendix J. Compared to

these algorithms, we see from Theorem 1 that `FedLin` guarantees linear convergence to $x^*$. Notably, the linear convergence rate we obtain in Theorem 1 under *both* objective and systems heterogeneity is the *best rate we know of in FL*, and matches that of `SCAFFOLD` [11] where only objective heterogeneity is considered.[3] The model of systems heterogeneity we study is taken from [23], where the authors provide guarantees only for the non-convex case under a bounded dissimilarity assumption. In contrast, our results cover all the three standard settings - strongly-convex, convex, and non-convex - without requiring any bounded dissimilarity assumption. For further related work on straggler-robust distributed learning algorithms (without objective heterogeneity or local steps), see [43–48].

## 4.1 The Price of Infrequent Communication

In this section, we take a closer look at the effect of performing multiple local steps on the convergence rate. To do so, we assume that all clients perform the same number of local steps $H$, i.e., there is no communication for $H$ consecutive time-steps between two communication rounds. Now consider a centralized baseline where each client can communicate with every other client at all times (i.e., even between rounds). In this case, since each client can always access $\nabla f(x)$, gradient descent yields

$$f(\bar{x}_{T+1}) - f(x^*) \leq \exp(-\frac{1}{\kappa}TH)(f(\bar{x}_1) - f(x^*)) \tag{9}$$

after $T$ rounds, with $H$ synchronized local iterations within each round. Based on Theorem 1, observe that we lose out by a factor of $H$ in the exponent relative to the centralized baseline. Notably, both in the centralized case, and in `FedLin`, each client queries the gradient of its local objective $H$ times in each round, thereby making $TH$ gradient queries over $T$ rounds. Thus, relative to a centralized baseline, `FedLin` incurs the same computational cost in terms of gradient queries, and reduces communication by a factor of $H$, at the expense of a convergence rate that is slower by a factor of $H$. We emphasize here that just as with `FedLin`, $H$ does not show up in the convergence rate (exponent) of algorithms like `FedSplit` [10] and `SCAFFOLD` [11] either.

The primary reason for the slower convergence rate (relative to a centralized baseline) stems from the need to set $\eta \propto 1/H$ to mitigate client-drift under objective heterogeneity. At this stage, one may conjecture that the above requirement is simply an artifact of a conservative analysis of Algorithm 1, and that a more refined analysis will reveal the utility of performing more local steps even in the heterogeneous setting. Our next result suggests otherwise; for a proof, see Appendix E.

**Theorem 5.** *(Lower bound for `FedLin`) Suppose $\delta_c = \delta_s = 1$, and $\tau_i = H, \eta_i = \eta, \forall i \in \mathcal{S}$. Then, given any $L \geq 14$ and $H \geq 2$, there exists an instance involving 2 clients where each $f_i(x), i \in \{1, 2\}$, is 1-strongly convex and $L$-smooth, and an initial condition $\bar{x}_1$, such that `FedLin` initialized from $\bar{x}_1$ generates a sequence of iterates $\{\bar{x}_t\}$ satisfying the following for any $T \geq 1$:*

$$\|\bar{x}_{T+1} - x^*\|^2 \geq \exp(-4T)\|\bar{x}_1 - x^*\|^2; f(\bar{x}_{T+1}) - f(x^*) \geq \exp(-4T)(f(\bar{x}_1) - f(x^*)). \tag{10}$$

**Main Takeaways:** There are several key implications of Theorem 5. First, it complements Theorem 1 by providing a matching lower bound. *We believe ours is the first work to provide a tight linear convergence rate analysis*: [11] and [10] only provide upper-bounds for `SCAFFOLD` and `FedSplit`, respectively. Second, our analysis of Theorem 5 in Appendix E indicates that there are problem instances where setting $\eta \propto 1/H$ is in fact *necessary* to guarantee convergence to $x^*$. As a result, for such problem instances, no matter how many local steps $H$ each client performs, the error at the end of $T$ rounds remains bounded below by an $H$-independent quantity, as is apparent from (10). Perhaps surprisingly, we show in Appendix E that the lower bound in Theorem 5 even applies to simple instances with non-identical quadratic losses (across clients) *where every $f_i(x)$ has the same minimum!* This is particularly insightful since it highlights the limitations of exploiting stale gradient terms in the local update rule (as is done in both `FedLin` and `SCAFFOLD`), and suggests the need for more informed updating schemes that explicitly take into account the level of statistical heterogeneity.

**Proof Idea for Theorem 5:** To establish Theorem 5, we set up an instance involving two clients with quadratic loss functions. Our main idea is to relate the convergence of `FedLin` to the Schur stability of an appropriate discrete-time linear time-invariant (LTI) system. Based on this connection, we show that guaranteeing stability necessitates setting $\eta \propto 1/H$, which immediately leads to the lower bound. We believe that the same technique can be used to establish a similar lower bound for `SCAFFOLD`.

---

[3] In a concurrent work [42], the authors develop a linearly converging algorithm called `S-Local-SVRG` for the finite-sum setting, but neither consider systems heterogeneity nor compression. Moreover, unlike the lower bound we develop for `FedLin` in Theorem 5, no lower bounds are provided for `S-Local-SVRG` in [42].

## 5 Gradient Sparsification at Server

In this section, our focus will be on addressing the following question: *For strongly convex and smooth deterministic functions, and in the presence of both objective and systems heterogeneity, can we still hope for linear convergence to $x^*$ when gradients are sparsified at the server?* Interestingly, we will show that not only is it possible to converge linearly to $x^*$, it is possible to do so *without any error-feedback*. Moreover, this claim holds regardless of how aggressive the server is in its sparsification scheme: it may even transmit just a single component of the aggregated gradient vector.

To isolate the impact of server-level sparsification, we will assume throughout this section that gradients are not sparsified at the clients, i.e., $\delta_c = 1$. Consequently, $h_{i,t+1} = \nabla f_i(\bar{x}_{t+1}), \forall i \in \mathcal{S}, \forall t \in \{1, \ldots, T\}$. We begin by considering a simpler variant of FedLin with no error-feedback at the server side, i.e., line 15 is skipped, and $g_{t+1}$ in line 14 of Algo. 1 is instead updated as follows

$$g_{t+1} = \mathcal{C}_{\delta_s}\left(\frac{1}{m}\sum_{i \in \mathcal{S}}\nabla f_i(\bar{x}_{t+1})\right) = \mathcal{C}_{\delta_s}\left(\nabla f(\bar{x}_{t+1})\right). \tag{11}$$

**Theorem 6.** (*Sparsification at server with no error-feedback*) *Suppose each $f_i(x)$ is $L$-smooth and $\mu$-strongly convex. Moreover, suppose $\tau_i \geq 1, \forall i \in \mathcal{S}$, and $\delta_c = 1$. Consider a variant of FedLin, where line 14 is replaced by equation* (11)*, and line 15 is skipped, i.e., there is no error-feedback. Then, with $\eta_i = \frac{1}{2(2+\sqrt{\delta_s})L\tau_i}, \forall i \in \mathcal{S}$, this variant of FedLin guarantees*

$$f(\bar{x}_{T+1}) - f(x^*) \leq \left(1 - \frac{1}{2\delta_s\left(2+\sqrt{\delta_s}\right)\kappa}\right)^T (f(\bar{x}_1) - f(x^*)).$$

**Main Takeaways:** From Theorem 6, we see that *even without error-feedback, it is possible to linearly converge to $x^*$*; the rate of convergence, however, is inversely proportional to $\delta_s^{\frac{3}{2}}$. Thus, Theorem 6 quantifies the trade-off between the level of sparsification at the server, and the rate of convergence. When there is no gradient compression, i.e., when $\delta_s = 1$, we exactly recover Theorem 1.

One may ask: *Is there any potential benefit to employing error-feedback when gradients are sparsified at the server?* Our next result answers this question in the affirmative.

**Theorem 7.** (*Sparsification at server with error-feedback*) *Suppose each $f_i(x)$ is $L$-smooth and $\mu$-strongly convex. Moreover, suppose $\tau_i \geq 1, \forall i \in \mathcal{S}$, and $\delta_c = 1$. Let the step-size for client $i$ be chosen as $\eta_i = \frac{1}{72L\delta_s\tau_i}$. Then, FedLin guarantees:*

$$f(\bar{x}_{T+1}) - f(x^*) \leq 2\kappa\left(1 - \frac{1}{96\delta_s\kappa}\right)^T (f(\bar{x}_1) - f(x^*)).$$

For proofs of Theorems 6 and 7, see Appendix G and I.

**Main Takeaways**: Comparing the guarantee of Theorem 6 with that of Theorem 7, we note that the convergence rate is inversely proportional to $\delta_s^{\frac{3}{2}}$ in the former, and inversely proportional to $\delta_s$ in the latter. Thus, the main message here is that *employing error-feedback leads to a faster convergence rate by improving the dependence of the rate on $\delta_s$.*

## 6 Gradient Sparsification at Clients

In this section, we will turn our attention to the case when gradients are sparsified at the clients prior to being transmitted to the server. Throughout this section, we will assume that gradients are not compressed any further at the server side, i.e., $\delta_s = 1$. To proceed, we will need to make the following bounded gradient dissimilarity assumption.

**Assumption 1.** *There exist constants $C \geq 1$ and $D \geq 0$ such that the following holds $\forall x \in \mathbb{R}^d$:*

$$\frac{1}{m}\sum_{i=1}^m \|\nabla f_i(x)\|^2 \leq C\|\nabla f(x)\|^2 + D. \tag{12}$$

The following is the main result of this section; for a proof, see Appendix H.

**Theorem 8.** *(Sparsification at clients with error-feedback)* *Suppose each $f_i(x)$ is L-smooth and $\mu$-strongly convex, and suppose Assumption 1 holds. Moreover, suppose $\tau_i \geq 1, \forall i \in \mathcal{S}$, and $\delta_s = 1$. Let the step-size for client $i$ be chosen as $\eta_i = \frac{\bar{\eta}}{\tau_i}$, where $\bar{\eta} \in (0,1)$ satisfies $\bar{\eta} \leq \frac{1}{72L\delta_c C}$. Then, FedLin guarantees:*

$$\|\bar{x}_{T+1} - x^*\|^2 \leq 2\left(1 - \frac{3}{4}\bar{\eta}\mu\right)^T \|\bar{x}_1 - x^*\|^2 + \frac{16}{3}\bar{\eta}\left(\frac{6}{\delta_c C} + \delta_c\right)\frac{D}{\mu}. \tag{13}$$

**Main Takeaways:** Intuitively, one would expect that sparsifying gradients at each client prior to aggregation at the server would inject more errors than when gradients are first accurately aggregated at the server, and then the aggregated gradient vector is sparsified: Theorems 6 and 8 support this intuition. For the former, we neither required error-feedback nor Assumption 1 to guarantee linear convergence to the global minimum $x^*$; for the latter, even with error-feedback and the bounded gradient dissimilarity assumption, we can establish linear convergence to *only a neighborhood of $x^*$*, in general. From (13), we note that the size of this neighborhood scales linearly with $D$ - a measure of objective heterogeneity. In particular, when $D = 0$, the iterates $\bar{x}_t$ converge *exactly* to $x^*$.

**Remark 1.** *To the best of our knowledge, our results in Sections 5 and 6 constitute the first formal analysis of biased gradient sparsification in FL. In particular, we significantly generalize the recent results in [49] for a single worker to a multi-client FL setting with both objective and systems heterogeneity. To arrive at these results, we develop a new potential-function based proof technique in Appendix H. For more related work on compression in distributed learning, see Appendix A.*

**Extensions:** We studied the effect of compressing information at the server and at the clients separately, with the goal of identifying the key differences between each of these mechanisms. The analysis techniques we developed in the process pave the way for studying various natural extensions: (i) combined sparsification at both the clients and the server; (ii) gradient sparsification in tandem with model parameter compression; and (iii) stochastic counterparts of Theorems 6, 7, and 8.

## 7 Experimental Results

In this section, we provide numerical results for FedLin on a least squares problem to validate our theory. In Appendix K, we also provide additional numerical results on a logistic regression problem.

For now, we consider the following least squares regression problem:

$$\min_{x \in \mathbb{R}^d} f(x) = \min_{x \in \mathbb{R}^d} \frac{1}{m} \sum_{i=1}^{m} \frac{1}{2}\|A_i x - b_i\|^2, \tag{14}$$

where $A_i \in \mathbb{R}^{500 \times 100}$ is a design matrix and $b_i \in \mathbb{R}^{500}$ is a response vector. The client objective functions, $f_i(x)$ are strongly convex. Assuming that all design matrices are full column rank, problem (14) admits a unique minimizer. To generate synthetic data, for each client $i \in \mathcal{S} = \{1, \dots, 20\}$, we generate $A_i$ and $b_i$ according to the model $b_i = A_i x_i + \varepsilon_i$, where $x_i$ is a weight vector and $\varepsilon_i \in \mathbb{R}^{500}$ is a disturbance. In particular, we generate $[A_i]_{jk} \overset{i.i.d.}{\sim} \mathcal{N}(0,1)$, and $\varepsilon_i \sim \mathcal{N}(0, 0.5I_{500})$, $\forall i \in \mathcal{S}$. To capture statistical heterogeneity, the entries of the local true parameter of client $i$ are modeled as $[x_i]_k \sim \mathcal{N}(u_i, 1)$, $k \in \{1, \dots, 100\}$, where $u_i \sim \mathcal{N}(0, \alpha)$ and $\alpha \geq 0$. Hence, $\alpha$ controls the level of statistical heterogeneity. To model the effect of systems heterogeneity, for each client $i \in \mathcal{S}$, the number of local steps is drawn uniformly and independently from $[2, 100]$. We will primarily focus on a deterministic setting here for our experiments; in Appendix L, we evaluate FedLin on a standard stochastic oracle model. Our experiments in Appendix L reveal that under a noisy oracle, FedLin guarantees linear convergence to a ball around the true minimum, exactly as suggested by Thm. 4.

**Gradient Sparsification at Server.** We first consider a variant of FedLin where gradient sparsification is implemented only at the server side and without any error-feedback. In particular, we consider the cases where $\delta_s \in \{2, 4\}$, which correspond to the implementation of a TOP-50 and a TOP-25 operator, respectively. For comparison, we also plot the resulting performance when no gradient sparsification is implemented at the server. To examine the effect of statistical heterogeneity on the performance of FedLin, we generate two synthetic datasets corresponding to two different levels of heterogeneity in the clients' local objectives, namely $\alpha = 10$ and $\alpha = 50$. As illustrated in Fig. 2, irrespective of the level of gradient sparsification on the server side, FedLin achieves linear convergence to the true minimum in the presence of both objective and systems heterogeneity, confirming Theorem 6. Also, both the convergence speed and accuracy of FedLin remain unaffected as the level of heterogeneity in the clients' objective functions increases.

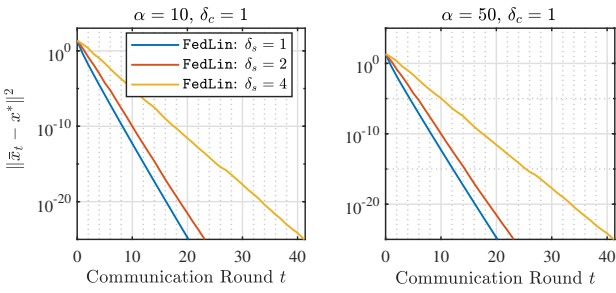

Figure 2: Server-side sparsification results for FedLin. The constant $\bar{\eta}$ is fixed at $10^{-2}$. **Left**: $\alpha = 10$. **Right**: $\alpha = 50$.

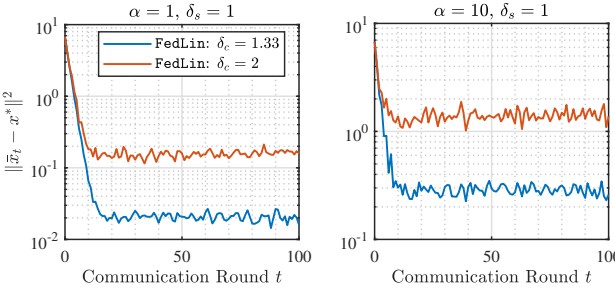

Figure 3: Client-side sparsification results for FedLin. The constant $\bar{\eta}$ is fixed at $5 \times 10^{-4}$. **Left**: $\alpha = 1$. **Right**: $\alpha = 10$.

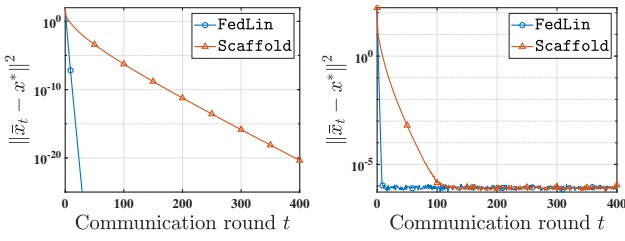

Figure 4: Comparison of FedLin with SCAFFOLD. **(Left)** Deterministic setting. **(Right)** General stochastic oracle model: unbiased gradients with variance $\sigma = 10^{-1}$.

**Gradient Sparsification at Clients.** Next, we implement gradient sparsification only at the clients' side, i.e. $\delta_s = 1$. In particular, we consider the cases where $\delta_c \in \{4/3, 2\}$, which correspond to the implementation of a TOP-75 and a TOP-50 operator, respectively. Once again, we generate two synthetic datasets with different levels of objective heterogeneity, namely $\alpha = 1$ and $\alpha = 10$. As illustrated in Fig. 3, unlike the server case, FedLin with sparsification at the clients' side converges linearly, but with a non-vanishing error that increases as the value of $\delta_c$ increases. This aligns with the conclusions of Theorem 8. Furthermore, the level of objective heterogeneity has a direct impact on the convergence error. In particular, for the same level of gradient sparsification, higher levels of objective heterogeneity result in larger values of the convergence error.

**Comparison with SCAFFOLD.** We now compare FedLin with SCAFFOLD on the least squares regression setup described above. To make a fair comparison, we assume that there is no systems heterogeneity or gradient compression. For implementing SCAFFOLD, we use Option II in their paper [11] for updating the control variates. We set the number of local steps $H = 20$, the statistical heterogeneity parameter $\alpha = 10$, and use a step-size of $10^{-3}$ for both algorithms (the step-size was tuned to get best results). For the deterministic setting, we note from Fig. 4 that FedLin converges much faster compared to SCAFFOLD. This trend persists when we perturb the gradients with zero-mean Gaussian noise with variance $\sigma = 10^{-1}$. We conjecture that the faster convergence of FedLin stems from the fact that it uses less stale gradient correction terms relative to the control variates of SCAFFOLD; see the discussion about the fixed point property of FedLin in Sec. 3.

## 8 Conclusion

We developed a novel algorithmic framework called FedLin to tackle some of the key challenges in FL, namely objective heterogeneity, systems heterogeneity, and imprecise communication. We showed that FedLin enjoys strong theoretical guarantees: (i) FedLin matches centralized rates, and, in particular, guarantees linear convergence to the global minimum under *arbitrary* objective and systems heterogeneity; and (ii) preserves linear convergence rates despite aggressive gradient sparsification. We also established a tight lower-bound for FedLin, highlighting that even mild statistical heterogeneity can end up hurting convergence rates - this is the first such result in FL. Our current approach requires two passes of communication between the clients and the server in each round. Moreover, our analysis does not account for partial client participation. As future work, we plan to address these limitations. We also plan to investigate other federated learning formulations (beyond supervised learning) where statistical heterogeneity can potentially help.

**Acknowledgement:** This work was supported by NSF Award 1837253, NSF CAREER award CIF 1943064, and the Air Force Office of Scientific Research Young Investigator Program (AFOSR-YIP) under award FA9550-20-1-0111.

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
