# A Related Work on Compression Techniques in Distributed Optimization and Learning

In order to alleviate the communication bottleneck in large-scale distributed computing systems, a complementary direction to reducing communication rounds is to compress the information being exchanged. Broadly speaking, compression either involves quantization [33, 50, 27, 26, 28–31, 15, 32] to reduce the precision of transmitted information, or biased sparsification [24, 25, 35, 34, 51, 52, 49, 53] to transmit only a few components of a vector with the largest magnitudes. In terms of quantization, while some of the earlier work focused on quantizing gradients [27, 26, 28], the convergence guarantees of these methods were subsequently improved in [29] where the authors proposed DIANA - a new algorithm based on quantizing *differences of gradients*. The DIANA technique was further generalized in [31] to account for a variety of compressors. An accelerated variant of DIANA was also proposed recently in [30]. Notably, unlike our work, neither DIANA nor its variants [31, 30] consider the effect of local steps.

As far as gradient sparsification is concerned, although empirical studies [24, 25] have revealed the benefits of aggressive sparsification techniques, theoretical guarantees for such methods, especially in a distributed setting, are few and far between. In our work, we study one such sparsification technique - the TOP-k operator - by exploiting the error-feedback mechanism. Notably, the error-feedback idea that we use was first introduced in [33] to study 1-bit SGD. Follow-up work has explored this idea for studying SignSGD in [51], and sparse SGD in [35, 34, 52] - all for single-worker settings. Very recently, [49] and [53] provide theoretical results on biased sparsification for a master-worker type distributed architecture; however, their analysis does not account for the effect of local steps.

# B  Proof of Propositions 1 and 2

The analysis in this section follows the techniques introduced in [8].

**Proof of Proposition 1: FedProx**

We consider a deterministic version of `FedProx` where clients perform $H$ local iterations according to the update rule in equation (2). Let

$$g_{i,\ell}^{(t)} \triangleq \nabla f_i(x_{i,\ell}^{(t)}) + \beta(x_{i,\ell}^{(t)} - \bar{x}_t), q_i^{(t)} \triangleq \sum_{\ell=0}^{H-1} g_{i,\ell}^{(t)}, \text{ and } q^{(t)} \triangleq \frac{1}{m} \sum_{i \in \mathcal{S}} q_i^{(t)}.$$

Then, Lemma 1 provides a recursive relationship in terms of $g_{i,\ell}^{(t)}$.

**Lemma 1.** *For each $i \in \mathcal{S}$, the following holds for all $\ell \in \{0, \cdots, H-1\}$:*

$$g_{i,\ell+1}^{(t)} = [I - \eta(A_i + \beta I)]g_{i,\ell}^{(t)}. \tag{15}$$

*Proof.* By definition of $g_{i,\ell}^{(t)}$, we have

$$g_{i,\ell}^{(t)} = (A_i + \beta I)x_{i,\ell}^{(t)} - A_i c_i - \beta \bar{x}_t. \tag{16}$$

Hence, we may write

$$x_{i,\ell}^{(t)} = (A_i + \beta I)^{-1}\left[g_{i,\ell}^{(t)} + A_i c_i + \beta \bar{x}_t\right], \tag{17}$$

where $(A_i + \beta I)$ is positive-definite since $A_i$ is positive-definite and $\beta \geq 0$. Plugging equation (17) into the local update rule, we get

$$\begin{aligned}
x_{i,\ell+1}^{(t)} &= x_{i,\ell}^{(t)} - \eta g_{i,\ell}^{(t)} \\
&= \left[(A_i + \beta I)^{-1} - \eta I\right]g_{i,\ell}^{(t)} + (A_i + \beta I)^{-1}\left(A_i c_i + \beta \bar{x}_t\right).
\end{aligned} \tag{18}$$

Combining equations (16) and (18), we get

$$\begin{aligned}
g_{i,\ell+1}^{(t)} &= (A_i + \beta I)x_{i,\ell+1}^{(t)} - A_i c_i - \beta \bar{x}_t \\
&= [I - \eta(A_i + \beta I)]g_{i,\ell}^{(t)}.
\end{aligned}$$

$\square$

For each client $i \in \mathcal{S}$, define the client distortion matrix

$$Q_i = \sum_{\ell=0}^{H-1} [I - \eta(A_i + \beta I)]^\ell, \tag{19}$$

and the client surrogate function

$$\tilde{f}_i(x) = \frac{1}{2}\|(Q_i A_i)^{1/2}(x - c_i)\|^2. \tag{20}$$

The global surrogate function $\tilde{f}(x)$ is hence defined as

$$\tilde{f}(x) = \frac{1}{m} \sum_{i \in \mathcal{S}} \tilde{f}_i(x) = \frac{1}{m} \sum_{i \in \mathcal{S}} \frac{1}{2}\|(Q_i A_i)^{1/2}(x - c_i)\|^2. \tag{21}$$

We now provide the proof of Proposition 1.

*Proof.* From Lemma 1, we have

$$g_{i,\ell}^{(t)} = [I - \eta(A_i + \beta I)]^\ell g_{i,0}^{(t)}$$
$$= [I - \eta(A_i + \beta I)]^\ell \nabla f_i(\bar{x}_t). \tag{22}$$

Hence, we may write

$$q_i^{(t)} = \sum_{\ell=0}^{H-1} g_{i,\ell}^{(t)}$$
$$= \sum_{\ell=0}^{H-1} [I - \eta(A_i + \beta I)]^\ell \nabla f_i(\bar{x}_t)$$
$$= Q_i \nabla f_i(\bar{x}_t)$$
$$= \nabla \tilde{f}_i(\bar{x}_t). \tag{23}$$

Moreover, by definition,

$$q(t) = \frac{1}{m} \sum_{i \in \mathcal{S}} q_i^{(t)}$$
$$= \frac{1}{m} \sum_{i \in \mathcal{S}} \nabla \tilde{f}_i(\bar{x}_t)$$
$$= \nabla \tilde{f}(\bar{x}_t), \tag{24}$$

where the second equality follows from equation (23), and the last one follows from the definition of the global surrogate function. As per equation (2), the global iterates in `FedProx` are updated as follows

$$\bar{x}_{t+1} = \bar{x}_t - \eta q^{(t)}$$
$$= \bar{x}_t - \eta \nabla \tilde{f}(\bar{x}_t). \tag{25}$$

Equation (25) is equivalent to performing one step of GD on the surrogate function $\tilde{f}(x)$ with a starting point $\bar{x}_t$. It follows that $T$ communication rounds of `FedProx` are equivalent to performing $T$ steps of parallel GD with $m$ workers on the global surrogate function $\tilde{f}(x)$. □

In what follows, we analyze some properties of the client surrogate functions $\tilde{f}_i(x)$.

**Lemma 2.** *For $\eta < \dfrac{1}{L+\beta}$, the client surrogate function $\tilde{f}_i(x)$ shares the same unique minimizer as the true local function $f_i(x)$, that is, $\tilde{x}_i^* = x_i^* = c_i, \forall i \in \mathcal{S}$, where $\tilde{x}_i^*$ is the minimizer of $\tilde{f}_i^*(x)$.*

*Proof.* Let $(\lambda_i, v_i)$ be a tuple of eigenvalue $\lambda_i$, and corresponding eigenvector $v_i$, of $A_i$, where $A_i$ is symmetric positive-definite. Then, $(1 - \eta(\lambda_i + \beta), v_i)$ is a tuple of eigenvalue and corresponding eigenvector of $D \triangleq [I - \eta(A_i + \beta)]$. For $0 < \eta \le \dfrac{1}{L+\beta} < \dfrac{1}{L_i + \beta}, \forall i \in \mathcal{S}$, we have $0 < 1 - \eta(\lambda_i + \beta) < 1$, and hence, $D$ is a symmetric positive-definite matrix. For any integer $\ell \ge 1$, $([1 - \eta(\lambda_i + \beta)]^\ell, v_i)$ is a tuple of eigenvalue and corresponding eigenvector of $D^\ell \triangleq [I - \eta(A_i + \beta)]^\ell$, and hence, $D^\ell$ is also a symmetric positive-definite matrix since $0 < \eta < \dfrac{1}{L_i + \beta}$. From equation (19), we have $Q_i = \sum_{\ell=0}^{H-1} [I - \eta(A_i + \beta)]^\ell = \sum_{\ell=0}^{H-1} D^\ell$. Hence, $(\sum_{\ell=0}^{H-1} [1 - \eta(\lambda_i + \beta)]^\ell, v_i)$ is a tuple of eigenvalue and corresponding eigenvector of $Q_i$ and, consequently, $Q_i$ is a symmetric positive-definite matrix. Based on the above results, since both $A_i$ and $Q_i$ are symmetric positive-definite matrices that are simultaneously diagonalizable, it follows that they commute. Hence, for $0 < \eta < \dfrac{1}{L+\beta}$, the product $Q_i A_i$ is a symmetric positive-definite matrix and the client surrogate function $\tilde{f}_i(x)$ admits a unique minimizer $\tilde{x}_i^* = x_i^* = c_i$. □

**Proof of Proposition 2: FedNova**

The proof of Proposition 2 follows roughly the same steps as the proof of Proposition 1. In particular, we consider a deterministic version of FedNova where client $i$ performs $\tau_i$ local iterations according to the update rules provided in equation (4). Let $g_{i,\ell}^{(t)} \triangleq \nabla f_i(x_{i,\ell}^{(t)})$, $q_i^{(t)} \triangleq \sum_{\ell=0}^{\tau_i-1} \alpha_i g_{i,\ell}^{(t)}$, and $q(t) \triangleq \frac{1}{m} \sum_{i \in \mathcal{S}} q_i^{(t)}$. Then, Lemma 3 provides a recursive relationship in terms of $g_{i,\ell}^{(t)}$.

**Lemma 3.** *For every $i \in \mathcal{S}$, we have for all $\ell \in \{0, \cdots, \tau_i - 1\}$:*

$$g_{i,\ell+1}^{(t)} = [I - \eta A_i] g_{i,\ell}^{(t)}. \tag{26}$$

For each client $i \in \mathcal{S}$, define the client distortion matrix

$$Q_i = \sum_{\ell=0}^{\tau_i-1} [I - \eta A_i]^\ell \alpha_i. \tag{27}$$

Let the client and global surrogate functions, namely $\tilde{f}_i(x)$ and $\tilde{f}(x)$, be defined as before. Then, the proof of Proposition 2 follows directly from Lemma 3. Finally, Lemma 4 provides additional insight on the client surrogate functions $\tilde{f}_i(x)$. Once again, the proof is omitted as it closely follows the steps of the proof of Lemma 2.

**Lemma 4.** *For $\eta < \frac{1}{L}$, the client surrogate function $\tilde{f}_i(x)$ shares the same unique minimizer as the true local function $f_i(x)$, that is, $\tilde{x}_i^* = x_i^* = c_i, \forall i \in \mathcal{S}$.*

It should be noted that while $f_i(x)$ and $\tilde{f}_i(x)$ share the same minimizer, the minimizer of the surrogate global function may be far off from the true minimizer $x^*$.

## C Useful Results and Facts

In this section, we will compile some results that will prove to be useful later in our analysis. We start by assembling some well-known facts about convex and smooth functions [54, 55].

- (**Smoothness**): Suppose $f(x)$ is $L$-smooth. Then, by definition, the following inequalities hold for any two points $x, y \in \mathbb{R}^d$:

$$\|\nabla f(x) - \nabla f(y)\| \leq L\|x - y\|, \text{ and} \tag{28}$$

$$f(y) - f(x) \leq \langle y - x, \nabla f(x) \rangle + \frac{L}{2}\|y - x\|^2. \tag{29}$$

- (**Smoothness**): Suppose $f(x)$ is $L$-smooth, and $x^* \in \operatorname{argmin}_{x \in \mathbb{R}^d} f(x)$. Then, we can upper-bound the magnitude of the gradient at any given point $y \in \mathbb{R}^d$ in terms of the objective sub-optimality at $y$, as follows:

$$\|\nabla f(y)\|^2 \leq 2L(f(y) - f(x^*)). \tag{30}$$

- (**Smoothness and Convexity**): Suppose $f(x)$ is $L$-smooth and convex. Then, the following holds for any two points $x, y \in \mathbb{R}^d$:

$$\langle \nabla f(y) - \nabla f(x), y - x \rangle \geq \frac{1}{L}\|\nabla f(y) - \nabla f(x)\|^2. \tag{31}$$

- (**Strong convexity**): Suppose $f(x)$ is $\mu$-strongly convex. Then, by definition, the following inequality holds for any two points $x, y \in \mathbb{R}^d$:

$$f(y) - f(x) \geq \langle y - x, \nabla f(x) \rangle + \frac{\mu}{2}\|y - x\|^2. \tag{32}$$

Using the above inequality, one can easily conclude that

$$\langle \nabla f(y) - \nabla f(x), y - x \rangle \geq \mu\|y - x\|^2. \tag{33}$$

- (**Strong convexity**): Suppose $f(x)$ is $\mu$-strongly convex, and $x^* = \operatorname{argmin}_{x \in \mathbb{R}^d} f(x)$. Then, we can lower-bound the magnitude of the gradient at any given point $y \in \mathbb{R}^d$ in terms of the objective sub-optimality at $y$, as follows:

$$\|\nabla f(y)\|^2 \geq 2\mu(f(y) - f(x^*)). \tag{34}$$

In addition to the above results, we will have occasion to make use of the following facts.

- Given any two vectors $x, y \in \mathbb{R}^d$, the following holds for any $\gamma > 0$:

$$\|x + y\|^2 \leq (1 + \gamma)\|x\|^2 + \left(1 + \frac{1}{\gamma}\right)\|y\|^2. \tag{35}$$

- Given $m$ vectors $x_1, \ldots, x_m \in \mathbb{R}^d$, the following is a simple application of Jensen's inequality:

$$\left\|\sum_{i=1}^m x_i\right\|^2 \leq m \sum_{i=1}^m \|x_i\|^2. \tag{36}$$

# D   Analysis under Objective Heterogeneity: FedLin resolves the Speed-Accuracy Conflict

In this section, we start by presenting a convergence analysis for FedLin that focuses solely on the aspect of heterogeneity in the clients' local objective functions. We do so to set up the basic proof structure that we will later build on for analyzing more involved settings. With this in mind, we will assume throughout this section that all clients perform the same number of local updates, i.e., $\tau_i = H, \forall i \in \mathcal{S}$. Additionally, we will assume that there is no gradient sparsification, i.e., $\delta_c = \delta_s = 1$. Based on the second assumption, observe that $\rho_{i,t} = 0, e_t = 0, \forall i \in \mathcal{S}, \forall t \in \{1, \ldots, T\}$. Thus, the local update rule for each client in line 5 of FedLin simplifies to

$$x_{i,\ell+1}^{(t)} = x_{i,\ell}^{(t)} - \eta_i(\nabla f_i(x_{i,\ell}^{(t)}) - \nabla f_i(\bar{x}_t) + \nabla f(\bar{x}_t)). \tag{37}$$

Let us denote by $\kappa = L/\mu$ the condition number of an $L$-smooth and $\mu$-strongly convex function.

**Theorem 9.** (***Heterogeneous Setting***) *Suppose each $f_i(x)$ is $L$-smooth and $\mu$-strongly convex. Moreover, suppose $\tau_i = H, \forall i \in \mathcal{S}$, and $\delta_c = \delta_s = 1$. Then, with $\eta_i = \eta = \frac{1}{6LH}, \forall i \in \mathcal{S}$, FedLin guarantees:*

$$f(\bar{x}_{T+1}) - f(x^*) \leq \left(1 - \frac{1}{6\kappa}\right)^T (f(\bar{x}_1) - f(x^*)). \tag{38}$$

When all clients optimize the same loss function, we have the following result.

**Proposition 3.** (***Homogeneous setting***) *Suppose all client objective functions are identical, i.e., $f_i(x) = f(x), \forall x \in \mathbb{R}^d, \forall i \in \mathcal{S}$, and $f(x)$ is $L$-smooth and $\mu$-strongly convex. Moreover, suppose $\tau_i = H, \forall i \in \mathcal{S}$, and $\delta_c = \delta_s = 1$. Then, with $\eta_i = \eta = \frac{1}{L}, \forall i \in \mathcal{S}$, FedLin guarantees:*

$$f(\bar{x}_{T+1}) - f(x^*) \leq \left(1 - \frac{1}{\kappa}\right)^{TH} (f(\bar{x}_1) - f(x^*)). \tag{39}$$

We first provide the proof of Proposition 3.

*Proof.* (**Proposition 3**) The proof follows from two simple observations. First, note that since $\nabla f_i(x) = \nabla f(x), \forall x \in \mathbb{R}^d$, the local update rule of each client $i \in \mathcal{S}$ reduces to $x_{i,\ell+1}^{(t)} = x_{i,\ell}^{(t)} - \eta \nabla f(x_{i,\ell}^{(t)})$. Second, based on the steps of FedLin, observe that the local iterates of the clients remain synchronized within each communication round, i.e., for any $t \in \{1, \ldots, T\}$, and any two clients $i, j \in \mathcal{S}$, it holds that $x_{i,\ell}^{(t)} = x_{j,\ell}^{(t)}, \forall \ell \in \{0, \ldots, H-1\}$. Thus, FedLin boils down to $m$ parallel and identical implementations of gradient descent on the global loss function $f(x)$, with $TH$ iterations performed by each client over $T$ rounds. The claim of the proposition then follows from standard results of centralized gradient descent applied to strongly convex and smooth deterministic objectives. $\square$

## D.1   Proof of Theorem 9

Let us fix a communication round $t \in \{1, \ldots, T\}$. Our goal will be to derive an upper-bound on the change in the value of the objective function $f(\cdot)$ over round $t$, i.e., we will be interested in bounding the quantity $f(\bar{x}_{t+1}) - f(\bar{x}_t)$. To lighten the notation, we will drop the superscript $t$ on the local iterates $x_{i,\ell}^{(t)}$, and simply refer to them as $x_{i,\ell}$; it should be understood from context that all such iterates pertain to round $t$. Given that $\delta_c = \delta_s = 1$, and $\eta_i = \eta, \forall i \in \mathcal{S}$, the local update rule in line 5 of FedLin takes the following simplified form:

$$x_{i,\ell+1} = x_{i,\ell} - \eta(\nabla f_i(x_{i,\ell}) - \nabla f_i(\bar{x}_t) + \nabla f(\bar{x}_t)). \tag{40}$$

Based on the above rule, and smoothness of each $f_i(x)$, the following lemma provides an upper-bound on $f(\bar{x}_{t+1}) - f(\bar{x}_t)$.

**Lemma 5.** *Suppose each $f_i(x)$ is $L$-smooth. Moreover, suppose $\tau_i = H, \forall i \in \mathcal{S}$, and $\delta_c = \delta_s = 1$. Then,* `FedLin` *guarantees:*

$$f(\bar{x}_{t+1}) - f(\bar{x}_t) \leq -\eta H \left(1 - \eta L H\right) \|\nabla f(\bar{x}_t)\|^2 + \left(\frac{\eta L}{m} \sum_{i=1}^{m} \sum_{\ell=0}^{H-1} \|x_{i,\ell} - \bar{x}_t\|\right) \|\nabla f(\bar{x}_t)\|$$

$$+ \frac{\eta^2 L^3 H}{m} \sum_{i=1}^{m} \sum_{\ell=0}^{H-1} \|x_{i,\ell} - \bar{x}_t\|^2. \tag{41}$$

*Proof.* Based on (40) and the fact that $x_{i,0} = \bar{x}_t, \forall i \in \mathcal{S}$, we have:

$$x_{i,H} = \bar{x}_t - \eta \sum_{\ell=0}^{H-1} \nabla f_i(x_{i,\ell}) - \eta H (\nabla f(\bar{x}_t) - \nabla f_i(\bar{x}_t)), \forall i \in \mathcal{S}. \tag{42}$$

Thus,

$$\bar{x}_{t+1} = \frac{1}{m} \sum_{i=1}^{m} x_{i,H} = \bar{x}_t - \frac{\eta}{m} \sum_{i=1}^{m} \sum_{\ell=0}^{H-1} \nabla f_i(x_{i,\ell}) - \frac{\eta H}{m} \sum_{i=1}^{m} (\nabla f(\bar{x}_t) - \nabla f_i(\bar{x}_t))$$

$$= \bar{x}_t - \frac{\eta}{m} \sum_{i=1}^{m} \sum_{\ell=0}^{H-1} \nabla f_i(x_{i,\ell}), \tag{43}$$

where for the second equality, we used the fact that $\nabla f(y) = \frac{1}{m} \sum_{i=1}^{m} \nabla f_i(y), \forall y \in \mathbb{R}^d$. Now since each $f_i(\cdot)$ is $L$-smooth, it is easy to verify that $f(\cdot)$ is also $L$-smooth. From (29), we then obtain:

$$f(\bar{x}_{t+1}) - f(\bar{x}_t) \leq \langle \bar{x}_{t+1} - \bar{x}_t, \nabla f(\bar{x}_t) \rangle + \frac{L}{2} \|\bar{x}_{t+1} - \bar{x}_t\|^2$$

$$= -\frac{\eta}{m} \sum_{i=1}^{m} \sum_{\ell=0}^{H-1} \langle \nabla f_i(x_{i,\ell}), \nabla f(\bar{x}_t) \rangle + \frac{L}{2} \left\| \frac{\eta}{m} \sum_{i=1}^{m} \sum_{\ell=0}^{H-1} \nabla f_i(x_{i,\ell}) \right\|^2, \tag{44}$$

where in the last step we used (43). To proceed, we will now separately bound each of the two terms that appear in (44). For the first term, observe that

$$-\frac{\eta}{m}\sum_{i=1}^{m}\sum_{\ell=0}^{H-1}\langle\nabla f_i(x_{i,\ell}),\nabla f(\bar{x}_t)\rangle = -\eta\Big\langle\frac{1}{m}\sum_{i=1}^{m}\sum_{\ell=0}^{H-1}(\nabla f_i(x_{i,\ell})-\nabla f_i(\bar{x}_t))$$

$$+\frac{1}{m}\sum_{i=1}^{m}\sum_{\ell=0}^{H-1}\nabla f_i(\bar{x}_t),\nabla f(\bar{x}_t)\Big\rangle$$

$$=-\eta\Big\langle\frac{1}{m}\sum_{i=1}^{m}\sum_{\ell=0}^{H-1}(\nabla f_i(x_{i,\ell})-\nabla f_i(\bar{x}_t))$$

$$+H\nabla f(\bar{x}_t),\nabla f(\bar{x}_t)\Big\rangle$$

$$=-\frac{\eta}{m}\Big\langle\sum_{i=1}^{m}\sum_{\ell=0}^{H-1}(\nabla f_i(x_{i,\ell})-\nabla f_i(\bar{x}_t)),\nabla f(\bar{x}_t)\Big\rangle$$

$$-\eta H\|\nabla f(\bar{x}_t)\|^2$$

$$\overset{(a)}{\leq}\frac{\eta}{m}\left(\Big\|\sum_{i=1}^{m}\sum_{\ell=0}^{H-1}(\nabla f_i(x_{i,\ell})-\nabla f_i(\bar{x}_t))\Big\|\right)\|\nabla f(\bar{x}_t)\|$$

$$-\eta H\|\nabla f(\bar{x}_t)\|^2$$

$$\overset{(b)}{\leq}\frac{\eta}{m}\left(\sum_{i=1}^{m}\sum_{\ell=0}^{H-1}\|\nabla f_i(x_{i,\ell})-\nabla f_i(\bar{x}_t)\|\right)\|\nabla f(\bar{x}_t)\|$$

$$-\eta H\|\nabla f(\bar{x}_t)\|^2$$

$$\overset{(c)}{\leq}\frac{\eta L}{m}\left(\sum_{i=1}^{m}\sum_{\ell=0}^{H-1}\|x_{i,\ell}-\bar{x}_t\|\right)\|\nabla f(\bar{x}_t)\|-\eta H\|\nabla f(\bar{x}_t)\|^2.$$

(45)

In the above steps, (a) follows from the Cauchy-Schwartz inequality, (b) follows from the triangle inequality, and (c) follows from the fact that each $f_i(\cdot)$ is $L$-smooth (see (28)). Next, we bound the second term in (44) as follows.

$$\frac{L}{2}\Big\|\frac{\eta}{m}\sum_{i=1}^{m}\sum_{\ell=0}^{H-1}\nabla f_i(x_{i,\ell})\Big\|^2 = \frac{\eta^2 L}{2}\Big\|\frac{1}{m}\sum_{i=1}^{m}\sum_{\ell=0}^{H-1}(\nabla f_i(x_{i,\ell})-\nabla f_i(\bar{x}_t))+\frac{1}{m}\sum_{i=1}^{m}\sum_{\ell=0}^{H-1}\nabla f_i(\bar{x}_t)\Big\|^2$$

$$=\frac{\eta^2 L}{2}\Big\|\frac{1}{m}\sum_{i=1}^{m}\sum_{\ell=0}^{H-1}(\nabla f_i(x_{i,\ell})-\nabla f_i(\bar{x}_t))+H\nabla f(\bar{x}_t)\Big\|^2$$

$$\overset{(a)}{\leq}\eta^2 L\Big\|\frac{1}{m}\sum_{i=1}^{m}\sum_{\ell=0}^{H-1}(\nabla f_i(x_{i,\ell})-\nabla f_i(\bar{x}_t))\Big\|^2+\eta^2 H^2 L\|\nabla f(\bar{x}_t)\|^2$$

$$\overset{(b)}{\leq}\frac{\eta^2 L}{m}\sum_{i=1}^{m}\Big\|\sum_{\ell=0}^{H-1}(\nabla f_i(x_{i,\ell})-\nabla f_i(\bar{x}_t))\Big\|^2+\eta^2 H^2 L\|\nabla f(\bar{x}_t)\|^2$$

$$\overset{(c)}{\leq}\frac{\eta^2 L H}{m}\sum_{i=1}^{m}\sum_{\ell=0}^{H-1}\|\nabla f_i(x_{i,\ell})-\nabla f_i(\bar{x}_t)\|^2+\eta^2 H^2 L\|\nabla f(\bar{x}_t)\|^2$$

$$\overset{(d)}{\leq}\frac{\eta^2 L^3 H}{m}\sum_{i=1}^{m}\sum_{\ell=0}^{H-1}\|x_{i,\ell}-\bar{x}_t\|^2+\eta^2 H^2 L\|\nabla f(\bar{x}_t)\|^2.$$

(46)

In the above steps, (a) follows from (35) with $\gamma=1$, (b) and (c) both follow from (36), and (d) is a consequence of the $L$-smoothness of $f_i(\cdot)$. Combining the bounds in equations (45) and (46) immediately leads to the claim of the lemma. $\qquad\square$

To simplify the bound in (41), it is apparent that we need to estimate how much the client iterates $x_{i,\ell}$ drift off from their value $\bar{x}_t$ at the beginning of the communication round, i.e., we want to derive a bound on the quantity $\|x_{i,\ell} - \bar{x}_t\|$. To that end, we will make use of the following lemma.

**Lemma 6.** *Suppose $f(x)$ is $L$-smooth and convex. Then, for any $\eta \in (0,1)$ satisfying $\eta \leq \frac{1}{L}$, and any two points $x, y \in \mathbb{R}^d$, we have*

$$\|y - x - \eta(\nabla f(y) - \nabla f(x))\| \leq \|y - x\|. \tag{47}$$

*If $f(x)$ is $\mu$-strongly convex, then the above inequality is strict, i.e., $\exists \lambda \in (0,1)$, such that*

$$\|y - x - \eta(\nabla f(y) - \nabla f(x))\| \leq \lambda \|y - x\|. \tag{48}$$

*Proof.* Given any two points $x, y \in \mathbb{R}^d$, we have

$$
\begin{aligned}
\|y - x - \eta(\nabla f(y) - \nabla f(x))\|^2 &= \|y - x\|^2 - 2\eta\langle y - x, \nabla f(y) - \nabla f(x)\rangle + \eta^2\|\nabla f(y) - \nabla f(x)\|^2 \\
&\leq \|y - x\|^2 - \eta(2 - \eta L)\langle y - x, \nabla f(y) - \nabla f(x)\rangle \\
&\leq \|y - x\|^2,
\end{aligned}
\tag{49}
$$

where the first inequality follows from (31), and the second from the fact that $\eta L \leq 1$, and $\langle y - x, \nabla f(y) - \nabla f(x)\rangle \geq 0$; the latter is a consequence of the convexity of $f(\cdot)$. This establishes (47) for the convex, smooth case. When $f(\cdot)$ is $\mu$-strongly convex, we can further use (33) to obtain

$$\|y - x - \eta(\nabla f(y) - \nabla f(x))\|^2 \leq (1 - \eta\mu(2 - \eta L))\|y - x\|^2. \tag{50}$$

Noting that $\eta L \leq 1$ establishes (48) with $\lambda = \sqrt{1 - \eta\mu}$. $\qquad\square$

With the above lemma in hand, we will now show that the drift $\|x_{i,\ell} - \bar{x}_t\|$ can be bounded in terms of $\|\nabla f(\bar{x}_t)\|$ - a measure of the sub-optimality at the beginning of the communication round.

**Lemma 7.** *Suppose each $f_i(x)$ is $L$-smooth and $\mu$-strongly convex. Moreover, suppose $\tau_i = H, \forall i \in \mathcal{S}$, $\delta_c = \delta_s = 1$, and $\eta \leq \frac{1}{L}$. Then, `FedLin` guarantees the following bound for each $i \in \mathcal{S}$, and $\forall \ell \in \{0, \ldots, H-1\}$:*

$$\|x_{i,\ell} - \bar{x}_t\| \leq \eta H \|\nabla f(\bar{x}_t)\|. \tag{51}$$

*Proof.* Fix any client $i \in \mathcal{S}$. From (40), we have

$$
\begin{aligned}
\|x_{i,\ell+1} - \bar{x}_t\| &= \|x_{i,\ell} - \bar{x}_t - \eta\left(\nabla f_i(x_{i,\ell}) - \nabla f_i(\bar{x}_t)\right) - \eta\nabla f(\bar{x}_t)\| \\
&\leq \|x_{i,\ell} - \bar{x}_t - \eta\left(\nabla f_i(x_{i,\ell}) - \nabla f_i(\bar{x}_t)\right)\| + \eta\|\nabla f(\bar{x}_t)\| \\
&\leq \lambda\|x_{i,\ell} - \bar{x}_t\| + \eta\|\nabla f(\bar{x}_t)\|,
\end{aligned}
\tag{52}
$$

where $\lambda = \sqrt{1 - \eta\mu} < 1$. The last inequality follows from Lemma 6 with $x = x_{i,\ell}$ and $y = \bar{x}_t$. Rolling out the final inequality in (52) from $\ell = 0$ yields:

$$
\begin{aligned}
\|x_{i,\ell} - \bar{x}_t\| &\leq \lambda^\ell \|x_{i,0} - \bar{x}_t\| + \left(\sum_{j=0}^{\ell-1} \lambda^j\right)\eta\|\nabla f(\bar{x}_t)\| \\
&\overset{(a)}{=} \left(\sum_{j=0}^{\ell-1} \lambda^j\right)\eta\|\nabla f(\bar{x}_t)\| \\
&\overset{(b)}{\leq} \eta\ell\|\nabla f(\bar{x}_t)\| \\
&\overset{(c)}{\leq} \eta H\|\nabla f(\bar{x}_t)\|.
\end{aligned}
\tag{53}
$$

For (a), we used the fact that $x_{i,0} = \bar{x}_t, \forall i \in \mathcal{S}$; for (b), we used the fact that $\lambda < 1$[4]; and for (c), we note that $\ell \leq H - 1$. $\qquad\square$

---

[4]Note that actually, in step (b), we used the looser bound for the convex setting, namely $\lambda \leq 1$. It seems unlikely that the tighter bound $\lambda < 1$ will buy us anything other than a potential improvement upon the constant $\frac{1}{6}$ appearing in the exponent in equation (38).

We are now equipped with all the ingredients needed to complete the proof of Theorem 9.

**Completing the proof of Theorem 9**: Combining the bounds in Lemma's 5 and 7, we obtain

$$f(\bar{x}_{t+1}) - f(\bar{x}_t) \leq -\eta H (1 - \eta LH) \|\nabla f(\bar{x}_t)\|^2 + \left( \frac{\eta L}{m} \sum_{i=1}^{m} \sum_{\ell=0}^{H-1} \eta H \|\nabla f(\bar{x}_t)\| \right) \|\nabla f(\bar{x}_t)\|$$

$$+ \frac{\eta^2 L^3 H}{m} \sum_{i=1}^{m} \sum_{\ell=0}^{H-1} (\eta H \|\nabla f(\bar{x}_t)\|)^2$$

$$= -\eta H (1 - \eta LH) \|\nabla f(\bar{x}_t)\|^2 + \eta^2 LH^2 \|\nabla f(\bar{x}_t)\|^2 + \eta^4 L^3 H^4 \|\nabla f(\bar{x}_t)\|^2$$

$$\leq -\eta H (1 - 3\eta LH) \|\nabla f(\bar{x}_t)\|^2,$$

(54)

where in the last step, we used the fact that the step-size $\eta$ satisfies $\eta LH \leq 1$. From (54) and (34), we obtain

$$f(\bar{x}_{t+1}) - f(x^*) \leq (1 - 2\eta\mu H (1 - 3\eta LH)) (f(\bar{x}_t) - f(x^*)). \tag{55}$$

With $\eta = \frac{1}{6LH}$, the above inequality takes the form

$$f(\bar{x}_{t+1}) - f(x^*) \leq \left( 1 - \frac{1}{6\kappa} \right) (f(\bar{x}_t) - f(x^*)), \text{ where } \kappa = \frac{L}{\mu}. \tag{56}$$

Using the above inequality recursively leads to the claim of the theorem.

# E    Proof of Theorem 5: Lower bound for `FedLin`

To prove Theorem 5, we will construct an example involving two clients. Let us start by defining the objective functions of the clients as follows.

$$f_1(x) = \frac{1}{2} x' \underbrace{\begin{bmatrix} 1 & 0 \\ 0 & 1 \end{bmatrix}}_{\mathbf{A}_1} x + b'x; \quad f_2(x) = \frac{1}{2} x' \underbrace{\begin{bmatrix} L & 0 \\ 0 & 1 \end{bmatrix}}_{\mathbf{A}_2} x - b'x,$$

where $b \in \mathbb{R}^2$ is any arbitrary vector, and $L \geq 14$. Here, we have used the notation $x'$ to indicate the transpose of a column vector $x$. From inspection, it is clear that each of the client objective functions is 1-strongly convex. Also, $f_1(x)$ and $f_2(x)$ are both $L$-smooth.[5] The global loss function is then given by

$$f(x) = \frac{1}{2} (f_1(x) + f_2(x)) = \frac{1}{2} x' \begin{bmatrix} \frac{L+1}{2} & 0 \\ 0 & 1 \end{bmatrix} x.$$

It is easy to see that the minimum of $f(x)$ is $x^* = \begin{bmatrix} 0 & 0 \end{bmatrix}'$. Let us fix a communication round $t \in \{1, \dots, T\}$. Now given that $\delta_c = \delta_s = 1$, and $\eta_i = \eta, i \in \{1, 2\}$, the local update rule for each of the clients takes the following form:

$$x_{i,\ell+1} = x_{i,\ell} - \eta(\nabla f_i(x_{i,\ell}) - \nabla f_i(\bar{x}_t) + \nabla f(\bar{x}_t)).$$

Simple calculations then lead to the following recursions:

$$x_{1,\ell+1} = (\mathbf{I} - \eta\mathbf{A}_1) x_{1,\ell} - \eta \begin{bmatrix} \frac{L-1}{2} & 0 \\ 0 & 0 \end{bmatrix} \bar{x}_t,$$

$$x_{2,\ell+1} = (\mathbf{I} - \eta\mathbf{A}_2) x_{2,\ell} - \eta \begin{bmatrix} \frac{1-L}{2} & 0 \\ 0 & 0 \end{bmatrix} \bar{x}_t.$$

Given the diagonal structure of $\mathbf{A}_1$ and $\mathbf{A}_2$, we can easily roll out the above recursions. Accordingly, for client 1, we obtain

$$x_{1,H} = (\mathbf{I} - \eta\mathbf{A}_1)^H \bar{x}_t - \eta \left( \sum_{j=0}^{H-1} (\mathbf{I} - \eta\mathbf{A}_1)^j \right) \begin{bmatrix} \frac{L-1}{2} & 0 \\ 0 & 0 \end{bmatrix} \bar{x}_t$$

$$= \begin{bmatrix} (1-\eta)^H \left( \frac{L+1}{2} \right) - \frac{L-1}{2} & 0 \\ 0 & (1-\eta)^H \end{bmatrix} \bar{x}_t.$$

Similarly, for client 2, we have

$$x_{2,H} = (\mathbf{I} - \eta\mathbf{A}_2)^H \bar{x}_t - \eta \left( \sum_{j=0}^{H-1} (\mathbf{I} - \eta\mathbf{A}_2)^j \right) \begin{bmatrix} \frac{1-L}{2} & 0 \\ 0 & 0 \end{bmatrix} \bar{x}_t$$

$$= \begin{bmatrix} (1-\eta L)^H \left( \frac{L+1}{2L} \right) - \frac{1-L}{2L} & 0 \\ 0 & (1-\eta)^H \end{bmatrix} \bar{x}_t.$$

Thus,

$$\bar{x}_{t+1} = \frac{1}{2}(x_{1,H} + x_{2,H}) = \underbrace{\begin{bmatrix} \left( (1-\eta)^H + \frac{(1-\eta L)^H}{L} \right) \left( \frac{L+1}{4} \right) - \frac{(L-1)^2}{4L} & 0 \\ 0 & (1-\eta)^H \end{bmatrix}}_{\mathbf{M}} \bar{x}_t. \quad (57)$$

In the rest of the proof, we will argue that for $\bar{x}_t$ to converge to $x^*$ based on the above recursion, $\eta$ must be chosen inversely proportional to $H$; the lower bound will then naturally follow. Let us start

---

[5]Strictly speaking, $f_1(x)$ is 1-smooth, but since $L > 1$, it is also $L$-smooth.

by noting that $\bar{x}_{t+1} = \mathbf{M}\bar{x}_t$ can be viewed as a discrete-time linear time-invariant (LTI) system where $\mathbf{M}$ is the state transition matrix. Since $x^* = [0 \quad 0]'$, guaranteeing $\bar{x}_t$ converges to $x^*$ regardless of the initial condition $\bar{x}_1$ is equivalent to arguing that $\mathbf{M}$ is a Schur stable matrix, i.e., all the eigenvalues of $\mathbf{M}$ lie strictly inside the unit circle. It is easy to see that the eigenvalues $\lambda_1(\eta, H, L)$ and $\lambda_2(\eta, H)$ of $\mathbf{M}$ are

$$\lambda_1(\eta, H, L) = \left( (1-\eta)^H + \frac{(1-\eta L)^H}{L} \right) \left( \frac{L+1}{4} \right) - \frac{(L-1)^2}{4L}; \quad \lambda_2(\eta, H) = (1-\eta)^H.$$

In order for $\mathbf{M}$ to be Schur stable, $\lambda_1(\eta, H, L) > -1$ is a necessary condition. We will now show that to satisfy this necessary condition, $\eta$ must scale inversely with $H$. Observe that

$$\lambda_1(\eta, H, L) > -1 \implies \left( (1-\eta)^H + \frac{(1-\eta L)^H}{L} \right) \left( \frac{L+1}{4} \right) - \frac{(L-1)^2}{4L} > -1$$

$$\implies (1-\eta)^H \left( \frac{(L+1)^2}{4L} \right) - \frac{(L-1)^2}{4L} > -1$$

$$\implies (1-\eta)^H > \frac{L^2 - 6L + 1}{(L+1)^2} \tag{58}$$

$$\implies (1-\eta)^H > \frac{1}{2}$$

$$\implies \frac{1}{1+\eta H} > \frac{1}{2}$$

$$\implies \eta < \frac{1}{H}.$$

For the second implication, we used the fact that $L > 1$. For the fourth implication, we note that

$$g(z) = \frac{z^2 - 6z + 1}{(z+1)^2}$$

is a monotonically increasing function of its argument for all $z > 1$. The claim then follows by noting that $g(14) > \frac{1}{2}$, and $L \geq 14$. For the second-last implication, we used the fact that for any $z \in (0, 1)$, and any positive integer $r$, the following is true

$$(1-z)^r \leq \frac{1}{1+rz}.$$

We conclude that $\eta < \frac{1}{H}$ is a necessary condition for $\mathbf{M}$ to be Schur stable.[6] Now let us look at the implication of this necessary condition on the eigenvalue $\lambda_2(\eta, H) = (1-\eta)^H$. Since $\lambda_2(\eta, H)$ is monotonically decreasing in $\eta$, the following holds for any $\eta$ that stabilizes $\mathbf{M}$:[7]

$$\lambda_2(\eta, H) > \lambda_2(\eta_c, H) = \left( 1 - \frac{1}{H} \right)^H \geq \left( 1 - \frac{1}{2} \right)^2 > \exp(-2), \forall H \geq 2,$$

where $\eta_c = \frac{1}{H}$. Here, we have used the fact that $\left( 1 - \frac{1}{z} \right)^z$ is a monotonically increasing function in $z$ for $z > 1$, and that $H \geq 2$. Now suppose FedLin is initialized with $\bar{x}_1 = [0 \quad \beta]'$, where $\beta > 0$. Based on (57), for any $T \geq 1$, we then have

$$\bar{x}_{T+1} = \begin{bmatrix} 0 \\ \lambda_2^T \beta \end{bmatrix} \geq \exp(-2T) \begin{bmatrix} 0 \\ \beta \end{bmatrix} = \exp(-2T)\bar{x}_1 \implies \|\bar{x}_{T+1} - x^*\|^2 \geq \exp(-4T)\|\bar{x}_1 - x^*\|^2.$$

Moreover, substituting the value of $\bar{x}_{T+1}$ above in the expression for $f(x)$, we obtain

$$f(\bar{x}_{T+1}) = \frac{1}{2}\lambda_2^{2T}\beta^2 = \lambda_2^{2T} f(\bar{x}_1) \geq \exp(-4T)f(\bar{x}_1) \tag{59}$$

$$\implies f(\bar{x}_{T+1}) - f(x^*) \geq \exp(-4T)(f(\bar{x}_1) - f(x^*)), \tag{60}$$

---

[6]Notice that when $H = 1$, this necessary condition translates to the trivial condition $\eta < 1$.

[7]That a stabilizing $\eta$ exists follows from Theorem 9. In particular, $\eta = \frac{1}{6LH}$ will render $\mathbf{M}$ Schur stable. However, our main goal is to show that even if there do exist values of $\eta$ larger than $\frac{1}{6LH}$ that guarantee convergence of $\bar{x}_t$ to $x^*$, such step-size values must obey $\eta < \frac{1}{H}$, which in turn leads to the $H$-independent lower bound in equation (10).

since $f(x^*) = 0$. We have thus established the desired lower bounds. Finally, note that all our arguments above hold for any $L \geq 14$, any $H \geq 2$, and any $T \geq 1$. This concludes the proof.

**Remark 2.** *Note that the quantity $b$ in the objective functions of the clients does not feature anywhere in the above analysis. As such, one possible choice of $b$ is simply $b = \begin{bmatrix} 0 & 0 \end{bmatrix}'$. For this choice of $b$, it is clear from inspection that both $f_1(x)$ and $f_2(x)$ have the same minimum, namely $x^* = \begin{bmatrix} 0 & 0 \end{bmatrix}$. Our analysis thus reveals that even when the client local objectives share the same minimum, the lower bound in equation* (10) *continues to hold.*

**Remark 3.** *Let us look more closely at the role played by $L$ in the above example. From inspection, larger the value of $L$, the less smooth the overall function $f(x)$, and more the objective heterogeneity. Specifically, increasing $L$ increases the heterogeneity between the client local objectives by increasing $\|\nabla f_1(x) - \nabla f_2(x)\|$. In terms of the impact of $L$ on the dynamics of $\bar{x}_t$, we note that $T_2$ in the expression for $\lambda_1(\eta, H, L)$ below becomes larger as we increase $L$.*

$$\lambda_1(\eta, H, L) = \underbrace{\left( (1 - \eta)^H + \frac{(1 - \eta L)^H}{L} \right) \left( \frac{L+1}{4} \right)}_{T_1} - \underbrace{\frac{(L-1)^2}{4L}}_{T_2}$$

*To keep $\lambda_1(\eta, H, L) > -1$, and thereby ensure stability of the recursion* (57), *we then need the coefficient of $(L + 1)/4$ in $T_1$ above to be adequately large, despite potentially large values of $H$: this is precisely what necessitates $\eta$ to scale inversely with $H$. To summarize, in the above example, large $L$ leads to less smoothness, more objective heterogeneity, and small step-size.*

# F   Proofs pertaining to Systems Heterogeneity in Section 4

As in Appendix D, we will focus on a fixed communication round $t \in \{1, \ldots, T\}$; all our subsequent arguments will apply identically to each such round. Given that $\delta_c = \delta_s = 1$, the local update rule that will be of relevance to us for the majority of this section is

$$x_{i,\ell+1} = x_{i,\ell} - \eta_i(\nabla f_i(x_{i,\ell}) - \nabla f_i(\bar{x}_t) + \nabla f(\bar{x}_t)), \tag{61}$$

where we have dropped the superscript of $t$ on the local iterates $x_{i,\ell}$ to simplify notation. Based on the discussion in Section 4, recall that the client step-sizes are chosen as follows:

$$\eta_i = \frac{\bar{\eta}}{\tau_i}, \forall i \in \mathcal{S},$$

where $\bar{\eta} \in (0, 1)$ is a flexible design parameter that we will specify based on the context.

## F.1   Proof of Theorem 1: Analysis for Strongly Convex loss functions

The proof of Theorem 1 largely mirrors that of Theorem 9. We start with the following analogue of Lemma 5.

**Lemma 8.** *Suppose each $f_i(x)$ is L-smooth. Moreover, suppose $\tau_i \geq 1, \forall i \in \mathcal{S}, \delta_c = \delta_s = 1$, and $\eta_i = \frac{\bar{\eta}}{\tau_i}, \forall i \in \mathcal{S}$, where $\bar{\eta} \in (0, 1)$. Then, `FedLin` guarantees:*

$$
f(\bar{x}_{t+1}) - f(\bar{x}_t) \leq -\bar{\eta}(1 - \bar{\eta}L)\|\nabla f(\bar{x}_t)\|^2 + \left( \frac{L}{m} \sum_{i=1}^{m} \eta_i \sum_{\ell=0}^{\tau_i - 1} \|x_{i,\ell} - \bar{x}_t\| \right) \|\nabla f(\bar{x}_t)\|
$$
$$
+ \frac{\bar{\eta}L^3}{m} \sum_{i=1}^{m} \eta_i \sum_{\ell=0}^{\tau_i - 1} \|x_{i,\ell} - \bar{x}_t\|^2. \tag{62}
$$

*Proof.* Using (61) and the fact that $x_{i,0} = \bar{x}_t, \forall i \in \mathcal{S}$, we have:

$$
x_{i,\tau_i} = \bar{x}_t - \eta_i \sum_{\ell=0}^{\tau_i - 1} \nabla f_i(x_{i,\ell}) - \eta_i \tau_i (\nabla f(\bar{x}_t) - \nabla f_i(\bar{x}_t))
$$
$$
= \bar{x}_t - \eta_i \sum_{\ell=0}^{\tau_i - 1} \nabla f_i(x_{i,\ell}) - \bar{\eta}(\nabla f(\bar{x}_t) - \nabla f_i(\bar{x}_t)), \forall i \in \mathcal{S}, \tag{63}
$$

where we used $\eta_i \tau_i = \bar{\eta}$ in the second step. Averaging the above iterates across clients, we obtain:

$$
\bar{x}_{t+1} = \frac{1}{m} \sum_{i=1}^{m} x_{i,\tau_i} = \bar{x}_t - \frac{1}{m} \sum_{i=1}^{m} \eta_i \sum_{\ell=0}^{\tau_i - 1} \nabla f_i(x_{i,\ell}) - \frac{\bar{\eta}}{m} \sum_{i=1}^{m} (\nabla f(\bar{x}_t) - \nabla f_i(\bar{x}_t))
$$
$$
= \bar{x}_t - \frac{1}{m} \sum_{i=1}^{m} \eta_i \sum_{\ell=0}^{\tau_i - 1} \nabla f_i(x_{i,\ell}), \tag{64}
$$

where for the second equality, we used the fact that $\nabla f(y) = \frac{1}{m} \sum_{i=1}^{m} \nabla f_i(y), \forall y \in \mathbb{R}^d$. Based on the $L$-smoothness of $f(\cdot)$ and (64), we then have

$$
f(\bar{x}_{t+1}) - f(\bar{x}_t) \leq \langle \bar{x}_{t+1} - \bar{x}_t, \nabla f(\bar{x}_t) \rangle + \frac{L}{2} \|\bar{x}_{t+1} - \bar{x}_t\|^2
$$
$$
= -\left\langle \frac{1}{m} \sum_{i=1}^{m} \eta_i \sum_{\ell=0}^{\tau_i - 1} \nabla f_i(x_{i,\ell}), \nabla f(\bar{x}_t) \right\rangle + \frac{L}{2} \left\| \frac{1}{m} \sum_{i=1}^{m} \eta_i \sum_{\ell=0}^{\tau_i - 1} \nabla f_i(x_{i,\ell}) \right\|^2. \tag{65}
$$

Just as in the proof of Theorem 9, we now proceed to separately bound each of the two terms above, starting with the first term, as follows.

$$
\begin{aligned}
-\left\langle \frac{1}{m} \sum_{i=1}^{m} \eta_i \sum_{\ell=0}^{\tau_i-1} \nabla f_i(x_{i,\ell}), \nabla f(\bar{x}_t) \right\rangle &= -\left\langle \frac{1}{m} \sum_{i=1}^{m} \eta_i \sum_{\ell=0}^{\tau_i-1} (\nabla f_i(x_{i,\ell}) - \nabla f_i(\bar{x}_t)) \right. \\
&\quad \left. + \frac{1}{m} \sum_{i=1}^{m} \eta_i \sum_{\ell=0}^{\tau_i-1} \nabla f_i(\bar{x}_t), \nabla f(\bar{x}_t) \right\rangle \\
&= -\left\langle \frac{1}{m} \sum_{i=1}^{m} \eta_i \sum_{\ell=0}^{\tau_i-1} (\nabla f_i(x_{i,\ell}) - \nabla f_i(\bar{x}_t)) \right. \\
&\quad \left. + \frac{1}{m} \sum_{i=1}^{m} \underbrace{\eta_i \tau_i}_{\bar{\eta}} \nabla f_i(\bar{x}_t), \nabla f(\bar{x}_t) \right\rangle \\
&= -\left\langle \frac{1}{m} \sum_{i=1}^{m} \eta_i \sum_{\ell=0}^{\tau_i-1} (\nabla f_i(x_{i,\ell}) - \nabla f_i(\bar{x}_t)), \nabla f(\bar{x}_t) \right\rangle \\
&\quad - \bar{\eta} \|\nabla f(\bar{x}_t)\|^2 \\
&\overset{(a)}{\leq} \frac{1}{m} \left( \left\| \sum_{i=1}^{m} \eta_i \sum_{\ell=0}^{\tau_i-1} (\nabla f_i(x_{i,\ell}) - \nabla f_i(\bar{x}_t)) \right\| \right) \|\nabla f(\bar{x}_t)\| \\
&\quad - \bar{\eta} \|\nabla f(\bar{x}_t)\|^2 \\
&\overset{(b)}{\leq} \frac{1}{m} \left( \sum_{i=1}^{m} \eta_i \sum_{\ell=0}^{\tau_i-1} \|\nabla f_i(x_{i,\ell}) - \nabla f_i(\bar{x}_t)\| \right) \|\nabla f(\bar{x}_t)\| \\
&\quad - \bar{\eta} \|\nabla f(\bar{x}_t)\|^2 \\
&\overset{(c)}{\leq} \frac{L}{m} \left( \sum_{i=1}^{m} \eta_i \sum_{\ell=0}^{\tau_i-1} \|x_{i,\ell} - \bar{x}_t\| \right) \|\nabla f(\bar{x}_t)\| - \bar{\eta} \|\nabla f(\bar{x}_t)\|^2,
\end{aligned}
$$
(66)

where (a) follows from the Cauchy-Schwartz inequality, (b) follows from the triangle inequality, and (c) follows from the $L$-smoothness of each $f_i(\cdot)$. For the second term in (65), observe that

$$
\begin{aligned}
\frac{L}{2} \left\| \frac{1}{m} \sum_{i=1}^{m} \eta_i \sum_{\ell=0}^{\tau_i-1} \nabla f_i(x_{i,\ell}) \right\|^2 &= \frac{L}{2} \left\| \frac{1}{m} \sum_{i=1}^{m} \eta_i \sum_{\ell=0}^{\tau_i-1} (\nabla f_i(x_{i,\ell}) - \nabla f_i(\bar{x}_t)) + \frac{1}{m} \sum_{i=1}^{m} \eta_i \sum_{\ell=0}^{\tau_i-1} \nabla f_i(\bar{x}_t) \right\|^2 \\
&= \frac{L}{2} \left\| \frac{1}{m} \sum_{i=1}^{m} \eta_i \sum_{\ell=0}^{\tau_i-1} (\nabla f_i(x_{i,\ell}) - \nabla f_i(\bar{x}_t)) + \bar{\eta} \nabla f(\bar{x}_t) \right\|^2 \\
&\overset{(a)}{\leq} L \left\| \frac{1}{m} \sum_{i=1}^{m} \eta_i \sum_{\ell=0}^{\tau_i-1} (\nabla f_i(x_{i,\ell}) - \nabla f_i(\bar{x}_t)) \right\|^2 + \bar{\eta}^2 L \|\nabla f(\bar{x}_t)\|^2 \\
&\overset{(b)}{\leq} \frac{L}{m} \sum_{i=1}^{m} \eta_i{}^2 \left\| \sum_{\ell=0}^{\tau_i-1} (\nabla f_i(x_{i,\ell}) - \nabla f_i(\bar{x}_t)) \right\|^2 + \bar{\eta}^2 L \|\nabla f(\bar{x}_t)\|^2 \\
&\overset{(c)}{\leq} \frac{L}{m} \sum_{i=1}^{m} \eta_i{}^2 \tau_i \sum_{\ell=0}^{\tau_i-1} \|\nabla f_i(x_{i,\ell}) - \nabla f_i(\bar{x}_t)\|^2 + \bar{\eta}^2 L \|\nabla f(\bar{x}_t)\|^2 \\
&\overset{(d)}{\leq} \frac{\bar{\eta} L^3}{m} \sum_{i=1}^{m} \eta_i \sum_{\ell=0}^{\tau_i-1} \|x_{i,\ell} - \bar{x}_t\|^2 + \bar{\eta}^2 L \|\nabla f(\bar{x}_t)\|^2.
\end{aligned}
$$
(67)

For (a), we used (35) with $\gamma = 1$; for (b) and (c), we used Jensen's inequality (36); and for (d), we used $\eta_i \tau_i = \bar{\eta}$, and the $L$-smoothness of $f_i(\cdot)$. Plugging in the bounds (66) and (67) in (65), and simplifying, we obtain (62). $\qquad\square$

**Remark 4.** *Note that the proof of Lemma 8 made no use of convexity. Thus, the same analysis will carry over when we will later study the non-convex setting.*

For both the strongly convex and convex settings, the following lemma provides a bound on the drift at client $i$.

**Lemma 9.** *Suppose each $f_i(x)$ is L-smooth and convex. Moreover, suppose $\tau_i \geq 1, \forall i \in \mathcal{S}$, $\delta_c = \delta_s = 1$, and $\eta_i \leq \frac{1}{L}, \forall i \in \mathcal{S}$. Then, FedLin guarantees the following bound for each $i \in \mathcal{S}$, and $\forall \ell \in \{0, \ldots, \tau_i - 1\}$:*

$$\|x_{i,\ell} - \bar{x}_t\| \leq \eta_i \tau_i \|\nabla f(\bar{x}_t)\|. \tag{68}$$

*Proof.* Based on (61), for any client $i \in \mathcal{S}$, we have

$$\begin{aligned}
\|x_{i,\ell+1} - \bar{x}_t\| &= \|x_{i,\ell} - \bar{x}_t - \eta_i \left(\nabla f_i(x_{i,\ell}) - \nabla f_i(\bar{x}_t)\right) - \eta_i \nabla f(\bar{x}_t)\| \\
&\leq \|x_{i,\ell} - \bar{x}_t - \eta_i \left(\nabla f_i(x_{i,\ell}) - \nabla f_i(\bar{x}_t)\right)\| + \eta_i \|\nabla f(\bar{x}_t)\| \\
&\leq \|x_{i,\ell} - \bar{x}_t\| + \eta_i \|\nabla f(\bar{x}_t)\|,
\end{aligned} \tag{69}$$

where in the last step, we used the bound (47) from Lemma 6 that applies to both the convex and strongly convex settings. From (69), and the fact that $x_{i,0} = \bar{x}_t, \forall i \in \mathcal{S}$, we immediately obtain

$$\begin{aligned}
\|x_{i,\ell} - \bar{x}_t\| &\leq \|x_{i,0} - \bar{x}_t\| + \eta_i \ell \|\nabla f(\bar{x}_t)\| \\
&\leq \eta_i \tau_i \|\nabla f(\bar{x}_t)\|,
\end{aligned} \tag{70}$$

which is the desired conclusion. $\qquad\square$

**Completing the proof of Theorem 1:** To complete the proof of Theorem 1, let us substitute the bound on the drift term from Lemma 9 in equation (62). This yields:

$$\begin{aligned}
f(\bar{x}_{t+1}) - f(\bar{x}_t) &\leq -\bar{\eta}\left(1 - \bar{\eta}L\right)\|\nabla f(\bar{x}_t)\|^2 + \left(\frac{L}{m}\sum_{i=1}^{m}\eta_i\sum_{\ell=0}^{\tau_i-1}\eta_i\tau_i\|\nabla f(\bar{x}_t)\|\right)\|\nabla f(\bar{x}_t)\| \\
&\quad + \frac{\bar{\eta}L^3}{m}\sum_{i=1}^{m}\eta_i\sum_{\ell=0}^{\tau_i-1}\left(\eta_i\tau_i\|\nabla f(\bar{x}_t)\|\right)^2 \\
&= -\bar{\eta}\left(1 - \bar{\eta}L\right)\|\nabla f(\bar{x}_t)\|^2 + \frac{L}{m}\sum_{i=1}^{m}(\eta_i\tau_i)^2\|\nabla f(\bar{x}_t)\|^2 \\
&\quad + \frac{\bar{\eta}L^3}{m}\sum_{i=1}^{m}(\eta_i\tau_i)^3\|\nabla f(\bar{x}_t)\|^2 \\
&= -\bar{\eta}\left(1 - \bar{\eta}L\right)\|\nabla f(\bar{x}_t)\|^2 + \bar{\eta}^2 L\|\nabla f(\bar{x}_t)\|^2 + \bar{\eta}^4 L^3\|\nabla f(\bar{x}_t)\|^2 \\
&\leq -\bar{\eta}\left(1 - 3\bar{\eta}L\right)\|\nabla f(\bar{x}_t)\|^2,
\end{aligned} \tag{71}$$

where in the third step, we used $\eta_i\tau_i = \bar{\eta}$, and for the last inequality, we set the flexible parameter $\bar{\eta}$ to satisfy $\bar{\eta}L \leq 1$. In particular, setting $\bar{\eta} = \frac{1}{6L}$, and using the fact that $f(\cdot)$ is $\mu$-strongly convex, we immediately obtain

$$f(\bar{x}_{t+1}) - f(x^*) \leq \left(1 - \frac{1}{6\kappa}\right)(f(\bar{x}_t) - f(x^*)), \text{ where } \kappa = \frac{L}{\mu}. \tag{72}$$

Finally, note that $\bar{\eta} = \frac{1}{6L}$ implies that the step-size for client $i$ is $\eta_i = \frac{1}{6L\tau_i}$, which satisfies the requirement $\eta_i L \leq 1$ for Lemma 9 to hold, and is precisely the choice of step-size in the statement of the theorem. This completes the proof of Theorem 1.

### F.2   Proof of Theorem 2: Analysis for Convex loss functions

In order to prove Theorem 2, we will use a slightly different approach than that in Theorem 1. Instead of focusing on the quantity $f(\bar{x}_{t+1}) - f(\bar{x}_t)$, we will be interested in bounding the distance to the optimal point $x^*$ at the end of the $t$-th communication round, namely $\|\bar{x}_{t+1} - x^*\|^2$. The following lemma is the starting point of our analysis.

**Lemma 10.** *Suppose each $f_i(x)$ is L-smooth and convex. Moreover, suppose $\tau_i \geq 1, \forall i \in \mathcal{S}$, $\delta_c = \delta_s = 1$, and $\eta_i = \frac{\bar{\eta}}{\tau_i}, \forall i \in \mathcal{S}$, where $\bar{\eta} \in (0,1)$. Then, FedLin guarantees:*

$$\|\bar{x}_{t+1} - x^*\|^2 \leq \|\bar{x}_t - x^*\|^2 - 2\bar{\eta}\left(f(\bar{x}_t) - f(x^*)\right) + \frac{L(1+2\bar{\eta}L)}{m}\sum_{i=1}^{m}\eta_i\sum_{\ell=0}^{\tau_i-1}\|x_{i,\ell} - \bar{x}_t\|^2 \tag{73}$$
$$+ 2\bar{\eta}^2\|\nabla f(\bar{x}_t)\|^2.$$

*Proof.* From (64), recall that

$$\bar{x}_{t+1} = \bar{x}_t - \frac{1}{m}\sum_{i=1}^{m}\eta_i\sum_{\ell=0}^{\tau_i-1}\nabla f_i(x_{i,\ell}). \tag{74}$$

Thus, we have

$$\|\bar{x}_{t+1} - x^*\|^2 = \|\bar{x}_t - x^*\|^2 - \frac{2}{m}\langle\bar{x}_t - x^*, \sum_{i=1}^{m}\eta_i\sum_{\ell=0}^{\tau_i-1}\nabla f_i(x_{i,\ell})\rangle + \left\|\frac{1}{m}\sum_{i=1}^{m}\eta_i\sum_{\ell=0}^{\tau_i-1}\nabla f_i(x_{i,\ell})\right\|^2. \tag{75}$$

To proceed, let us first bound the cross-term as follows.

$$-\frac{2}{m}\langle\bar{x}_t - x^*, \sum_{i=1}^{m}\eta_i\sum_{\ell=0}^{\tau_i-1}\nabla f_i(x_{i,\ell})\rangle = -\frac{2}{m}\sum_{i=1}^{m}\eta_i\sum_{\ell=0}^{\tau_i-1}\bigg(\langle\bar{x}_t - x_{i,\ell}, \nabla f_i(x_{i,\ell})\rangle$$

$$+ \langle x_{i,\ell} - x^*, \nabla f_i(x_{i,\ell})\rangle\bigg)$$

$$\overset{(a)}{\leq} -\frac{2}{m}\sum_{i=1}^{m}\eta_i\sum_{\ell=0}^{\tau_i-1}\bigg(f_i(\bar{x}_t) - f_i(x_{i,\ell}) - \frac{L}{2}\|x_{i,\ell} - \bar{x}_t\|^2 +$$

$$\langle x_{i,\ell} - x^*, \nabla f_i(x_{i,\ell})\rangle\bigg)$$

$$\overset{(b)}{\leq} -\frac{2}{m}\sum_{i=1}^{m}\eta_i\sum_{\ell=0}^{\tau_i-1}\bigg(f_i(\bar{x}_t) - f_i(x_{i,\ell}) + f_i(x_{i,\ell}) - f_i(x^*)\bigg)$$

$$+ \frac{L}{m}\sum_{i=1}^{m}\eta_i\sum_{\ell=0}^{\tau_i-1}\|x_{i,\ell} - \bar{x}_t\|^2$$

$$= -\frac{2}{m}\sum_{i=1}^{m}\eta_i\tau_i\left(f_i(\bar{x}_t) - f_i(x^*)\right) + \frac{L}{m}\sum_{i=1}^{m}\eta_i\sum_{\ell=0}^{\tau_i-1}\|x_{i,\ell} - \bar{x}_t\|^2$$

$$= -2\bar{\eta}\left(f(\bar{x}_t) - f(x^*)\right) + \frac{L}{m}\sum_{i=1}^{m}\eta_i\sum_{\ell=0}^{\tau_i-1}\|x_{i,\ell} - \bar{x}_t\|^2. \tag{76}$$

In the above steps, for (a) we used the fact that $f_i(\cdot)$ is L-smooth; see equation (29). For (b), we used the fact that $\langle x_{i,\ell} - x^*, \nabla f_i(x_{i,\ell})\rangle \geq f_i(x_{i,\ell}) - f_i(x^*)$ - a consequence of the convexity of $f_i(\cdot)$. Finally, to arrive at the last step, we used $\eta_i\tau_i = \bar{\eta}$.

Now observe that while deriving the bound in Eq. (67) of Lemma 8, we only used the fact that each $f_i(\cdot)$ is L-smooth. Thus, the same bound applies in our current setting. In particular, we have

$$\left\|\frac{1}{m}\sum_{i=1}^{m}\eta_i\sum_{\ell=0}^{\tau_i-1}\nabla f_i(x_{i,\ell})\right\|^2 \leq \frac{2\bar{\eta}L^2}{m}\sum_{i=1}^{m}\eta_i\sum_{\ell=0}^{\tau_i-1}\|x_{i,\ell} - \bar{x}_t\|^2 + 2\bar{\eta}^2\|\nabla f(\bar{x}_t)\|^2. \tag{77}$$

Combining the above bound with that in (76) immediately leads to (73). $\square$

**Completing the proof of Theorem 2:** Suppose $\eta_i L \leq 1$. Then, based on Lemma 9, recall that

$$\|x_{i,\ell} - \bar{x}_t\| \leq \eta_i\tau_i\|\nabla f(\bar{x}_t)\|, \forall i \in \mathcal{S}, \forall \ell \in \{0, \ldots \tau_i - 1\}. \tag{78}$$

Substituting the above bound on the drift term in (73) yields:

$$\|\bar{x}_{t+1} - x^*\|^2 \leq \|\bar{x}_t - x^*\|^2 - 2\bar{\eta}\left(f(\bar{x}_t) - f(x^*)\right) + \frac{L(1 + 2\bar{\eta}L)}{m}\sum_{i=1}^{m}\eta_i\sum_{\ell=0}^{\tau_i-1}(\eta_i\tau_i)^2\|\nabla f(\bar{x}_t)\|^2$$

$$+ 2\bar{\eta}^2\|\nabla f(\bar{x}_t)\|^2$$

$$\stackrel{(a)}{\leq} \|\bar{x}_t - x^*\|^2 - 2\bar{\eta}\left(f(\bar{x}_t) - f(x^*)\right) + \frac{3L}{m}\sum_{i=1}^{m}(\eta_i\tau_i)^3\|\nabla f(\bar{x}_t)\|^2 + 2\bar{\eta}^2\|\nabla f(\bar{x}_t)\|^2$$

$$\stackrel{(b)}{=} \|\bar{x}_t - x^*\|^2 - 2\bar{\eta}\left(f(\bar{x}_t) - f(x^*)\right) + 3\bar{\eta}^3 L\|\nabla f(\bar{x}_t)\|^2 + 2\bar{\eta}^2\|\nabla f(\bar{x}_t)\|^2$$

$$\stackrel{(c)}{\leq} \|\bar{x}_t - x^*\|^2 - 2\bar{\eta}\left(f(\bar{x}_t) - f(x^*)\right) + 5\bar{\eta}^2\|\nabla f(\bar{x}_t)\|^2$$

$$\stackrel{(d)}{\leq} \|\bar{x}_t - x^*\|^2 - 2\bar{\eta}\left(1 - 5\bar{\eta}L\right)\left(f(\bar{x}_t) - f(x^*)\right).$$

(79)

In the above steps, for (a), we set $\bar{\eta}$ to satisfy $\bar{\eta}L \leq 1$; for (b), we used $\eta_i\tau_i = \bar{\eta}$; for (c), we once again used $\bar{\eta}L \leq 1$; and finally, for (d), we used the fact that $f(\cdot)$ is $L$-smooth; refer to equation (30). Now rearranging terms in (79), we obtain

$$2\bar{\eta}\left(1 - 5\bar{\eta}L\right)\left(f(\bar{x}_t) - f(x^*)\right) \leq \|\bar{x}_t - x^*\|^2 - \|\bar{x}_{t+1} - x^*\|^2.$$

(80)

Now let $\bar{\eta} = \frac{1}{10L}$, implying $\eta_i = \frac{1}{10L\tau_i}, \forall i \in \mathcal{S}$. Clearly, the conditions $\eta_i L \leq 1$ and $\bar{\eta}L \leq 1$ are then satisfied, and we have

$$f(\bar{x}_t) - f(x^*) \leq 10L\left(\|\bar{x}_t - x^*\|^2 - \|\bar{x}_{t+1} - x^*\|^2\right).$$

(81)

Summing the above inequality from $t = 1$ to $t = T$ leads to a telescoping sum on the R.H.S., and we obtain

$$f\left(\frac{1}{T}\sum_{t=1}^{T}\bar{x}_t\right) - f(x^*) \stackrel{(a)}{\leq} \frac{1}{T}\sum_{t=1}^{T}\left(f(\bar{x}_t) - f(x^*)\right) \leq \frac{10L}{T}\left(\|\bar{x}_1 - x^*\|^2 - \|\bar{x}_{T+1} - x^*\|^2\right),$$

(82)

where (a) follows from the convexity of $f(\cdot)$. This completes the proof of Theorem 2.

### F.3 Proof of Theorem 3: Analysis for Non-convex loss functions

To analyze the non-convex setting, the first key observation that we make is that the claim in Lemma 8 relies only on smoothness of the client loss functions $f_i(\cdot)$, and requires no assumptions of convexity. Thus, the bound in (62) applies to the non-convex setting as well. However, the bound on the drift term $\|x_{i,\ell} - \bar{x}_t\|$ that we derived in Lemma 9 did make use of convexity of each $f_i(\cdot)$, and hence, is no longer applicable. We thus need a way to bound $\|x_{i,\ell} - \bar{x}_t\|$ without making any assumptions of convexity; this is precisely the subject of the following lemma.

**Lemma 11.** *Suppose each $f_i(x)$ is $L$-smooth. Moreover, suppose $\tau_i \geq 1, \forall i \in \mathcal{S}$, $\delta_c = \delta_s = 1$, and $\eta_i \leq \frac{1}{L\tau_i}, \forall i \in \mathcal{S}$. Then, $\texttt{FedLin}$ guarantees the following bound for each $i \in \mathcal{S}$, and $\forall \ell \in \{0, \ldots, \tau_i - 1\}$:*

$$\|x_{i,\ell} - \bar{x}_t\| \leq 3\eta_i\tau_i\|\nabla f(\bar{x}_t)\|.$$

(83)

*Proof.* From (61), we have

$$\begin{aligned}\|x_{i,\ell+1} - \bar{x}_t\| &= \|x_{i,\ell} - \bar{x}_t - \eta_i\left(\nabla f_i(x_{i,\ell}) - \nabla f_i(\bar{x}_t)\right) - \eta_i\nabla f(\bar{x}_t)\| \\ &\leq \|x_{i,\ell} - \bar{x}_t\| + \eta_i\|\nabla f_i(x_{i,\ell}) - \nabla f_i(\bar{x}_t)\| + \eta_i\|\nabla f(\bar{x}_t)\| \\ &\stackrel{(a)}{\leq} (1 + \eta_i L)\|x_{i,\ell} - \bar{x}_t\| + \eta_i\|\nabla f(\bar{x}_t)\| \\ &\stackrel{(b)}{\leq} \left(1 + \frac{1}{\tau_i}\right)\|x_{i,\ell} - \bar{x}_t\| + \eta_i\|\nabla f(\bar{x}_t)\|,\end{aligned}$$

(84)

where (a) follows from the $L$-smoothness of $f_i(\cdot)$, and (b) follows by noting $\eta_i \leq \frac{1}{L\tau_i}$. Let $\alpha_i = 1 + \frac{1}{\tau_i}$.
We then have

$$
\begin{aligned}
\|x_{i,\ell} - \bar{x}_t\| &\leq \alpha_i^\ell \underbrace{\|x_{i,0} - \bar{x}_t\|}_{=0} + \left(\sum_{j=0}^{\ell-1} \alpha_i^j\right) \eta_i \|\nabla f(\bar{x}_t)\| \\
&= \left(\sum_{j=0}^{\ell-1} \alpha_i^j\right) \eta_i \|\nabla f(\bar{x}_t)\| \\
&= \left(\frac{\alpha_i^\ell - 1}{\alpha_i - 1}\right) \eta_i \|\nabla f(\bar{x}_t)\| \\
&\leq \eta_i \tau_i \left(1 + \frac{1}{\tau_i}\right)^\ell \|\nabla f(\bar{x}_t)\| \\
&\leq \eta_i \tau_i \left(1 + \frac{1}{\tau_i}\right)^{\tau_i} \|\nabla f(\bar{x}_t)\| \\
&\leq \exp(1) \eta_i \tau_i \|\nabla f(\bar{x}_t)\| \leq 3 \eta_i \tau_i \|\nabla f(\bar{x}_t)\|,
\end{aligned}
\tag{85}
$$

which is the desired conclusion. $\qquad\square$

**Remark 5.** *In comparison with Lemma 9 where we derived bounds on the drift for the strongly convex and convex settings, the requirement on the step-size for bounding the drift in Lemma 11 is more stringent: whereas $\eta_i \leq \frac{1}{L}$ sufficed for Lemma 9 to hold, we need $\eta_i \leq \frac{1}{L\tau_i}$ for the non-convex setting in Lemma 11. Also, the bound in Lemma 11 is worse than that in Lemma 9 by a factor of 3.*

**Completing the proof of Theorem 3:** By substituting the bound on the drift from Lemma 11 in equation (62), we obtain

$$
\begin{aligned}
f(\bar{x}_{t+1}) - f(\bar{x}_t) &\leq -\bar{\eta}(1 - \bar{\eta}L) \|\nabla f(\bar{x}_t)\|^2 + \left(\frac{L}{m} \sum_{i=1}^m \eta_i \sum_{\ell=0}^{\tau_i-1} 3\eta_i \tau_i \|\nabla f(\bar{x}_t)\|\right) \|\nabla f(\bar{x}_t)\| \\
&\quad + \frac{\bar{\eta}L^3}{m} \sum_{i=1}^m \eta_i \sum_{\ell=0}^{\tau_i-1} (3\eta_i \tau_i \|\nabla f(\bar{x}_t)\|)^2 \\
&= -\bar{\eta}(1 - \bar{\eta}L) \|\nabla f(\bar{x}_t)\|^2 + \frac{3L}{m} \sum_{i=1}^m (\eta_i \tau_i)^2 \|\nabla f(\bar{x}_t)\|^2 \\
&\quad + \frac{9\bar{\eta}L^3}{m} \sum_{i=1}^m (\eta_i \tau_i)^3 \|\nabla f(\bar{x}_t)\|^2 \\
&= -\bar{\eta}(1 - \bar{\eta}L) \|\nabla f(\bar{x}_t)\|^2 + 3\bar{\eta}^2 L \|\nabla f(\bar{x}_t)\|^2 + 9\bar{\eta}^4 L^3 \|\nabla f(\bar{x}_t)\|^2 \\
&\leq -\bar{\eta}(1 - 13\bar{\eta}L) \|\nabla f(\bar{x}_t)\|^2,
\end{aligned}
\tag{86}
$$

where we used $\eta_i \tau_i = \bar{\eta}$ and $\bar{\eta}L \leq 1$ in the above steps. Now suppose $\bar{\eta} = \frac{1}{26L}$, implying $\eta_i = \frac{1}{26L\tau_i} < \frac{1}{L\tau_i}$. Observe that this choice of $\bar{\eta}$ fulfils the step-size requirements for Lemma 11 to hold. Plugging $\bar{\eta} = \frac{1}{26L}$ in (86) yields

$$
\|\nabla f(\bar{x}_t)\|^2 \leq 52L\big(f(\bar{x}_t) - f(\bar{x}_{t+1})\big).
\tag{87}
$$

Summing the above inequality from $t = 1$ to $t = T$, we obtain as desired

$$
\min_{t \in [T]} \|\nabla f(\bar{x}_t)\|^2 \leq \frac{1}{T} \sum_{t=1}^T \|\nabla f(\bar{x}_t)\|^2 \leq \frac{52L}{T}\big(f(\bar{x}_1) - f(\bar{x}_{T+1})\big).
\tag{88}
$$

### F.4  Proof of Theorem 4: Analysis for Strongly Convex loss functions with Noise

In this section, we focus on analyzing the performance of FedLin under a general stochastic oracle model, subject to arbitrary objective and systems heterogeneity. For each $i \in \mathcal{S}$ and $x \in \mathbb{R}^d$, let $q_i(x)$ be an unbiased estimate of the gradient $\nabla f_i(x)$ with variance bounded above by $\sigma^2$. Our goal is to then analyze the following noisy update rule for FedLin:

$$x_{i,\ell+1}^{(t)} = x_{i,\ell}^{(t)} - \eta_i(q_i(x_{i,\ell}^{(t)}) - q_i(\bar{x}_t) + q(\bar{x}_t)), \tag{89}$$

where $q(x) \triangleq 1/m \sum_{i \in \mathcal{S}} q_i(x), \forall x \in \mathbb{R}^d$. For our subsequent analysis, we will use $\mathcal{F}_{i,\ell}^{(t)}$ to denote the filtration that captures all the randomness up to the $\ell$-th local step of client $i$ in round $t$. We will also use $\mathcal{F}^{(t)}$ to represent the filtration capturing all the randomness up to the end of round $t-1$. With a slight abuse of notation, $\mathcal{F}_{i,-1}^{(t)}$ is to be interpreted as $\mathcal{F}^{(t)}, \forall i \in \mathcal{C}$.

We begin our analysis of Theorem 4 with the following lemma.

**Lemma 12.** *Suppose each $f_i(x)$ is $L$-smooth. Moreover, suppose $\tau_i \geq 1, \forall i \in \mathcal{S}, \delta_c = \delta_s = 1$ and $\eta_i = \dfrac{\bar{\eta}}{\tau_i}, \forall i \in \mathcal{S}$, where $\bar{\eta} \in (0,1)$. Under the stochastic oracle model defined in Section 4, FedLin guarantees:*

$$\mathbb{E}\Big[\|\bar{x}_{t+1} - x^*\|^2\Big] \leq \mathbb{E}\Big[\|\bar{x}_t - x^*\|^2\Big] - 2\bar{\eta}\mathbb{E}\Big[f(\bar{x}_t) - f(x^*)\Big] + 4\bar{\eta}^2\mathbb{E}\Big[\|\nabla f(\bar{x}_t)\|^2\Big]$$
$$+ \frac{L(1 + 6\bar{\eta}L)}{m} \sum_{i=1}^m \eta_i \sum_{\ell=0}^{\tau_i-1} \mathbb{E}\Big[\|\bar{x}_t - x_{i,\ell}\|^2\Big] + 16\bar{\eta}^2\sigma^2. \tag{90}$$

*Proof.* From the noisy local update rule of FedLin in Eq. (89), we have:

$$\bar{x}_{t+1} = \bar{x}_t - \frac{1}{m} \sum_{i=1}^m \eta_i \sum_{\ell=0}^{\tau_i-1} \Big[q_i(x_{i,\ell}) - q_i(\bar{x}_t) + q(\bar{x}_t)\Big]$$
$$= \bar{x}_t - \frac{1}{m} \sum_{i=1}^m \eta_i \sum_{\ell=0}^{\tau_i-1} q_i(x_{i,\ell}). \tag{91}$$

Hence, we obtain

$$\|\bar{x}_{t+1} - x^*\|^2 = \|\bar{x}_t - x^*\|^2 + \underbrace{\left\|\frac{1}{m} \sum_{i=1}^m \eta_i \sum_{\ell=0}^{\tau_i-1} q_i(x_{i,\ell})\right\|^2}_{\mathcal{A}_1}$$
$$\underbrace{- 2\Big\langle \bar{x}_t - x^*, \frac{1}{m} \sum_{i=1}^m \eta_i \sum_{\ell=0}^{\tau_i-1} q_i(x_{i,\ell})\Big\rangle}_{\mathcal{A}_2}. \tag{92}$$

We begin by bounding the term $\mathcal{A}_1$ in (92) as follows:

$$\mathcal{A}_1 = \left\|\frac{1}{m} \sum_{i=1}^m \eta_i \sum_{\ell=0}^{\tau_i-1} q_i(x_{i,\ell})\right\|^2$$
$$= \left\|\frac{1}{m} \sum_{i=1}^m \eta_i \sum_{\ell=0}^{\tau_i-1} (q_i(x_{i,\ell}) - q_i(\bar{x}_t) + q_i(\bar{x}_t))\right\|^2$$
$$= \left\|\frac{1}{m} \sum_{i=1}^m \eta_i \sum_{\ell=0}^{\tau_i-1} (q_i(x_{i,\ell}) - q_i(\bar{x}_t)) + \bar{\eta}q(\bar{x}_t)\right\|^2 \tag{93}$$
$$\leq 2\underbrace{\left\|\frac{1}{m} \sum_{i=1}^m \eta_i \sum_{\ell=0}^{\tau_i-1} (q_i(x_{i,\ell}) - q_i(\bar{x}_t))\right\|^2}_{\mathcal{T}_1} + \underbrace{2\bar{\eta}^2\|q(\bar{x}_t)\|^2}_{\mathcal{T}_2}.$$

The term $\mathcal{T}_1$ in (93) can be upper bounded as follows:

$$
\begin{aligned}
\mathcal{T}_1 &= 2\left\| \frac{1}{m} \sum_{i=1}^{m} \eta_i \sum_{\ell=0}^{\tau_i-1} \left( q_i(x_{i,\ell}) - q_i(\bar{x}_t) \right) \right\|^2 \\
&\leq \frac{2}{m} \sum_{i=1}^{m} \eta_i^2 \tau_i \sum_{\ell=0}^{\tau_i-1} \| q_i(x_{i,\ell}) - q_i(\bar{x}_t) \|^2 \\
&= \frac{2\bar{\eta}}{m} \sum_{i=1}^{m} \eta_i \sum_{\ell=0}^{\tau_i-1} \| q_i(x_{i,\ell}) - q_i(\bar{x}_t) \|^2 \\
&= \frac{2\bar{\eta}}{m} \sum_{i=1}^{m} \eta_i \sum_{\ell=0}^{\tau_i-1} \| q_i(x_{i,\ell}) - \nabla f_i(x_{i,\ell}) + \nabla f_i(x_{i,\ell}) - \nabla f_i(\bar{x}_t) + \nabla f_i(\bar{x}_t) - q_i(\bar{x}_t) \|^2 \\
&\overset{(b)}{\leq} \frac{6\bar{\eta}}{m} \sum_{i=1}^{m} \eta_i \sum_{\ell=0}^{\tau_i-1} \left( \| q_i(x_{i,\ell}) - \nabla f_i(x_{i,\ell}) \|^2 + \| \nabla f_i(x_{i,\ell}) - \nabla f_i(\bar{x}_t) \|^2 + \| \nabla f_i(\bar{x}_t) - q_i(\bar{x}_t) \|^2 \right),
\end{aligned}
$$

(94)

where steps $(a)$ and $(b)$ follow from an application of equation (36). Taking expectation on both sides of equation (94), we obtain:

$$
\begin{aligned}
\mathbb{E}[\mathcal{T}_1] &\leq \frac{6\bar{\eta}}{m} \sum_{i=1}^{m} \eta_i \sum_{\ell=0}^{\tau_i-1} \left( \mathbb{E}\left[ \mathbb{E}\left[ \| q_i(x_{i,\ell}) - \nabla f_i(x_{i,\ell}) \|^2 | \mathcal{F}_{i,\ell-1}^{(t)} \right] \right] + \mathbb{E}\left[ \| \nabla f_i(x_{i,\ell}) - \nabla f_i(\bar{x}_t) \|^2 \right] \right. \\
&\quad \left. + \mathbb{E}\left[ \mathbb{E}\left[ \| \nabla f_i(\bar{x}_t) - q_i(\bar{x}_t) \|^2 | \mathcal{F}^{(t)} \right] \right] \right) \\
&\overset{(a)}{\leq} \frac{6\bar{\eta}}{m} \sum_{i=1}^{m} \eta_i \sum_{\ell=0}^{\tau_i-1} \left( 2\sigma^2 + L^2 \mathbb{E}\left[ \| x_{i,\ell} - \bar{x}_t \|^2 \right] \right) \\
&\leq 12\bar{\eta}^2 \sigma^2 + \frac{6\bar{\eta} L^2}{m} \sum_{i=1}^{m} \eta_i \sum_{\ell=0}^{\tau_i-1} \mathbb{E}\left[ \| x_{i,\ell} - \bar{x}_t \|^2 \right].
\end{aligned}
$$

(95)

For $(a)$, we used the following facts: (i) $x_{i,\ell}^{(t)}$ is $\mathcal{F}_{i,\ell-1}^{(t)}$-adapted; (ii) $\bar{x}_t$ is $\mathcal{F}^{(t)}$-adapted; (iii) the variance of the noise model is bounded above by $\sigma^2$, and (iv) $L$-smoothness of the loss functions.

Next, the term $\mathcal{T}_2$ in (93) can be upper bounded as follows:

$$
\begin{aligned}
\mathcal{T}_2 &= 2\bar{\eta}^2 \| q(\bar{x}_t) \|^2 \\
&= 2\bar{\eta}^2 \| q(\bar{x}_t) - \nabla f(\bar{x}_t) + \nabla f(\bar{x}_t) \|^2 \\
&= 2\bar{\eta}^2 \left\| \frac{1}{m} \sum_{i=1}^{m} (q_i(\bar{x}_t) - \nabla f_i(\bar{x}_t)) + \nabla f(\bar{x}_t) \right\|^2 \\
&\leq 4\bar{\eta}^2 \left\| \frac{1}{m} \sum_{i=1}^{m} (q_i(\bar{x}_t) - \nabla f_i(\bar{x}_t)) \right\|^2 + 4\bar{\eta}^2 \| \nabla f(\bar{x}_t) \|^2 \\
&\overset{(a)}{\leq} \frac{4\bar{\eta}^2}{m} \sum_{i=1}^{m} \| q_i(\bar{x}_t) - \nabla f_i(\bar{x}_t) \|^2 + 4\bar{\eta}^2 \| \nabla f(\bar{x}_t) \|^2,
\end{aligned}
$$

(96)

where step $(a)$ follows from an application of equation (36). Taking expectation on both sides of equation (96), and using the bounded-variance property, we obtain:

$$
\mathbb{E}[\mathcal{T}_2] \leq 4\bar{\eta}^2 \left( \sigma^2 + \mathbb{E}\left[ \| \nabla f(\bar{x}_t) \|^2 \right] \right).
$$

(97)

We now proceed to bound the expectation of the term $\mathcal{A}_2$ in (92) as follows:

$$
\begin{aligned}
\mathbb{E}[\mathcal{A}_2] &= \mathbb{E}\left[-2\left\langle \bar{x}_t - x^*, \frac{1}{m}\sum_{i=1}^{m}\eta_i\sum_{\ell=0}^{\tau_i-1}q_i(x_{i,\ell})\right\rangle\right] \\
&= \frac{-2}{m}\sum_{i=1}^{m}\eta_i\sum_{\ell=0}^{\tau_i-1}\mathbb{E}\left[\langle \bar{x}_t - x^*, q_i(x_{i,\ell})\rangle\right] \\
&= \frac{-2}{m}\sum_{i=1}^{m}\eta_i\sum_{\ell=0}^{\tau_i-1}\mathbb{E}\left[\mathbb{E}\left[\langle \bar{x}_t - x^*, q_i(x_{i,\ell})\rangle|\mathcal{F}_{i,\ell-1}^{(t)}\right]\right] \\
&\overset{(a)}{=} \frac{-2}{m}\sum_{i=1}^{m}\eta_i\sum_{\ell=0}^{\tau_i-1}\mathbb{E}\left[\langle \bar{x}_t - x^*, \nabla f_i(x_{i,\ell})\rangle\right] \\
&\overset{(b)}{\leq} \frac{-2}{m}\sum_{i=1}^{m}\eta_i\sum_{\ell=0}^{\tau_i-1}\mathbb{E}\left[f_i(\bar{x}_t) - f_i(x^*) - \frac{L}{2}\|x_{i,\ell} - x^*\|^2\right] \\
&= -2\bar{\eta}\mathbb{E}\left[f(\bar{x}_t) - f(x^*)\right] + \frac{L}{m}\sum_{i=1}^{m}\eta_i\sum_{\ell=0}^{\tau_i-1}\mathbb{E}\left[\|x_{i,\ell} - \bar{x}_t\|^2\right].
\end{aligned}
\tag{98}
$$

For step $(a)$, we used the unbiasedness property of the noise model, and for step $(b)$, we employed the same reasoning as that used to arrive at equation (76).

Taking expectation on both sides of equation (92), and combining the bounds in equations (95), (97) and (98) establishes the claim of Lemma 12. $\qquad\square$

For $L$-smooth and $\mu$-strongly convex loss functions, the following lemma provides a bound on the drift at client $i$.

**Lemma 13.** *Suppose each $f_i(x)$ is L-smooth and $\mu$-strongly convex. Moreover, suppose $\tau_i \geq 1$, $\forall i \in \mathcal{S}$, $\delta_c = \delta_s = 1$ and $\eta_i = \dfrac{\bar{\eta}}{\tau_i}$, $\forall i \in \mathcal{S}$, where $\bar{\eta} \in (0,1)$. Under the stochastic oracle model defined in Section 4, FedLin guarantees the following bound for each $i \in \mathcal{S}$, and $\forall \ell \in \{0,\ldots,\tau_i-1\}$:*

$$
\mathbb{E}\left[\|x_{i,\ell} - \bar{x}_t\|^2\right] \leq 6\bar{\eta}^2\mathbb{E}\left[\|\nabla f(\bar{x}_t)\|^2\right] + 27\eta_i\bar{\eta}\sigma^2.
\tag{99}
$$

*Proof.* From the noisy local update rule of Fedlin in Eq. (89), we have:

$$
\begin{aligned}
x_{i,\ell+1} &= x_{i,\ell} - \eta_i\big(q_i(x_{i,\ell}) - q_i(\bar{x}_t) + q(\bar{x}_t)\big) \\
&= x_{i,\ell} - \eta_i\big(q_i(x_{i,\ell}) - \nabla f_i(x_{i,\ell}) + \nabla f_i(x_{i,\ell}) - \nabla f_i(\bar{x}_t) \\
&\quad + \nabla f_i(\bar{x}_t) - q_i(\bar{x}_t) + q(\bar{x}_t)\big).
\end{aligned}
\tag{100}
$$

Hence, we have:

$$
\begin{aligned}
x_{i,\ell+1} - \bar{x}_t &= (x_{i,\ell} - \bar{x}_t) - \eta_i\big(\underbrace{\nabla f_i(x_{i,\ell}) - \nabla f_i(\bar{x}_t) + \nabla f(\bar{x}_t)}_{\mathcal{V}}\big) \\
&\quad - \eta_i\big(\underbrace{q_i(x_{i,\ell}) - \nabla f_i(x_{i,\ell}) + \nabla f_i(\bar{x}_t) - q_i(\bar{x}_t) + q(\bar{x}_t) - \nabla f(\bar{x}_t)}_{\mathcal{W}}\big)
\end{aligned}
\tag{101}
$$

Consequently, we may write:

$$
\begin{aligned}
\|x_{i,\ell+1} - \bar{x}_t\|^2 &= \|(x_{i,\ell} - \bar{x}_t - \eta_i\mathcal{V}) - \eta_i\mathcal{W}\|^2 \\
&= \|(x_{i,\ell} - \bar{x}_t - \eta_i\mathcal{V})\|^2 + \eta_i^2\|\mathcal{W}\|^2 - 2\langle x_{i,\ell} - \bar{x}_t - \eta_i\mathcal{V}, \eta_i\mathcal{W}\rangle.
\end{aligned}
\tag{102}
$$

Taking expectation on both sides of equation (102), we have:

$$
\begin{aligned}
\mathbb{E}\left[\|x_{i,\ell+1} - \bar{x}_t\|^2\right] &= \mathbb{E}\left[\|(x_{i,\ell} - \bar{x}_t - \eta_i\mathcal{V})\|^2\right] + \eta_i^2\mathbb{E}\left[\|\mathcal{W}\|^2\right] \\
&\quad - 2\mathbb{E}\left[\mathbb{E}\left[\langle x_{i,\ell} - \bar{x}_t - \eta_i\mathcal{V}, \eta_i\mathcal{W}\rangle|\mathcal{F}_{i,\ell-1}^{(t)}\right]\right] \\
&\overset{(a)}{=} \underbrace{\mathbb{E}\left[\|x_{i,\ell} - \bar{x}_t - \eta_i\mathcal{V}\|^2\right]}_{\mathcal{A}_1} + \eta_i^2\underbrace{\mathbb{E}\left[\|\mathcal{W}\|^2\right]}_{\mathcal{A}_2}.
\end{aligned}
\tag{103}
$$

For $(a)$, we used the following facts: (i) $x_{i,\ell}$, $\bar{x}_t$, and $\mathcal{V}$, are all adapted to $\mathcal{F}_{i,\ell-1}^{(t)}$, and (ii) $\mathbb{E}\big[\mathcal{W}|\mathcal{F}_{i,\ell-1}^{(t)}\big]=0$ based on fact (i) and the unbiasedness property of the noise model.

We now proceed to bound the term $\mathcal{A}_1$ in equation (103) as follows:

$$
\begin{aligned}
\mathbb{E}\big[\|(x_{i,\ell}-\bar{x}_t-\eta_i\mathcal{V})\|^2\big] &= \mathbb{E}\big[\|x_{i,\ell}-\bar{x}_t-\eta_i(\nabla f_i(x_{i,\ell})-\nabla f_i(\bar{x}_t))-\eta_i\nabla f(\bar{x}_t)\|^2\big] \\
&\le (1+\frac{1}{\gamma})\mathbb{E}\big[\|x_{i,\ell}-\bar{x}_t\|^2\big]+(1+\gamma)\eta_i^2\mathbb{E}\big[\|\nabla f(\bar{x}_t)\|^2\big].
\end{aligned}
\tag{104}
$$

The above inequality follows from the application of equation (35) and Lemma 6.

To bound the term $\mathcal{A}_2$ in equation (103), we proceed as follows:

$$
\begin{aligned}
\|\mathcal{W}\|^2 &= \|q_i(x_{i,\ell})-\nabla f_i(x_{i,\ell})+\nabla f_i(\bar{x}_t)-q_i(\bar{x}_t)+q(\bar{x}_t)-\nabla f(\bar{x}_t)\|^2 \\
&\le 3\|q_i(x_{i,\ell})-\nabla f_i(x_{i,\ell})\|^2+3\|\nabla f_i(\bar{x}_t)-q_i(\bar{x}_t)\|^2 \\
&\quad+3\|q(\bar{x}_t)-\nabla f(\bar{x}_t)\|^2 \\
&\le 3\|q_i(x_{i,\ell})-\nabla f_i(x_{i,\ell})\|^2+3\|\nabla f_i(\bar{x}_t)-q_i(\bar{x}_t)\|^2 \\
&\quad+\frac{3}{m}\sum_{i=1}^{m}\|q_i(\bar{x}_t)-\nabla f_i(\bar{x}_t)\|^2.
\end{aligned}
\tag{105}
$$

Taking expectation on both sides of Eq. (105) and using the bounded-variance property, we obtain:

$$
\mathbb{E}\big[\|\mathcal{W}\|^2\big] \le 3(3\sigma^2) = 9\sigma^2.
\tag{106}
$$

Combining equations (104) and (106), equation (103) becomes:

$$
\mathbb{E}\big[\|x_{i,\ell+1}-\bar{x}_t\|^2\big] \le (1+\frac{1}{\gamma})\mathbb{E}\big[\|x_{i,\ell}-\bar{x}_t\|^2\big]+(1+\gamma)\eta_i^2\mathbb{E}\big[\|\nabla f(\bar{x}_t)\|^2\big]+9\eta_i^2\sigma^2.
\tag{107}
$$

Iterating equation (107) and using $x_{i,0}=\bar{x}_t$, we obtain:

$$
\begin{aligned}
\mathbb{E}\big[\|x_{i,\ell}-\bar{x}_t\|^2\big] &\le \left[(1+\gamma)\eta_i^2\mathbb{E}\big[\|\nabla f(\bar{x}_t)\|^2\big]+9\eta_i^2\sigma^2\right]\left(\sum_{j=0}^{\ell-1}\Big(1+\frac{1}{\gamma}\Big)^j\right) \\
&\le 6\bar{\eta}^2\mathbb{E}\big[\|\nabla f(\bar{x}_t)\|^2\big]+27\eta_i\bar{\eta}\sigma^2,
\end{aligned}
\tag{108}
$$

where we set $\gamma=\tau_i$, and used $\eta_i\tau_i=\bar{\eta}$.

**Completing the proof of Theorem 4:** By substituting the bound on the drift from Lemma 13 in equation (90), and for $12\bar{\eta}L\le 1$, we obtain:

$$
\begin{aligned}
\mathbb{E}\Big[\|\bar{x}_{t+1}-x^*\|^2\Big] &\le \mathbb{E}\Big[\|\bar{x}_t-x^*\|^2\Big]-2\bar{\eta}\mathbb{E}\Big[f(\bar{x}_t)-f(x^*)\Big]+4\bar{\eta}^2\mathbb{E}\Big[\|\nabla f(\bar{x}_t)\|^2\Big] \\
&\quad+2L\Big(6\bar{\eta}^3\mathbb{E}\Big[\|\nabla f(\bar{x}_t)\|^2\Big]+27\bar{\eta}^3\sigma^2\Big)+16\bar{\eta}^2\sigma^2 \\
&\le \mathbb{E}\Big[\|\bar{x}_t-x^*\|^2\Big]-2\bar{\eta}\mathbb{E}\Big[f(\bar{x}_t)-f(x^*)\Big]+6\bar{\eta}^2\mathbb{E}\Big[\|\nabla f(\bar{x}_t)\|^2\Big]+25\bar{\eta}^2\sigma^2 \\
&\overset{(a)}{\le} \mathbb{E}\Big[\|\bar{x}_t-x^*\|^2\Big]-\mathbb{E}\Big[2\bar{\eta}(1-6\bar{\eta}L)(f(\bar{x}_t)-f(x^*))\Big]+25\bar{\eta}^2\sigma^2 \\
&\overset{(b)}{\le} (1-\frac{\bar{\eta}\mu}{2})\mathbb{E}\Big[\|\bar{x}_t-x^*\|^2\Big]+25\bar{\eta}^2\sigma^2.
\end{aligned}
\tag{109}
$$

In the above steps, $(a)$ and $(b)$ follow from the $L$-smoothness and $\mu$-strong convexity of the loss functions, respectively. This completes the proof of Theorem 4. $\qquad\square$

# G Proof of Theorem 6: Gradient Sparsification at Server with no Error-Feedback

In our subsequent analysis, we will make use of three basic properties of a `TOP-k` operator that are summarized in the following lemma.

**Lemma 14.** *Let $\mathcal{C}_\delta : \mathbb{R}^d \to \mathbb{R}^d$ denote the* `TOP-k` *operator, where $\delta = d/k$, and $k \in \{1, \dots, d\}$. Then, given any vector $x \in \mathbb{R}^d$, the following three properties hold.*

- ***Property 1**: $\langle \mathcal{C}_\delta(x), x \rangle = \|\mathcal{C}_\delta(x)\|^2$.*

- ***Property 2**: $\|\mathcal{C}_\delta(x)\|^2 \geq \frac{1}{\delta} \|x\|^2$.*

- ***Property 3**: $\|x - \mathcal{C}_\delta(x)\|^2 \leq \left(1 - \frac{1}{\delta}\right) \|x\|^2$.*

All three properties stated above follow almost directly from the definition of the `TOP-k` operator. For a formal proof, see [49]. We start with the following lemma.

**Lemma 15.** *Suppose each $f_i(x)$ is $L$-smooth. Moreover, suppose $\tau_i \geq 1, \forall i \in \mathcal{S}, \delta_c = 1$, and $\eta_i = \frac{\bar{\eta}}{\tau_i}, \forall i \in \mathcal{S}$, where $\bar{\eta} \in (0,1)$. Then, the variant of* `FedLin` *described in the statement of Theorem 6 guarantees:*

$$
\begin{aligned}
f(\bar{x}_{t+1}) - f(\bar{x}_t) \leq -\bar{\eta}\left(1 - \bar{\eta}L\right) \|g_t\|^2 + &\left(\frac{L}{m} \sum_{i=1}^{m} \eta_i \sum_{\ell=0}^{\tau_i - 1} \|x_{i,\ell} - \bar{x}_t\|\right) \|\nabla f(\bar{x}_t)\| \\
&+ \frac{\bar{\eta}L^3}{m} \sum_{i=1}^{m} \eta_i \sum_{\ell=0}^{\tau_i - 1} \|x_{i,\ell} - \bar{x}_t\|^2.
\end{aligned}
\tag{110}
$$

*Proof.* For the setting under consideration, the local update rule at client $i$ takes the form

$$
x_{i,\ell+1} = x_{i,\ell} - \eta_i(\nabla f_i(x_{i,\ell}) - \nabla f_i(\bar{x}_t) + g_t),
\tag{111}
$$

where

$$
g_t = \mathcal{C}_{\delta_s}\left(\frac{1}{m} \sum_{i=1}^{m} \nabla f_i(\bar{x}_t)\right) = \mathcal{C}_{\delta_s}\left(\nabla f(\bar{x}_t)\right).
\tag{112}
$$

Using $x_{i,0} = \bar{x}_t, \forall i \in \mathcal{S}$, we then have:

$$
\begin{aligned}
x_{i,\tau_i} &= \bar{x}_t - \eta_i \sum_{\ell=0}^{\tau_i - 1} \nabla f_i(x_{i,\ell}) - \eta_i \tau_i (g_t - \nabla f_i(\bar{x}_t)) \\
&= \bar{x}_t - \eta_i \sum_{\ell=0}^{\tau_i - 1} \nabla f_i(x_{i,\ell}) - \bar{\eta}(g_t - \nabla f_i(\bar{x}_t)), \forall i \in \mathcal{S},
\end{aligned}
\tag{113}
$$

where we used $\eta_i \tau_i = \bar{\eta}$ in the second step. Thus,

$$
\begin{aligned}
\bar{x}_{t+1} = \frac{1}{m} \sum_{i=1}^{m} x_{i,\tau_i} &= \bar{x}_t - \frac{1}{m} \sum_{i=1}^{m} \eta_i \sum_{\ell=0}^{\tau_i - 1} \nabla f_i(x_{i,\ell}) - \frac{\bar{\eta}}{m} \sum_{i=1}^{m} (g_t - \nabla f_i(\bar{x}_t)) \\
&= \bar{x}_t - \frac{1}{m} \sum_{i=1}^{m} \eta_i \sum_{\ell=0}^{\tau_i - 1} \nabla f_i(x_{i,\ell}) - \bar{\eta}\left(g_t - \nabla f(\bar{x}_t)\right).
\end{aligned}
\tag{114}
$$

Compared to (64), note that we have an additional error term $\bar{\eta}(g_t - \nabla f(\bar{x}_t))$ that shows up as a consequence of gradient sparsification at the server. Nonetheless, we proceed exactly as before, and

bound $f(\bar{x}_{t+1}) - f(\bar{x}_t)$ as follows:

$$f(\bar{x}_{t+1}) - f(\bar{x}_t) \leq \langle \bar{x}_{t+1} - \bar{x}_t, \nabla f(\bar{x}_t) \rangle + \frac{L}{2} \|\bar{x}_{t+1} - \bar{x}_t\|^2$$

$$= \Big\langle -\frac{1}{m} \sum_{i=1}^{m} \eta_i \sum_{\ell=0}^{\tau_i-1} \nabla f_i(x_{i,\ell}) + \bar{\eta}(\nabla f(\bar{x}_t) - g_t), \nabla f(\bar{x}_t) \Big\rangle$$

$$+ \frac{L}{2} \Big\| \frac{1}{m} \sum_{i=1}^{m} \eta_i \sum_{\ell=0}^{\tau_i-1} \nabla f_i(x_{i,\ell}) + \bar{\eta}(g_t - \nabla f(\bar{x}_t)) \Big\|^2$$

$$\overset{(a)}{=} -\Big\langle \frac{1}{m} \sum_{i=1}^{m} \eta_i \sum_{\ell=0}^{\tau_i-1} (\nabla f_i(x_{i,\ell}) - \nabla f_i(\bar{x}_t)), \nabla f(\bar{x}_t) \Big\rangle - \bar{\eta} \langle g_t, \nabla f(\bar{x}_t) \rangle$$

$$+ \frac{L}{2} \Big\| \frac{1}{m} \sum_{i=1}^{m} \eta_i \sum_{\ell=0}^{\tau_i-1} (\nabla f_i(x_{i,\ell}) - \nabla f_i(\bar{x}_t)) + \bar{\eta} g_t \Big\|^2$$

$$\overset{(b)}{=} -\Big\langle \frac{1}{m} \sum_{i=1}^{m} \eta_i \sum_{\ell=0}^{\tau_i-1} (\nabla f_i(x_{i,\ell}) - \nabla f_i(\bar{x}_t)), \nabla f(\bar{x}_t) \Big\rangle - \bar{\eta} \|g_t\|^2$$

$$+ \frac{L}{2} \Big\| \frac{1}{m} \sum_{i=1}^{m} \eta_i \sum_{\ell=0}^{\tau_i-1} (\nabla f_i(x_{i,\ell}) - \nabla f_i(\bar{x}_t)) + \bar{\eta} g_t \Big\|^2$$

$$\overset{(c)}{\leq} -\bar{\eta}(1 - \bar{\eta}L) \|g_t\|^2 \underbrace{-\Big\langle \frac{1}{m} \sum_{i=1}^{m} \eta_i \sum_{\ell=0}^{\tau_i-1} (\nabla f_i(x_{i,\ell}) - \nabla f_i(\bar{x}_t)), \nabla f(\bar{x}_t) \Big\rangle}_{T_1}$$

$$+ \underbrace{L \Big\| \frac{1}{m} \sum_{i=1}^{m} \eta_i \sum_{\ell=0}^{\tau_i-1} (\nabla f_i(x_{i,\ell}) - \nabla f_i(\bar{x}_t)) \Big\|^2}_{T_2}$$

$$\overset{(d)}{\leq} -\bar{\eta}(1 - \bar{\eta}L) \|g_t\|^2 + \Big( \frac{L}{m} \sum_{i=1}^{m} \eta_i \sum_{\ell=0}^{\tau_i-1} \|x_{i,\ell} - \bar{x}_t\| \Big) \|\nabla f(\bar{x}_t)\|$$

$$+ \frac{\bar{\eta} L^3}{m} \sum_{i=1}^{m} \eta_i \sum_{\ell=0}^{\tau_i-1} \|x_{i,\ell} - \bar{x}_t\|^2.$$

(115)

In the above steps, for arriving at (a), we made the following observation:

$$\bar{\eta} \nabla f(\bar{x}_t) = \frac{1}{m} \sum_{i=1}^{m} \bar{\eta} \nabla f_i(\bar{x}_t) = \frac{1}{m} \sum_{i=1}^{m} \eta_i \tau_i \nabla f_i(\bar{x}_t) = \frac{1}{m} \sum_{i=1}^{m} \eta_i \sum_{\ell=0}^{\tau_i-1} \nabla f_i(\bar{x}_t). \qquad (116)$$

For (b), observe that $\langle g_t, \nabla f(\bar{x}_t) \rangle = \langle \mathcal{C}_{\delta_s}(\nabla f(\bar{x}_t)), \nabla f(\bar{x}_t) \rangle = \|g_t\|^2$, where the second equality follows from Property 1 of the TOP-k operator in Lemma 14. For (c), we used (35) with $\gamma = 1$. For (d), we followed the arguments used to arrive at (66) and (67) to bound $T_1$ and $T_2$, respectively. $\qquad \square$

**Completing the proof of Theorem 6:** To complete the proof of Theorem 6, we start by noting that if the step-size at client $i$ satisfies $\eta_i \leq \frac{1}{L}$, then arguments identical to those used for proving Lemma 9 can be used to conclude that

$$\|x_{i,\ell} - \bar{x}_t\| \leq \eta_i \tau_i \|g_t\|. \tag{117}$$

Substituting the above bound in (110) yields:

$$
\begin{aligned}
f(\bar{x}_{t+1}) - f(\bar{x}_t) &\leq -\bar{\eta}\left(1 - \bar{\eta}L\right)\|g_t\|^2 + \left(\frac{L}{m}\sum_{i=1}^{m}\eta_i\sum_{\ell=0}^{\tau_i-1}\eta_i\tau_i\|g_t\|\right)\|\nabla f(\bar{x}_t)\| \\
&\quad + \frac{\bar{\eta}L^3}{m}\sum_{i=1}^{m}\eta_i\sum_{\ell=0}^{\tau_i-1}\left(\eta_i\tau_i\|g_t\|\right)^2 \\
&= -\bar{\eta}\left(1 - \bar{\eta}L\right)\|g_t\|^2 + \frac{L}{m}\sum_{i=1}^{m}(\eta_i\tau_i)^2\|g_t\|\|\nabla f(\bar{x}_t)\| + \frac{\bar{\eta}L^3}{m}\sum_{i=1}^{m}(\eta_i\tau_i)^3\|g_t\|^2 \\
&\stackrel{(a)}{\leq} -\bar{\eta}\left(1 - \bar{\eta}L\right)\|g_t\|^2 + \frac{\sqrt{\delta_s}L}{m}\sum_{i=1}^{m}(\eta_i\tau_i)^2\|g_t\|^2 + \frac{\bar{\eta}L^3}{m}\sum_{i=1}^{m}(\eta_i\tau_i)^3\|g_t\|^2 \\
&\stackrel{(b)}{\leq} -\bar{\eta}\left(1 - \left(2 + \sqrt{\delta_s}\right)\bar{\eta}L\right)\|g_t\|^2 \\
&\stackrel{(c)}{\leq} -\frac{\bar{\eta}}{\delta_s}\left(1 - \left(2 + \sqrt{\delta_s}\right)\bar{\eta}L\right)\|\nabla f(\bar{x}_t)\|^2 \\
&\stackrel{(d)}{\leq} -\frac{2\bar{\eta}\mu}{\delta_s}\left(1 - \left(2 + \sqrt{\delta_s}\right)\bar{\eta}L\right)\left(f(\bar{x}_t) - f(x^*)\right).
\end{aligned}
\tag{118}
$$

In the above steps, for (a), we used Property 2 of the TOP-k operator in Lemma 14 to conclude that $\|\nabla f(\bar{x}_t)\| \leq \sqrt{\delta_s}\|\mathcal{C}_{\delta_s}(\nabla f(\bar{x}_t))\| = \sqrt{\delta_s}\|g_t\|$. For (b), we used $\eta_i\tau_i = \bar{\eta}$ and $\bar{\eta}L \leq 1$. For (c), we once again used the second property of the TOP-k operator, and for (d), we used the fact that $f(\cdot)$ is $\mu$-strongly convex (refer to (34)). Setting $\bar{\eta} = \frac{1}{2(2+\sqrt{\delta_s})L}$ and rearranging terms then leads to

$$f(\bar{x}_{t+1}) - f(x^*) \leq \left(1 - \frac{1}{2\delta_s\left(2 + \sqrt{\delta_s}\right)\kappa}\right)(f(\bar{x}_t) - f(x^*)), \text{ where } \kappa = \frac{L}{\mu}. \tag{119}$$

Using the above inequality recursively, we obtain the desired conclusion.

# H  Proof of Theorem 8: Gradient Sparsification at Clients

The proof of Theorem 8 is somewhat more involved than Theorem 6. Let us begin by compiling the governing equations for the setting under consideration.

$$
\begin{aligned}
x_{i,\ell+1} &= x_{i,\ell} - \eta_i(\nabla f_i(x_{i,\ell}) - \nabla f_i(\bar{x}_t) + g_t) \\
h_{i,t} &= \mathcal{C}_{\delta_c}\left(\rho_{i,t-1} + \nabla f_i(\bar{x}_t)\right) \\
\rho_{i,t} &= \rho_{i,t-1} + \nabla f_i(\bar{x}_t) - h_{i,t}.
\end{aligned}
\tag{120}
$$

The first and the third equations hold for every communication round $t \in \{1, \ldots, T\}$, whereas the second equation holds $\forall t \in \{2, \ldots, T\}$. Moreover, we have $h_{i,1} = \nabla f_i(\bar{x}_1)$, and $\rho_{i,0} = \rho_{i,1} = 0, \forall i \in \mathcal{S}$, i.e., the initial gradient compression errors are 0 at each client. Since there is no further gradient sparsification at the server, we have $g_t = \frac{1}{m}\sum_{i=1}^{m} h_{i,t}$. It then follows that

$$
\rho_t = \rho_{t-1} + \nabla f(\bar{x}_t) - g_t,
\tag{121}
$$

where $\rho_t \triangleq \frac{1}{m}\sum_{i=1}^{m}\rho_{i,t}$. To simplify the analysis, let us define a virtual sequence $\{\tilde{x}_t\}$ as follows:[8]

$$
\tilde{x}_t \triangleq \bar{x}_t - \bar{\eta}\rho_{t-1},
\tag{122}
$$

where $\bar{\eta} = \eta_i \tau_i, \forall i \in \mathcal{S}$. Now observe that

$$
\begin{aligned}
\tilde{x}_{t+1} &= \bar{x}_{t+1} - \bar{\eta}\rho_t \\
&= \bar{x}_t - \frac{1}{m}\sum_{i=1}^{m}\eta_i\sum_{\ell=0}^{\tau_i-1}\nabla f_i(x_{i,\ell}) - \bar{\eta}\left(g_t - \nabla f(\bar{x}_t)\right) - \bar{\eta}\left(\rho_{t-1} + \nabla f(\bar{x}_t) - g_t\right) \\
&= \tilde{x}_t - \frac{1}{m}\sum_{i=1}^{m}\eta_i\sum_{\ell=0}^{\tau_i-1}\nabla f_i(x_{i,\ell}).
\end{aligned}
\tag{123}
$$

The second equality follows from (114) and (121), and the third follows from the definition of $\tilde{x}_t$ in (122). Interestingly, note that the recursion for $\tilde{x}_t$ that we just derived in (123) resembles that for $\bar{x}_t$ in (64) where there was no effect of gradient sparsification. This simplified recursion reveals the utility of the virtual sequence.

**Proof idea:** In order to argue that $\bar{x}_t$ converges to $x^*$, it clearly suffices to argue that the virtual sequence $\tilde{x}_t$ converges to $x^*$, and $\bar{x}_t$ converges to $\tilde{x}_t$. To achieve this, we will employ the following Lyapunov function in our analysis:

$$
\psi_t \triangleq \|\tilde{x}_t - x^*\|^2 + \bar{\eta}^2 V_{t-1}, \text{ where } V_t \triangleq \frac{1}{m}\sum_{i=1}^{m}\|\rho_{i,t}\|^2.
\tag{124}
$$

The choice of the above Lyapunov function is specific to our setting, and accounts for the effects of systems heterogeneity and gradient sparsification. In the following lemma, we bound the first part of the Lyapunov function, namely the distance of the virtual iterate $\tilde{x}_t$ from the optimal point $x^*$.

**Lemma 16.** *Suppose each $f_i(x)$ is $L$-smooth and $\mu$-strongly convex, and suppose Assumption 1 holds. Moreover, suppose $\tau_i \geq 1, \forall i \in \mathcal{S}$, and $\delta_s = 1$. Let the step-size for client $i$ be chosen as $\eta_i = \frac{\bar{\eta}}{\tau_i}$, where $\bar{\eta} \in (0,1)$ satisfies $\bar{\eta} \leq \frac{1}{2LC}$. Then, `FedLin` guarantees:*

$$
\|\tilde{x}_{t+1} - x^*\|^2 \leq \|\tilde{x}_t - x^*\|^2 - 2\bar{\eta}(f(\tilde{x}_t) - f(x^*)) + 14\bar{\eta}^2\|\nabla f(\tilde{x}_t)\|^2 + 24\bar{\eta}^3 LV_{t-1} + 12\bar{\eta}^3 LD.
\tag{125}
$$

*Proof.* From (123), we obtain

$$
\|\tilde{x}_{t+1} - x^*\|^2 = \|\tilde{x}_t - x^*\|^2 - \frac{2}{m}\langle\tilde{x}_t - x^*, \sum_{i=1}^{m}\eta_i\sum_{\ell=0}^{\tau_i-1}\nabla f_i(x_{i,\ell})\rangle + \left\|\frac{1}{m}\sum_{i=1}^{m}\eta_i\sum_{\ell=0}^{\tau_i-1}\nabla f_i(x_{i,\ell})\right\|^2.
\tag{126}
$$

---

[8]We note that virtual sequences and perturbed iterates are commonly used to simplify proofs in the context of analyzing compression schemes [52, 49], and asynchronous methods [56].

To bound the second and third terms in the above equation, we can follow exactly the same steps as those in Lemma 10. In particular, Lemma 10 relied on $L$-smoothness and convexity of each $f_i(\cdot)$ - each of these properties apply to our current setting. Referring to (73), we thus have

$$\|\tilde{x}_{t+1} - x^*\|^2 \leq \|\tilde{x}_t - x^*\|^2 - 2\bar{\eta}\left(f(\tilde{x}_t) - f(x^*)\right) + \frac{L(1 + 2\bar{\eta}L)}{m} \sum_{i=1}^{m} \eta_i \sum_{\ell=0}^{\tau_i - 1} \|x_{i,\ell} - \tilde{x}_t\|^2$$

$$+ 2\bar{\eta}^2 \|\nabla f(\tilde{x}_t)\|^2. \tag{127}$$

To bound the term $\|\tilde{x}_t - x_{i,\ell}\|^2$, start by observing that

$$\|\tilde{x}_t - x_{i,\ell}\|^2 = \|\tilde{x}_t - \bar{x}_t + \bar{x}_t - x_{i,\ell}\|^2$$
$$\leq 2\|\tilde{x}_t - \bar{x}_t\|^2 + 2\|\bar{x}_t - x_{i,\ell}\|^2 \tag{128}$$
$$= 2\bar{\eta}^2 \|\rho_{t-1}\|^2 + 2\|\bar{x}_t - x_{i,\ell}\|^2,$$

where the last equality follows from (122). Since $\bar{\eta} \leq \frac{1}{L}$, following the same arguments as in Lemma 9, we have $\|\bar{x}_t - x_{i,\ell}\| \leq \eta_i \tau_i \|g_t\| = \bar{\eta}\|g_t\|$. Thus,

$$\|\tilde{x}_t - x_{i,\ell}\|^2 \leq 2\bar{\eta}^2 \left(\|\rho_{t-1}\|^2 + \|g_t\|^2\right). \tag{129}$$

Next, note that

$$\|g_t\|^2 = \left\|\frac{1}{m}\sum_{i=1}^{m} h_{i,t}\right\|^2$$

$$\overset{(a)}{\leq} \frac{1}{m}\sum_{i=1}^{m} \|h_{i,t}\|^2$$

$$\overset{(b)}{\leq} \frac{1}{m}\sum_{i=1}^{m} \|\rho_{i,t-1} + \nabla f_i(\bar{x}_t)\|^2$$

$$\overset{(c)}{\leq} \frac{2}{m}\sum_{i=1}^{m} \|\rho_{i,t-1}\|^2 + \frac{2}{m}\sum_{i=1}^{m} \|\nabla f_i(\bar{x}_t)\|^2 \tag{130}$$

$$\overset{(d)}{=} 2V_{t-1} + \frac{2}{m}\sum_{i=1}^{m} \|\nabla f_i(\bar{x}_t)\|^2$$

$$\overset{(e)}{\leq} 2V_{t-1} + 2C\|\nabla f(\bar{x}_t)\|^2 + 2D.$$

In the above steps, (a) follows from Jensen's inequality; (b) follows from the fact that $h_{i,t} = \mathcal{C}_{\delta_c}\left(\rho_{i,t-1} + \nabla f_i(\bar{x}_t)\right)$, and the definition of the TOP-k operation; (c) follows from (35) with $\gamma = 1$; (d) follows from the definition of $V_{t-1}$, and (e) follows from Assumption 1. Finally, observe that

$$\|\nabla f(\bar{x}_t)\|^2 = \|\nabla f(\bar{x}_t) - \nabla f(\tilde{x}_t) + \nabla f(\tilde{x}_t)\|^2$$
$$\leq 2\|\nabla f(\bar{x}_t) - \nabla f(\tilde{x}_t)\|^2 + 2\|\nabla f(\tilde{x}_t)\|^2$$
$$\overset{(a)}{\leq} 2L^2\|\bar{x}_t - \tilde{x}_t\|^2 + 2\|\nabla f(\tilde{x}_t)\|^2 \tag{131}$$
$$\overset{(b)}{=} 2\bar{\eta}^2 L^2\|\rho_{t-1}\|^2 + 2\|\nabla f(\tilde{x}_t)\|^2$$
$$\overset{(c)}{\leq} 2\bar{\eta}^2 L^2 V_{t-1} + 2\|\nabla f(\tilde{x}_t)\|^2,$$

where for (a), we used the $L$-smoothness of $f(\cdot)$; for (b), we used (122), and for (c), we used Jensen's inequality to bound $\|\rho_{t-1}\|^2$ by $V_{t-1}$. Combining the bounds (129), (130) and (131), we obtain:

$$\|\tilde{x}_t - x_{i,\ell}\|^2 \leq 2\bar{\eta}^2 \left(\|\rho_{t-1}\|^2 + 2V_{t-1} + 2C\|\nabla f(\bar{x}_t)\|^2 + 2D\right)$$
$$\leq 2\bar{\eta}^2 \left((3 + 4\bar{\eta}^2 L^2 C)V_{t-1} + 4C\|\nabla f(\tilde{x}_t)\|^2 + 2D\right) \tag{132}$$
$$\leq 4\bar{\eta}^2 \left(2V_{t-1} + 2C\|\nabla f(\tilde{x}_t)\|^2 + D\right).$$

For the second inequality, we once again used Jensen's to conclude $\|\rho_{t-1}\|^2 \leq V_{t-1}$; for the last inequality, we used $\bar{\eta}^2 L^2 C \leq \bar{\eta}^2 L^2 C^2 \leq \frac{1}{4}$, which in turn follows from $C \geq 1$, and the fact that $\bar{\eta} \leq \frac{1}{2LC}$ based on our choice of step-size. Plugging the bound on $\|\tilde{x}_t - x_{i,\ell}\|^2$ in (127), we have

$$
\begin{aligned}
\|\tilde{x}_{t+1} - x^*\|^2 &\leq \|\tilde{x}_t - x^*\|^2 - 2\bar{\eta}\left(f(\tilde{x}_t) - f(x^*)\right) + 2\bar{\eta}^2\|\nabla f(\tilde{x}_t)\|^2 \\
&\quad + \frac{L(1 + 2\bar{\eta}L)}{m}\sum_{i=1}^{m}\eta_i\sum_{\ell=0}^{\tau_i-1}4\bar{\eta}^2\left(2V_{t-1} + 2C\|\nabla f(\tilde{x}_t)\|^2 + D\right) \\
&\leq \|\tilde{x}_t - x^*\|^2 - 2\bar{\eta}\left(f(\tilde{x}_t) - f(x^*)\right) + 2\bar{\eta}^2\|\nabla f(\tilde{x}_t)\|^2 \\
&\quad + \frac{3L}{m}\sum_{i=1}^{m}4\bar{\eta}^3\left(2V_{t-1} + 2C\|\nabla f(\tilde{x}_t)\|^2 + D\right) \\
&= \|\tilde{x}_t - x^*\|^2 - 2\bar{\eta}\left(f(\tilde{x}_t) - f(x^*)\right) + 2\bar{\eta}^2\left(1 + 12\bar{\eta}LC\right)\|\nabla f(\tilde{x}_t)\|^2 \\
&\quad + 24\bar{\eta}^3 LV_{t-1} + 12\bar{\eta}^3 LD \\
&\leq \|\tilde{x}_t - x^*\|^2 - 2\bar{\eta}\left(f(\tilde{x}_t) - f(x^*)\right) + 14\bar{\eta}^2\|\nabla f(\tilde{x}_t)\|^2 + 24\bar{\eta}^3 LV_{t-1} + 12\bar{\eta}^3 LD.
\end{aligned}
\tag{133}
$$

For the second inequality, we used $\bar{\eta}L \leq 1$ and $\bar{\eta} = \eta_i\tau_i$, and for the last inequality, we used $\bar{\eta}LC \leq \frac{1}{2}$. This concludes the proof. $\qquad\square$

In the next lemma, we derive a recursion to bound $V_t$ - a measure of the sparsification error.

**Lemma 17.** *Suppose the conditions stated in Lemma 16 hold. Then, we have*

$$
V_t \leq \left(1 - \frac{1}{2\delta_c} + 4\bar{\eta}^2 L^2\delta_c C\right)V_{t-1} + 4\delta_c C\|\nabla f(\tilde{x}_t)\|^2 + 2\delta_c D.
\tag{134}
$$

*Proof.* Let us observe that

$$
\begin{aligned}
V_t &= \frac{1}{m}\sum_{i=1}^{m}\|\rho_{i,t}\|^2 \\
&= \frac{1}{m}\sum_{i=1}^{m}\|\rho_{i,t-1} + \nabla f_i(\bar{x}_t) - h_{i,t}\|^2 \\
&= \frac{1}{m}\sum_{i=1}^{m}\|\rho_{i,t-1} + \nabla f_i(\bar{x}_t) - \mathcal{C}_{\delta_c}\left(\rho_{i,t-1} + \nabla f_i(\bar{x}_t)\right)\|^2 \\
&\leq \left(1 - \frac{1}{\delta_c}\right)\frac{1}{m}\sum_{i=1}^{m}\|\rho_{i,t-1} + \nabla f_i(\bar{x}_t)\|^2 \\
&\leq \left(1 - \frac{1}{\delta_c}\right)(1+\gamma)V_{t-1} + \left(1 - \frac{1}{\delta_c}\right)\left(1 + \frac{1}{\gamma}\right)\frac{1}{m}\sum_{i=1}^{m}\|\nabla f_i(\bar{x}_t)\|^2.
\end{aligned}
\tag{135}
$$

For the second-last inequality, we used Property 3 of the TOP-k operator in Lemma 14; for the last inequality, we used the definition of $V_{t-1}$ and the relaxed triangle inequality (35). Now in order for $V_t$ to contract over time, we must have

$$
\left(1 - \frac{1}{\delta_c}\right)(1+\gamma) < 1 \implies \gamma < \frac{1}{\delta_c - 1}.
$$

Accordingly, suppose $\gamma = \frac{1}{2(\delta_c - 1)}$. Simple calculations then yield

$$
\left(1 - \frac{1}{\delta_c}\right)(1+\gamma) = 1 - \frac{1}{2\delta_c}; \qquad \left(1 - \frac{1}{\delta_c}\right)\left(1 + \frac{1}{\gamma}\right) = \left(1 - \frac{1}{\delta_c}\right)(2\delta_c - 1) < 2\delta_c.
$$

Substituting the above bounds in (135), and invoking Assumption 1, we obtain

$$
\begin{aligned}
V_t &\leq \left(1 - \frac{1}{2\delta_c}\right) V_{t-1} + 2\delta_c \left(C\|\nabla f(\bar{x}_t)\|^2 + D\right) \\
&\leq \left(1 - \frac{1}{2\delta_c}\right) V_{t-1} + 2\delta_c C \left(2\bar{\eta}^2 L^2 V_{t-1} + 2\|\nabla f(\tilde{x}_t)\|^2\right) + 2\delta_c D \\
&= \left(1 - \frac{1}{2\delta_c} + 4\bar{\eta}^2 L^2 \delta_c C\right) V_{t-1} + 4\delta_c C\|\nabla f(\tilde{x}_t)\|^2 + 2\delta_c D,
\end{aligned}
\tag{136}
$$

where for the second inequality, we used (131). $\qquad \square$

Now that we have a handle over each of the two components of the Lyapunov function $\psi_{t+1}$, we are in a position to complete the proof of Theorem 8.

**Completing the proof of Theorem 8:** Suppose $\bar{\eta}$ is chosen such that $\bar{\eta} \leq \frac{1}{72L\delta_c C}$. Note that this choice of $\bar{\eta}$ meets the requirements for Lemmas 16 and 17 to hold. Now based on Lemmas 16 and 17, and the definition of $\psi_t$, we have

$$
\begin{aligned}
\psi_{t+1} &= \|\tilde{x}_{t+1} - x^*\|^2 + \bar{\eta}^2 V_t \\
&\leq \|\tilde{x}_t - x^*\|^2 - 2\bar{\eta}\left(f(\tilde{x}_t) - f(x^*)\right) + 2\bar{\eta}^2 (7 + 2\delta_c C)\|\nabla f(\tilde{x}_t)\|^2 \\
&\quad + \left(1 - \frac{1}{2\delta_c} + 4\bar{\eta}^2 L^2 \delta_c C + 24\bar{\eta}L\right) \bar{\eta}^2 V_{t-1} + 2\bar{\eta}^2 (6\bar{\eta}L + \delta_c)D \\
&\overset{(a)}{\leq} \|\tilde{x}_t - x^*\|^2 - 2\bar{\eta}\left(f(\tilde{x}_t) - f(x^*)\right) + 18\bar{\eta}^2 \delta_c C\|\nabla f(\tilde{x}_t)\|^2 \\
&\quad + \left(1 - \frac{1}{2\delta_c} + 28\bar{\eta}L\right) \bar{\eta}^2 V_{t-1} + 2\bar{\eta}^2 \left(\frac{6}{\delta_c C} + \delta_c\right) D \\
&\overset{(b)}{\leq} \|\tilde{x}_t - x^*\|^2 - 2\bar{\eta}\left(1 - 18\bar{\eta}L\delta_c C\right)\left(f(\tilde{x}_t) - f(x^*)\right) + \left(1 - \frac{1}{2\delta_c} + 28\bar{\eta}L\right) \bar{\eta}^2 V_{t-1} \\
&\quad + 2\bar{\eta}^2 \left(\frac{6}{\delta_c C} + \delta_c\right) D \\
&\overset{(c)}{\leq} \left(1 - \frac{3}{4}\bar{\eta}\mu\right) \|\tilde{x}_t - x^*\|^2 + \left(1 - \frac{1}{2\delta_c} + 28\bar{\eta}L\right) \bar{\eta}^2 V_{t-1} + 2\bar{\eta}^2 \left(\frac{6}{\delta_c C} + \delta_c\right) D.
\end{aligned}
\tag{137}
$$

For (a), we used $\delta_c \geq 1, C \geq 1$, and $\bar{\eta}L\delta_c C \leq 1$ to simplify the preceding inequality; for (b), we used the $L$-smoothness of $f(\cdot)$; for (c), we used the fact that $\bar{\eta}L\delta_c C \leq \frac{1}{72}$, and that $f(\cdot)$ is $\mu$-strongly convex. Now given our choice of $\bar{\eta}$, observe that

$$
1 - \frac{1}{2\delta_c} + 28\bar{\eta}L \leq 1 - \frac{1}{2\delta_c C} + \frac{28}{72\delta_c C} = 1 - \frac{1}{9\delta_c C} < 1 - \frac{1}{96\delta_c C\kappa} \leq 1 - \frac{3}{4}\bar{\eta}\mu,
$$

where $\kappa = \frac{L}{\mu}$. Thus,

$$
\begin{aligned}
\psi_{t+1} &\leq \left(1 - \frac{3}{4}\bar{\eta}\mu\right) \left(\|\tilde{x}_t - x^*\|^2 + \bar{\eta}^2 V_{t-1}\right) + 2\bar{\eta}^2 \left(\frac{6}{\delta_c C} + \delta_c\right) D \\
&= \left(1 - \frac{3}{4}\bar{\eta}\mu\right) \psi_t + 2\bar{\eta}^2 \left(\frac{6}{\delta_c C} + \delta_c\right) D.
\end{aligned}
\tag{138}
$$

Using the above inequality recursively, we obtain

$$
\begin{aligned}
\psi_{T+1} &\leq \left(1 - \frac{3}{4}\bar{\eta}\mu\right)^T \psi_1 + 2\bar{\eta}^2 \left(\frac{1 - \left(1 - \frac{3}{4}\bar{\eta}\mu\right)^T}{1 - \left(1 - \frac{3}{4}\bar{\eta}\mu\right)}\right) \left(\frac{6}{\delta_c C} + \delta_c\right) D \\
&\leq \left(1 - \frac{3}{4}\bar{\eta}\mu\right)^T \psi_1 + \frac{8}{3}\bar{\eta} \left(\frac{6}{\delta_c C} + \delta_c\right) \frac{D}{\mu}.
\end{aligned}
\tag{139}
$$

Now since $\rho_{i,0} = 0, \forall i \in \mathcal{S}$, we have $\rho_0 = 0$, and $V_0 = 0$. It thus follows that $\tilde{x}_1 = \bar{x}_1 - \bar{\eta}\rho_0 = \bar{x}_1$, and $\psi_1 = \|\tilde{x}_1 - x^*\|^2 + \bar{\eta}^2 V_0 = \|\bar{x}_1 - x^*\|^2$. Finally, observe that

$$
\begin{aligned}
\|\bar{x}_{T+1} - x^*\|^2 &= \|\bar{x}_{T+1} - \tilde{x}_{T+1} + \tilde{x}_{T+1} - x^*\|^2 \\
&\leq 2\|\tilde{x}_{T+1} - x^*\|^2 + 2\|\bar{x}_{T+1} - \tilde{x}_{T+1}\|^2 \\
&= 2\|\tilde{x}_{T+1} - x^*\|^2 + 2\bar{\eta}^2\|\rho_T\|^2 \\
&\leq 2\left(\|\tilde{x}_{T+1} - x^*\|^2 + \bar{\eta}^2 V_T\right) \\
&= 2\psi_{T+1}.
\end{aligned}
\tag{140}
$$

Based on the above discussion, and (139), we have

$$
\|\bar{x}_{T+1} - x^*\|^2 \leq 2\left(1 - \frac{3}{4}\bar{\eta}\mu\right)^T \|\bar{x}_1 - x^*\|^2 + \frac{16}{3}\bar{\eta}\left(\frac{6}{\delta_c C} + \delta_c\right)\frac{D}{\mu},
\tag{141}
$$

which is precisely the desired conclusion.

# I  Proof of Theorem 7: Gradient Sparsification at Server with Error-Feedback

The proof of Theorem 7 follows similar conceptual steps as that of Theorem 8. Thus, we will only sketch the main arguments, leaving the reader to verify the details. Given that $\delta_c = 1$, we note that the governing equations for this setting are as follows.

$$
\begin{aligned}
x_{i,\ell+1} &= x_{i,\ell} - \eta_i(\nabla f_i(x_{i,\ell}) - \nabla f_i(\bar{x}_t) + g_t) \\
g_t &= \mathcal{C}_{\delta_s}\left(e_{t-1} + \nabla f(\bar{x}_t)\right) \\
e_t &= e_{t-1} + \nabla f(\bar{x}_t) - g_t.
\end{aligned}
\tag{142}
$$

The first and the third equations above hold for every communication round $t \in \{1, \ldots, T\}$, whereas the second holds for every $t \in \{2, \ldots, T\}$. Initially, we have $g_1 = \nabla f(\bar{x}_1)$, and $e_1 = e_0 = 0$. Let us define the sequence $\{\tilde{x}_t\}$ as follows:

$$
\tilde{x}_t = \bar{x}_t - \bar{\eta} e_{t-1},
\tag{143}
$$

where $\bar{\eta} = \eta_i \tau_i, \forall i \in \mathcal{S}$. Then, based on the definition of the virtual sequence, and (142), it is easy to verify that

$$
\tilde{x}_{t+1} = \tilde{x}_t - \frac{1}{m} \sum_{i=1}^{m} \eta_i \sum_{\ell=0}^{\tau_i-1} \nabla f_i(x_{i,\ell}).
\tag{144}
$$

Following exactly the same steps as in the proof of Lemma 10, we obtain

$$
\begin{aligned}
\|\tilde{x}_{t+1} - x^*\|^2 &\leq \|\tilde{x}_t - x^*\|^2 - 2\bar{\eta}\left(f(\tilde{x}_t) - f(x^*)\right) + \frac{L(1 + 2\bar{\eta}L)}{m} \sum_{i=1}^{m} \eta_i \sum_{\ell=0}^{\tau_i-1} \|x_{i,\ell} - \tilde{x}_t\|^2 \\
&\quad + 2\bar{\eta}^2 \|\nabla f(\tilde{x}_t)\|^2.
\end{aligned}
\tag{145}
$$

Just as in the proof of Theorem 8, our next task is to derive a bound on $\|x_{i,\ell} - \tilde{x}_t\|^2$. To this end, we start with

$$
\|\tilde{x}_t - x_{i,\ell}\|^2 \leq 2\bar{\eta}^2\left(\|e_{t-1}\|^2 + \|g_t\|^2\right).
\tag{146}
$$

To arrive at the above inequality, we used (143), and the fact that $\|\bar{x}_t - x_{i,\ell}\| \leq \bar{\eta}\|g_t\|$. Next, observe that

$$
\begin{aligned}
\|g_t\|^2 &= \|\mathcal{C}_{\delta_s}\left(e_{t-1} + \nabla f(\bar{x}_t)\right)\|^2 \\
&\leq \|e_{t-1} + \nabla f(\bar{x}_t)\|^2 \\
&\leq 2\|e_{t-1}\|^2 + 2\|\nabla f(\bar{x}_t)\|^2.
\end{aligned}
\tag{147}
$$

Using the smoothness of $\nabla f(\cdot)$, we also have

$$
\begin{aligned}
\|\nabla f(\bar{x}_t)\|^2 &= \|\nabla f(\bar{x}_t) - \nabla f(\tilde{x}_t) + \nabla f(\tilde{x}_t)\|^2 \\
&\leq 2\|\nabla f(\bar{x}_t) - \nabla f(\tilde{x}_t)\|^2 + 2\|\nabla f(\tilde{x}_t)\|^2 \\
&\leq 2L^2\|\bar{x}_t - \tilde{x}_t\|^2 + 2\|\nabla f(\tilde{x}_t)\|^2 \\
&= 2\bar{\eta}^2 L^2 \|e_{t-1}\|^2 + 2\|\nabla f(\tilde{x}_t)\|^2.
\end{aligned}
\tag{148}
$$

Suppose $\bar{\eta}$ is chosen such that $\bar{\eta}L \leq \frac{1}{2}$. Combining the bounds we have derived above, we can then obtain

$$
\|\tilde{x}_t - x_{i,\ell}\|^2 \leq 8\bar{\eta}^2\left(\|e_{t-1}\|^2 + \|\nabla f(\tilde{x}_t)\|^2\right).
\tag{149}
$$

Substituting the above bound in (145), and simplifying the resulting inequality leads to

$$
\|\tilde{x}_{t+1} - x^*\|^2 \leq \|\tilde{x}_t - x^*\|^2 - 2\bar{\eta}\left(f(\tilde{x}_t) - f(x^*)\right) + 14\bar{\eta}^2\|\nabla f(\tilde{x}_t)\|^2 + 24\bar{\eta}^3 L\|e_{t-1}\|^2.
\tag{150}
$$

Given the dependence of the above inequality on the gradient sparsification error $e_{t-1}$, we next proceed to derive a recursion for bounding $\|e_t\|$. We follow similar steps as in Lemma 17.

$$
\begin{aligned}
\|e_t\|^2 &= \|e_{t-1} + \nabla f(\bar{x}_t) - \mathcal{C}_{\delta_s}(e_{t-1} + \nabla f(\bar{x}_t))\|^2 \\
&\overset{(a)}{\leq} \left(1 - \frac{1}{\delta_s}\right) \|e_{t-1} + \nabla f(\bar{x}_t)\|^2 \\
&\overset{(b)}{\leq} \left(1 - \frac{1}{\delta_s}\right)(1+\gamma)\|e_{t-1}\|^2 + \left(1 - \frac{1}{\delta_s}\right)\left(1 + \frac{1}{\gamma}\right)\|\nabla f(\bar{x}_t)\|^2 \\
&\overset{(c)}{\leq} \left(1 - \frac{1}{2\delta_s}\right)\|e_{t-1}\|^2 + 2\delta_s\|\nabla f(\bar{x}_t)\|^2 \\
&\overset{(d)}{\leq} \left(1 - \frac{1}{2\delta_s} + 4\bar{\eta}^2 L^2 \delta_s\right)\|e_{t-1}\|^2 + 4\delta_s\|\nabla f(\tilde{x}_t)\|^2.
\end{aligned}
\tag{151}
$$

In the above steps, for (a), we used Property 3 of the TOP-k operator in Lemma 14; for (b), we used (35); for (c), we set $\gamma = \frac{1}{2(\delta_s - 1)}$; finally, for (d), we used the bound on $\|\nabla f(\bar{x}_t)\|^2$ in (148). We now have all the individual pieces required to complete the proof of Theorem 7. To proceed, let us define the following Lyapunov function:

$$
\psi_t \triangleq \|\tilde{x}_t - x^*\|^2 + \bar{\eta}^2 \|e_{t-1}\|^2.
$$

Referring to (150) and (151), using the fact that $f(\cdot)$ is $\mu$-strongly convex, and following similar arguments as in the proof of Theorem 8, we obtain

$$
\begin{aligned}
\psi_{t+1} &\leq \|\tilde{x}_t - x^*\|^2 - 2\bar{\eta}(1 - 18\bar{\eta}L\delta_s)(f(\tilde{x}_t) - f(x^*)) + \left(1 - \frac{1}{2\delta_s} + 28\bar{\eta}L\right)\bar{\eta}^2\|e_{t-1}\|^2 \\
&\leq \left(1 - \frac{3}{4}\bar{\eta}\mu\right)\|\tilde{x}_t - x^*\|^2 + \left(1 - \frac{1}{2\delta_s} + 28\bar{\eta}L\right)\bar{\eta}^2\|e_{t-1}\|^2 \\
&\leq \left(1 - \frac{3}{4}\bar{\eta}\mu\right)\left(\|\tilde{x}_t - x^*\|^2 + \bar{\eta}^2\|e_{t-1}\|^2\right) \\
&= \left(1 - \frac{1}{96\delta_s\kappa}\right)\psi_t.
\end{aligned}
\tag{152}
$$

In the last two steps, we used $\bar{\eta} = \frac{1}{72L\delta_s}$, implying $\eta_i = \frac{1}{72L\delta_s\tau_i}$. The rest of the proof follows by recursively using the above inequality in conjunction with the following easily verifiable facts:

$$
\psi_1 = \|\bar{x}_1 - x^*\|^2 \quad \text{since } e_0 = 0; \quad \|\bar{x}_{T+1} - x^*\|^2 \leq 2\psi_{T+1}.
$$

## J Simulation Results for `FedSplit`

In [10], the authors introduce `FedSplit` - an algorithmic framework based on operator-splitting procedures. Given an initial global model $\bar{x}_1$, the update rule of `FedSplit` is given by

$$
\begin{aligned}
y_i^{(t)} &= \texttt{prox\_update}_i(2\bar{x}_t - z_i^{(t)}), \\
z_i^{(t+1)} &= z_i^{(t)} + 2(y_i^{(t)} - \bar{x}_t), \\
\bar{x}_{t+1} &= \frac{1}{m} \sum_{i \in \mathcal{S}} z_i^{(t+1)},
\end{aligned}
\tag{153}
$$

where $z_i^{(1)} = \bar{x}_1$, $i \in \mathcal{S}$. The local update at client $i$ is defined in terms of a proximal solver $\texttt{prox\_update}_i(\cdot)$. Ideally, this proximal solver would be an exact evaluation of the following proximal operator for some step-size $s > 0$:

$$
\mathbf{prox}_{sf_i}(u) := \underset{x \in \mathbb{R}^d}{\operatorname{argmin}} \left\{ \underbrace{f_i(x) + \frac{1}{2s} \|u - x\|^2}_{h_i(x)} \right\}.
\tag{154}
$$

As suggested in [10], in practice, `FedSplit` would be implemented using an approximate proximal solver. One way to do so, as clearly detailed in [10], is to run $e$ steps of gradient descent on $h_i(x)$ using a suitably chosen step-size $\alpha$. The latter is precisely the method we use to numerically implement `FedSplit`. As per Corollary 1 in [10], `FedSplit` achieves linear convergence to a neighborhood of the global minimum for any value of $e$. In what follows, we show that `FedSplit` may diverge even under the simplest of settings. In particular, we consider an instance of problem (1) where two clients with simple quadratics attempt to minimize the global objective function (1) using `FedSplit`. The local objective function of client 1 is given by

$$
f_1(x) = \frac{1}{2} x^T \underbrace{\begin{bmatrix} L & 0 \\ 0 & \mu \end{bmatrix}}_{A_1} x - \underbrace{\begin{bmatrix} 1 \\ 1 \end{bmatrix}^T}_{B_1^T} x,
$$

and the local objective function of client 2 is given by

$$
f_2(x) = \frac{1}{2} x^T \underbrace{\begin{bmatrix} \mu & 0 \\ 0 & \mu \end{bmatrix}}_{A_2} x - \underbrace{\begin{bmatrix} -1 \\ 2 \end{bmatrix}^T}_{B_2^T} x,
$$

where $L = 1000$, and $\mu = 1$. The step-sizes corresponding to the proximal operator and gradient descent, namely $s$ and $\alpha$, respectively, are chosen as per Corollary 1 in [10]. Furthermore, we run $e$ rounds of gradient descent per communication round $t$ for $e \in \{1, \cdots, 41\}$. Given the fact that the implementation code of `FedSplit` is not publicly available, we note that our local implementation of the scheme has diverged for all odd values of $e$ between 1 and 41, inclusive. It should be noted, however, that our implementation of `FedSplit` converged for some even values of $e$ in the considered range, as shown in Figure 5. We have further observed that increasing the ratio $\kappa = L/\mu$ beyond 1000 causes `FedSplit` to diverge for values of $e$ higher than 41 as well.

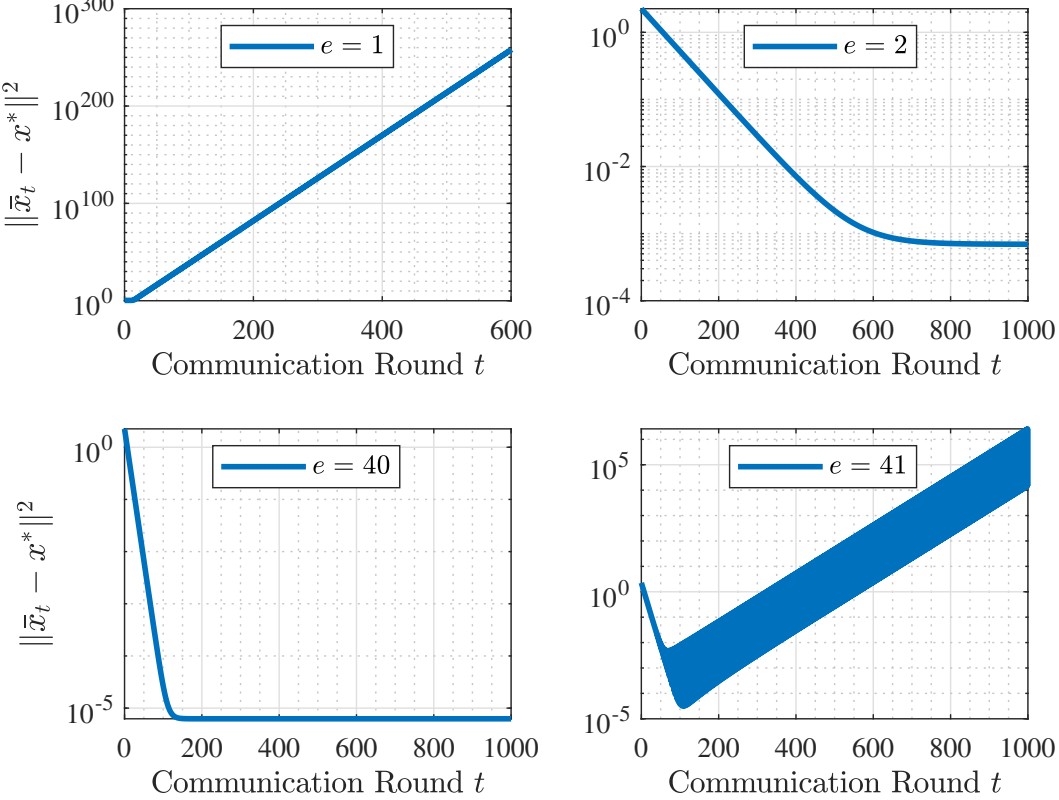

Figure 5: Simulation results for `FedSplit` for $e \in \{1, 2, 40, 41\}$.

# K   Additional Experimental Results on Logistic Regression

All the presented simulations are performed on a machine running Windows 8.1 with a 1.8 GHz Intel Core i7 processor, using `MATLAB` R2019. In this section, we provide additional numerical results for `FedLin` on a logistic regression problem. For each client $i \in \mathcal{S}$, the design matrix $A_i \in \mathbb{R}^{m_i \times d}$ is a collection of $m_i$ feature vectors, with the $j$-th feature vector denoted by $a_{ji}$, $j \in \{1, \cdots, m_i\}$. In turn, every feature vector $a_{ji}$ is associated with a class label $b_{ji} \in \{+1, -1\}$. In a logistic regression problem, the conditional probability of observing a positive class label $b_{ji} = +1$ for a given feature vector $a_{ji}$ is

$$\mathcal{P}\big(b_{ji} = +1\big) = \frac{1}{1 + e^{-a_{ji}^T x}}, \tag{155}$$

where $x \in \mathbb{R}^d$ is an unknown parameter vector to be estimated. The maximum likelihood estimate of the parameter vector $x$ is then the solution of the following convex optimization problem

$$\min_{x \in \mathbb{R}^d} f(x) = \min_{x \in \mathbb{R}^d} \frac{1}{m} \sum_{i=1}^{m} \underbrace{\sum_{j=1}^{m_i} \log(1 + e^{-b_{ji} a_{ji}^T x})}_{f_i(x)}. \tag{156}$$

The client objective functions, $f_i(x)$, are both smooth and convex. To generate synthetic data, for each client $i \in \mathcal{S} = \{1, \cdots, 10\}$, we generate the design matrix $A_i$ and the corresponding class labels according to the model (155), where $x \in \mathbb{R}^{100}$ and $A_i \in \mathbb{R}^{500 \times 100}$. In particular, the entries of the design matrix are modeled as $[A_i]_{jk} \overset{i.i.d.}{\sim} \mathcal{N}(0, 1)$ for $j \in \{1, \cdots, 500\}$ and $k \in \{1, \cdots, 100\}$. The entries of the true parameter vector $x$ are modeled as $[x]_\ell \overset{i.i.d.}{\sim} \mathcal{N}(0, 1)$ for $\ell \in \{1, \cdots, 100\}$. To model the effect of systems heterogeneity, we allow clients to perform different numbers of local iterations. In particular, for each client $i \in \mathcal{S}$, the number of local iterations is drawn independently from a uniform distribution over the range $[2, 50]$, i.e, $\tau_i \in [2, 50]$, $\forall i \in \mathcal{S}$.

**Gradient Sparsification at Server.** We first consider a variant of `FedLin` where gradient sparsification is implemented only at the server side without any error-feedback. In particular, we consider the cases where $\delta_s \in \{2, 4\}$, which correspond to the implementation of a `TOP-50` and a `TOP-25` operator on the communicated gradients, respectively. For comparison, we also plot the resulting performance when no gradient sparsification is implemented at the server side, i.e. $\delta_s = 1$. As illustrated in Figure 6, regardless of the level of gradient sparsification at the server side, `FedLin` always converges to the true minimizer.

**Gradient Sparsification at Clients.** Next, we implement gradient sparsification only at the clients' side, i.e. $\delta_s = 1$. In particular, we consider the cases where $\delta_c \in \{1.25, 1.67\}$, which correspond to the implementation of a `TOP-80` and a `TOP-60` operator on the communicated local gradients, respectively. As illustrated in Figure 7, unlike the server case, `FedLin` with sparsification at the clients' side converges with a non-vanishing convergence error, which increases as the value of $\delta_c$ increases.

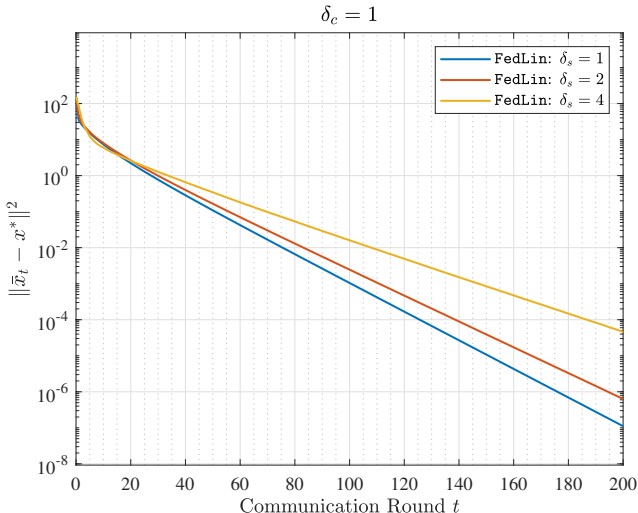

Figure 6: Simulation results for `FedLin` where gradient sparsification is implemented at the server side. The constant $\bar{\eta}$ is fixed at $0.15$ across all clients.

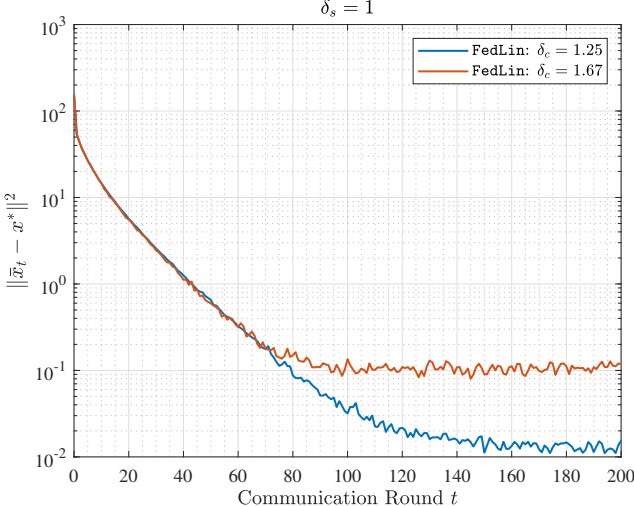

Figure 7: Simulation results for `FedLin` where gradient sparsification is implemented at the clients' side. The constant $\bar{\eta}$ is fixed at $0.1$ across all clients.

# L  Simulation Results for `FedLin` with Noisy Gradients

In this section, we provide numerical results for `FedLin` under noisy client gradients to validate the theoretical results of Theorem 4. In particular, we consider the least square problem of Section 7 with $\delta_s = \delta_c = 1$ and $\alpha = 10$. All the remaining parameters are kept the same. To simulate noisy gradients, we add zero-mean Gaussian noise with variance $\sigma^2 \in \{10^{-5}, 10^{-3}, 10^{-1}\}$.

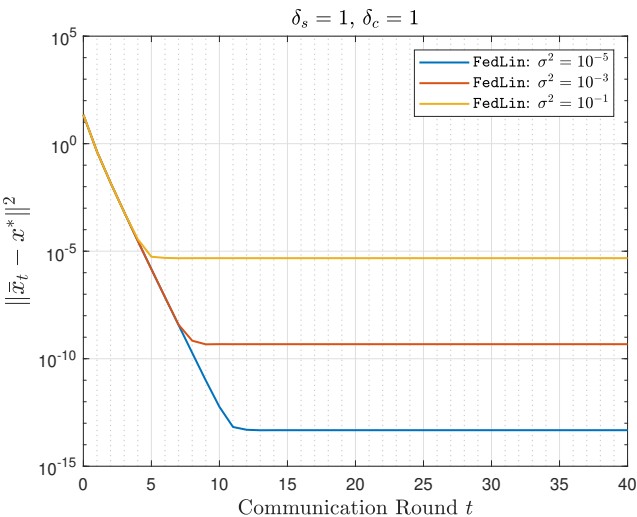

Figure 8: Simulation results for `FedLin` under noisy client gradients. The constant $\bar{\eta}$ is fixed at $10^{-2}$ across all clients.

As illustrated in Figure 8, `FedLin` under noisy gradients converges with a non-vanishing error-floor, which increases as the variance of the noise increases. Thus, our simulations here corroborate the theory in Theorem 4.