# OpenReview forum: "Linear Convergence in Federated Learning: Tackling Client Heterogeneity and Sparse Gradients"
_NeurIPS.cc/2021/Conference — NeurIPS 2021 Poster_

### Official Review · Reviewer_LmKH · 2021-07-07

**Rating:** 5
**Confidence:** 4

**Summary:**

This paper considers a standard distributed optimization setup in which clients periodically coordinate with a server to train a global model. The focus is on the deterministic setting, in which each worker computes full local gradients. It is first highlighted that standard algorithms for this problem do not converge to the true global minimizer without trading off speed (for example through diminishing step-sizes).

Thus, the FedLin algorithm, which converges linearly to the true minimizer, is introduced to get the best of both worlds. Several theorems covering various settings (strongly convex, convex, non-convex, stochastic) are given, and show that FedLin matches the communication complexity of parallel GD, regardless of the number of local steps used. Then, a matching lower bound is given for FedLin to show that the strongly convex convergence result is tight.

Then, several gradient sparsification schemes are investigated (at the server and at the workers), which show that server-side sparsification is far less impactful than client-side sparsification. The impact of gradient sparsification is evaluated experimentally in Section 7.

**Limitations And Societal Impact:**

No foreseeable societal impact.

**Main Review:**

The fact that standard FL algorithms do not converge to the true minimizer is good to recall, but it's quite well-known so maybe too much space is spent on this. Otherwise, the algorithm is quite natural, and achieves its objective of providing linear convergence to the true minimizer in the deterministic setting. This being said, I have two major concerns, that I develop below.

1) Convergence rates:
I have very mixed feelings about Theorems 1-4 and the beginning of Section 4.1. It is nice to know that we can do local steps, but they do not seem to improve anything in this deterministic setting. Indeed, taking only one local step each time is enough. The step-size is divided by the number of local steps, and so workers end up doing a step of the same size as if they were only doing one step, with a similar result apparently. This only increases the computational cost, with no benefits.

I believe that it would be great to highlight at least one setting in which this is not the case. Something like "it cannot be better in the general setting but in some cases you really gain things", because after reading Section 4 I don't see why I would use FedLin over parallel GD. It seems like FedLin was proposed and analyzed for the sake of performing local steps, instead of actually bringing something to the table.

Besides, there is a matching lower bound, but it is only matching for the FedLin algorithm so it does not imply that other algorithms of this kind cannot do better. I know it is hard to use a general lower bound because it would require acceleration to match it, but if you consider the lower bound from [A], you obtain that you need at least $O(\sqrt{\kappa})$ communications to solve the problem (adapt the proof, or argue that you consider a graph of diameter 1 so time $\tau \sqrt{\kappa}$, which means $\sqrt{\kappa}$ communications). "Translated" into a non-accelerated algorithm, this means that any algorithms requires time at least $\kappa$ communications, and so clearly there is no hope to beat parallel-GD in this setting without further assumptions (e.g., on the local functions to allow for more parallelism). This kind of reasoning seems more convincing to me than Theorem 5 since it is not restricted to FedLin.


2) Compression:
Why compress the gradients but not the parameters? They are the same dimension, and so it seems that the complicated compression scheme can only gain a factor 2 in general since full dimensional parameters will need to be sent anyway. Besides, both the server and the workers need to send full parameters so there is no upstream/downstream argument to be made on this part either. I have a hard time convincing myself of the interest of the compression scheme since it seems to affect only half of the actual communications.

[A] Scaman, Kevin, et al. "Optimal algorithms for smooth and strongly convex distributed optimization in networks." international conference on machine learning. PMLR, 2017.


=====================================

After the response and discussing with the other reviewers, I recognize that this paper advances the understanding of FL algorithms and thus raised my score accordingly.

**Time Spent Reviewing:**

2

---

> ### Author Response · Authors · 2021-08-10
> **Response to Reviewer LmKH**
>
> We thank the Reviewer for their detailed and valuable feedback on our work. Below, we discuss each of the Reviewer's concerns and explain how we plan to address them in the revised version of our manuscript.
>
> Reviewer: The fact that standard FL algorithms do not converge to the true minimizer is good to recall, but it's quite well-known so maybe too much space is spent on this. Otherwise, the algorithm is quite natural, and achieves its objective of providing linear convergence to the true minimizer in the deterministic setting.
>
> Response: The Reviewer's recommendation is well taken. In the revised manuscript, we will be less elaborate while commenting on known facts about popular FL algorithms.  That being said, we would like to mention that as far as we are aware, ours is the first work to point out that very recently proposed algorithms like FedNova (Wang et al., NeuRIPS 2020) and FedSplit (Pathak et al., NeuRIPS 2020) also fail to converge linearly to the correct minimum. As such, we believe that these observations are quite valuable and informative.
>
> Reviewer: Convergence rates: I have very mixed feelings about Theorems 1-4 and the beginning of Section 4.1. It is nice to know that we can do local steps, but they do not seem to improve anything in this deterministic setting. Indeed, taking only one local step each time is enough. The step-size is divided by the number of local steps, and so workers end up doing a step of the same size as if they were only doing one step, with a similar result apparently. This only increases the computational cost, with no benefits. I believe that it would be great to highlight at least one setting in which this is not the case. Something like "it cannot be better in the general setting but in some cases you really gain things", because after reading Section 4 I don't see why I would use FedLin over parallel GD. It seems like FedLin was proposed and analyzed for the sake of performing local steps, instead of actually bringing something to the table.
>
> Response: We understand the Reviewer's concern. In this context, we would like to first highlight that the rate we obtain in Theorem 1 is the *best* linear convergence rate we are aware of among all FL algorithms, when considering *arbitrary* objective heterogeneity. Indeed, competitive algorithms like Scaffold and FedSplit exhibit the same phenomenon: the exponent of convergence decays with T, as opposed to TH, where H is the number of local steps. Instead of hiding this fact, we have stated it explicitly in Section 4.1, where we provide our lower-bound result. Notably, the popular Scaffold algorithm also employs a learning rate that scales inversely with H; however, no lower bound analysis is provided in that paper.
>
> Given our result in Theorem 5, and the fact that no FL algorithm (including ours) has been able to establish tangible benefits of performing multiple local steps, we conjecture that the issue lies not in the algorithms themselves, but rather in the *arbitrary* data heterogeneity assumption. Under more relaxed assumptions on heterogeneity, it is quite possible that FedLin (or variants thereof) may demonstrate benefits of performing multiple local steps - this is a subject of our ongoing work. Having said that, we would like to reiterate that the primary focus of our paper is to develop a unified algorithmic framework that simultaneously tackles some of the key challenges in FL: objective heterogeneity, systems heterogeneity, and compression. In this context, our analysis sheds light on when one can achieve linear convergence despite the above challenges. In particular, since we wanted our results to be as general as possible, we did not place any assumptions on the level of heterogeneity.
>
> Thus, to sum up, we would like the Reviewer to take into consideration the following facts: (i) that local steps do not have any tangible benefits is a shared limitation among all FL algorithms we are aware of, and is mainly a consequence of the *arbitrary* data heterogeneity assumption; (ii) unlike the papers that propose competitive algorithms like Scaffold and FedSplit, our work provides a tight lower bound analysis that highlights why and how heterogeneity can hurt; and (iii) we believe that the insights obtained from Theorem 5 are valuable towards identifying algorithms and problem instances where local steps can help despite heterogeneity; no other FL paper we are aware of provides a similar tight linear convergence rate analysis.
>
> Response to Reviewer's comment on lower bound: First of all, we thank the Reviewer for the pointer to Reference [A]. We will take a closer look at [A] and attempt to craft an alternate lower-bound argument along the lines suggested by the Reviewer. That being said, we would like the Reviewer to take into consideration the following points.
>
> (i) Even algorithmic-specific lower bounds are rare in FL. As we have mentioned above, despite the flurry of papers on this topic, our work seems to be the only one to provide a tight linear convergence rate analysis.
>
> (ii) While we agree that there may be alternate (more elegant) ways of establishing an algorithm-independent lower bound, we believe that our analysis in Theorem 5 is valuable and quite informative, as also noted by Reviewers Xwfq and 2DHS. The main reason for this is the simplicity of the construction and its corresponding implications. Gradient-tracking (FedLin) and variance-reduction (Scaffold) are two very popular techniques to deal with data heterogeneity in machine learning and optimization communities. The benefit of our lower-bound analysis being algorithm-specific is that it reveals the limitations of such specific techniques (in the context of FL) by showing that even for very simply constructed non-identical quadratic loss functions, the lower bound in Theorem 5 continues to hold. Perhaps the most surprising takeaway from this analysis is that these quadratics may even share the same minimum.
>
> Another key point to note is the following: the crux of our lower bound argument is that the step-size (learning rate) needs to necessarily scale inversely with the number of local steps to control the client-drift. This helps to shed light upon the choice of the learning rate for both our algorithm and Scaffold; we are unaware of any other concrete theoretical evidence that substantiates such a choice of learning rate. In contrast, while the argument sketched by the Reviewer is by no doubt valuable and insightful, it does not seem to be directly tied to FL-specific issues such as local steps and the client-drift phenomenon.
>
> A general algorithm-independent lower bound analysis on a complex instance would potentially obscure the above key takeaways. In contrast, we believe our simple,  direct analysis will make it more accessible for a reader to gain intuition regarding why stale gradient correction terms (FedLin) or control variates (Scaffold) may hurt sometimes. Moreover, in our humble opinion, a careful understanding of such a phenomenon will likely lead to the design of more informed local updating schemes.
>
> Reviewer: Why compress the gradients but not the parameters? They are the same dimension, and so it seems that the complicated compression scheme can only gain a factor 2 in general since full-dimensional parameters will need to be sent anyway.
>
> Response: The Reviewer is right that our current approach does not consider compression of model parameters,  Reviewer wkH7 raises a similar concern. That being said, we believe that the techniques developed in this paper can be extended in a fairly reasonable way to study model compression in tandem with gradient sparsification. The reason why we chose to focus only on the latter is twofold.
>
> First, there is no prior work that studies biased compression in FL. Given the lack of prior work on this topic, we wanted to isolate the key challenges associated with analyzing biased compression in a federated setting involving local steps, data heterogeneity, and systems heterogeneity. As we discuss in Remark 1, such an analysis is non-trivial and requires new potential-function-based proofs that we develop in this paper. Despite these challenges, we provide a comprehensive analysis of the impacts of both up-link and down-link sparsification that sheds light upon the differences between these mechanisms.
>
> Second, we would like to point out that compression is just one of the topics we study in this paper. Our work covers other aspects as well such as motivating and developing our algorithm, studying the setting without compression, deriving lower bounds to analyze the tightness of our algorithm, etc. Thus, it seemed unlikely that we would be able to do justice to both model compression and gradient sparsification in this one paper, given its other content.
>
> To sum up, studying compression of model parameters is indeed a ripe avenue for future research (as identified by the Reviewer). We believe our work lays the necessary foundation to study such a topic and takes a significant step in that direction.  We will add a discussion on this matter to the revised paper.
>
> For the Reviewer's convenience, we summarize our responses to the main comments.
>
> - Convergence rates in Theorems 1-4: They are the best rates we know of under *arbitrary* heterogeneity.
>
> -  Algorithm-specific lower bound analysis: Our analysis provides valuable insights regarding specific popular techniques such as gradient-tracking and variance reduction. The simplicity of the constructed instance also has implications as we discussed above. Moreover, no other FL paper provides a tight linear rate analysis.
>
> - Compression: Our work is the first to perform a principled theoretical analysis of biased compression in FL, despite the challenges it poses. In doing so, we believe it provides the necessary intuition and foundation to study compression of model parameters (in tandem with gradient-sparsification) as future work.

---

### Official Review · Reviewer_wkH7 · 2021-07-14

**Rating:** 8
**Confidence:** 3

**Summary:**

This paper proposes and analyzes a framework for Federated optimization called FedLin. It combines client-specific local steps with (possibly) biased compression at the server and/or clients and guarantees linear convergence to the exact minimum. Algorithmic contributions are supported with lower bounds for the case of infrequent communication and carefully designed synthetic experiments on least squares and logistic regression problems, which highlight theoretical claims.

**Limitations And Societal Impact:**

Limitations of the work were mostly addressed. Negative societal impact concerns are not applicable due to the theoretical nature of the paper.

**Main Review:**

This paper is a solid piece of mainly theoretical work on the problem of linear convergence in the Federated optimization setting, and I recommend acceptance.

The paper is clearly written and well-structured. Related work is mostly adequately cited but can be complemented with the latest works on local methods with compression. I think that a much more detailed discussion is needed to compare with the briefly mentioned work [1]. I understand that it appeared quite recently, but it is essential for publication as it also suggested methods linearly converging to the exact optimum and combined with variance-reduction. It would also be helpful to include a comparison table with differences to the previous methods (e.g., with uncompressed methods like FedSplit, FedNova, SCAFFOLD, S-Local-(S)GD/SVRG from [1]) to make it easier for readers to understand the contributions.

The authors present only synthetic experiments, but they are appropriate in this case as they were carefully designed and decently support the theoretical claims. However, I recommend including real-data experiments (e.g., with LibSVM datasets) in the final version as it will help to understand the actual practical performance and limitations of the presented approach.

The main drawback of the proposed algorithm FedLin is that it requires sending uncompressed model vectors $x_{i, \tau_{i}}^{(t)}$ from workers to the server and $\overline{x}_{t+1}$ back at every communication round, which may not be efficient/possible in the federated learning setting. That is why the part on the compression seems is not very valuable from the practical side.

## Post-rebuttal
I have read other reviews and authors' answers. I found both of them very insightful and believe that the published paper will be significantly improved with the proposed suggestions in mind. To conclude, I keep my score as it is.

***

[1] E. Gorbunov, et al. “Local sgd: Unified theory and new efficient methods”. In International Conference on Artificial Intelligence and Statistics, pages 486 3556–3564. PMLR, 2021.

**Time Spent Reviewing:**

8

---

> ### Author Response · Authors · 2021-08-10
> **Response to Reviewer wkH7**
>
> We thank the Reviewer for their detailed and valuable feedback on our work. Below, we discuss each of the Reviewer's
> concerns, and explain how we plan to address them in the revised version of our manuscript.
>
> Reviewer: The paper is clearly written and well-structured. Related work is mostly adequately cited but can be complemented with the latest works on local methods with compression. I think that a much more detailed discussion is needed to compare with the briefly mentioned work [1]. I understand that it appeared quite recently, but it is essential for publication as it also suggested methods linearly converging to the exact optimum and combined with variance-reduction. It would also be helpful to include a comparison table with differences to the previous methods (e.g., with uncompressed methods like FedSplit, FedNova, SCAFFOLD, S-Local-(S)GD/SVRG from [1]) to make it easier for readers to understand the contributions.
>
> Response: We thank the Reviewer for their encouraging comments. We tried to accumulate all the FL methods we know of that employ compression in Appendix A. We would be grateful if the Reviewer could point us to any references that we have missed. We would be happy to cite and discuss them in the revised manuscript.
>
> Ref [1] is a concurrent work that we became aware of while preparing our own manuscript. We will make sure to elaborate on the key differences between our work and [1] in the revised paper. As far as we understand, [1] explores the specific setting where the loss function at each client can be expressed as a finite-sum. For this setting, the authors study how local variance reduction methods can help achieve linear convergence. The key differences of our work with [1] are as follows: (i) we do not assume a finite-sum structure, and as such, our algorithm differs from S-Local-SVRG; (ii) our approach can handle systems heterogeneity and biased compression while [1] does not consider such issues; and (iii) we provide tight lower bounds for our method whereas [1] does not provide any lower bounds for S-Local-SVRG.
>
> Including a table to highlight the key differences with prominent existing algorithms is indeed a great suggestion. Based on the Reviewer's recommendation, we will include such a table in our revised paper.
>
> Reviewer: The authors present only synthetic experiments, but they are appropriate in this case as they were carefully designed and decently support the theoretical claims. However, I recommend including real-data experiments (e.g., with LibSVM datasets) in the final version as it will help to understand the actual practical performance and limitations of the presented approach.
>
> Response: We appreciate the Reviewer's feedback and will include experiments on real data in our updated manuscript.
>
> Reviewer wkH7: The main drawback of the proposed algorithm FedLin is that it requires sending uncompressed model vectors $x^{(t)}__{i,\tau_i}$ from workers to the server and $\bar{x}_{t+1}$ back at every communication round, which may not be efficient/possible in the federated learning setting.  That is why the part on the compression seems is not very valuable from the practical side.
>
> Response: The Reviewer rightly points out that we do not consider compression of model parameters; Reviewer LmKH raises a similar concern. That being said, we believe that the techniques developed in this paper can be extended in a fairly natural way to study model compression in tandem with gradient sparsification. The reason why we chose to focus only on the latter is twofold.
>
> First, there is no prior work that studies biased compression in FL. Given the lack of prior work on this topic, we wanted to isolate the key challenges associated with analyzing biased compression in a federated setting involving local steps, data heterogeneity, and systems heterogeneity. As we discuss in Remark 1, such an analysis is non-trivial and requires new potential-function-based proofs that we develop in this paper. Despite these challenges, we provide a comprehensive analysis of the impacts of both up-link and down-link sparsification that sheds light upon the differences between these mechanisms. We would also like to note here that none of the popular
> algorithms like FedAvg, FedProx, Scaffold, FedSplit, and FedNova consider compression (neither of the model vectors nor of the gradients).
>
> Second, we would like to point out that compression is just one of the topics we study in this paper. Our work covers other aspects as well such as motivating and developing our algorithm, studying the setting without compression, deriving lower bounds to analyze the tightness of our algorithm, etc. Thus, it seemed unlikely that we would be able to do justice to both model compression and gradient sparsification in this one paper, given its other content.
>
> To sum up, studying compression of model parameters is indeed a ripe avenue for future research (as identified by the Reviewer). We believe our work lays the necessary foundation to study such a topic and takes a significant step in that direction.

---

### Official Review · Reviewer_2DHS · 2021-07-15

**Rating:** 8
**Confidence:** 4

**Summary:**

This paper proposes an optimization method for federated learning, FedLin, that is specifically designed to ensure fast convergence even in the presence of heterogeneous data. Notably, the authors show that FedLin converges to the global optimum (or just a critical point, in the case of non-convex functions) without decaying the client learning rate. This is in contrast to methods such as FedAvg, FedProx, and FedNova, which require decaying the client loss in order to circumvent "objective mismatch", where the methods are actually optimizing an altered surrogate loss.

This work also derives lower bounds matching the convergence of FedLin on quadratics, that explain the price and cost of performing multiple local update steps. They also analyze convergence using gradient sparsification at both the client and server. Finally, the paper shows the empirical benefits of FedLin over other methods on synthetic least squares and logistic regression problems.

**Ethical Concerns:**

None.

**Limitations And Societal Impact:**

The authors have some discussion of limitations of the work in Section 4. However, I would argue that the primary limitation is not addressed, and should at least be discussed. Currently, FedLin effectively requires a full participation among clients. It also requires a kind of two-stage communication process in each round, where clients train, communicate with the server, and then compute a full gradient over their data. It is worth noting that this does increase the system-level complexity of FedLin as compared to methods such as FedAvg. I do not think that contending with partial participation need be in the scope of this work (as the full participation case aligns well with cross-silo federated learning), but I do believe this limitation should be mentioned.

**Main Review:**

I think this paper is a well-written paper with a clear focal idea, and a suite of great results to back up this idea. More specifically, I really enjoy that this paper essentially shows that certain convergence-accuracy trade-offs in federated learning (that exist for methods such as FedAvg, FedProx) can actually be circumvented in certain settings. The convergence results for FedLin are impressive, and **simple**. This is actually something wish to highlight. Client heterogeneity has caused a preponderance of assumptions needed to analyze the convergence of FL methods (eg. bounded gradient dissimilarity, specific learning rate schedules, etc.). This paper does away with all of this, and simply assumes smoothness. The results have no extra "problem-dependent" terms that are shrouded in asymptotics. The fact that the authors can also incorporate sparsification relatively easily into the framework is only an additional benefit.

I also want to highlight the fact that the authors attempted to ensure that their results were as tight as possible via Section 4, something that can be challenging in federated learning. The takeaways from Theorem 5 are extremely informative, and give a succinct summary of the cost and gain of "local training steps" in federated learning. My only comment here is a question for future work: Can a similar lower bound be derived for a class of federated optimization algorithms that generalizes FedLin and other related methods (eg. SCAFFOLD, FedAvg)?

The experimental results are somewhat brief, and focus on synthetic data. I do not think this is a downside of the paper. Many works have noted that convergence failures of FedAvg can be shown in synthetic quadratic settings with just 2 clients. Moreover, this work is fundamentally a theoretical one (and a strong one at that), and I do not subscribe to the notion that all papers must offer all possible theoretical results and experimental results. I will note that even from an empirical side, the experiments that the authors do in Appendix J evincing the non-convergence of FedSplit in certain settings are also extremely valuable. One suggestion for improving these results would be to have a version of Figure 2 that compares FedLin to FedAvg, FedProx, SCAFFOLD, FedSplit, etc.

One contention I have with the work is that it ignores a number of works that first proposed and analyzed the speed-accuracy trade-off discussed in the abstract and in Section 2. In particular, it is worth noting that works such as (Malinovsky et al., 2020; Charles and Konecny, 2020; Fallah et al, 2020, and Pathak and Wainwright, 2020) showed that various federated learning and meta-learning methods (including FedAvg, FedProx, and FOMAML) are actually optimizing surrogate objectives. (Charles and Konecny, 2021) also formalized the actual underlying trade-off (as Propositions 1 & 2 don't actually show a trade-off they simply show non-optimality of the methods). Moreover, Propositions 1 and 2 are special cases of Theorem 1 of (Charles and Konecny, 2021).

This is all to say that I think that the paper would be improved by 1) giving proper credit to the works that developed the idea of speed-accuracy trade-offs and 2) focusing on what common facets of methods such as FedProx and FedNova you are going to change in order to break that trade-off. More succinctly, breaking these trade-offs is the novel, exciting contribution of this work, not showing that the trade-offs exists.

$\\\\$

# Review Update

After reading the other reviews and the author feedback, I have kept my score as is. In particular, I think the convergence rates are still useful, if only as a benchmark against other methods (such as FedAvg) which do not converge globally, even in deterministic settings, unless parameters such as learning rate or number of steps is decayed so as to essentially degenerate to FedSGD.

**Time Spent Reviewing:**

4

---

> ### Author Response · Authors · 2021-08-10
> **Response to Reviewer 2DHS**
>
> We thank the Reviewer for their detailed and valuable feedback on our work. Below, we discuss each of the Reviewer's
> concerns, and explain how we plan to address them in the revised version of our manuscript.
>
> Reviewer: I also want to highlight the fact that the authors attempted to ensure that their results were as tight as possible via Section 4, something that can be challenging in federated learning. The takeaways from Theorem 5 are extremely informative, and give a succinct summary of the cost and gain of "local training steps" in federated learning. My only comment here is a question for future work: Can a similar lower bound be derived for a class of federated optimization algorithms that generalizes FedLin and other related methods (eg. SCAFFOLD, FedAvg)?
>
> Response: First of all, we would like to thank the Reviewer for their encouraging comments. The question regarding the lower bound in Theorem 5 and its implications for Scaffold is indeed an important one; we note that a similar query was made by Reviewer Xwfq. As we explain to Reviewer Xwfq as well, the primary reason why the number of local steps $H$ do not show up in the lower bound is because of the staleness of the gradient correction term. Since this term is based on quantities computed at the beginning of a given round, as one performs more local steps within the round, the gradient correction term becomes stale and starts to hurt more than it helps. As we discuss in lines 177-188 of our paper, the control variates in Scaffold are staler than the gradient correction term in FedLin, owing to
> which Scaffold does not, in general, satisfy a desirable fixed point property that FedLin does. Thus, at least intuitively, we believe that any detrimental effect of staleness that manifests itself in the convergence rate of FedLin will have similar implications for Scaffold. We will add a discussion along these lines to the revised manuscript, and attempt to prove a result that formalizes this intuition in the appendix.
>
> Whether such an approach would carry over to algorithms that generalize FedLin is something we are not sure of; presumably, this would depend on the structure of the algorithm. A couple of related comments are as follows. The primary focus of the lower bound in Theorem 5 is the *rate* of linear convergence. Thus, it would make the most sense to use a similar technique for algorithms that are known to linearly converge. It may potentially also be applicable in the context of showing that FedSplit diverges on certain instances. For algorithms such as FedAvg that do not converge linearly to the correct minimum, the focus of the lower bound is the *bias* or *objective mismatch* term. Both the Scaffold paper and the work by Charles and Konecny provide really nice analyses on such *bias* effects.
>
> Reviewer: The experimental results are somewhat brief, and focus on synthetic data. I do not think this is a downside of the paper. Many works have noted that convergence failures of FedAvg can be shown in synthetic quadratic settings with just 2 clients. Moreover, this work is fundamentally a theoretical one (and a strong one at that), and I do not subscribe to the notion that all papers must offer all possible theoretical results and experimental results. I will note that even from an empirical side, the experiments that the authors do in Appendix J evincing the non-convergence of FedSplit in certain settings are also extremely valuable. One suggestion for improving these results would be to have a version of Figure 2 that compares FedLin to FedAvg, FedProx, SCAFFOLD, FedSplit, etc.
>
> Response: We thank the Reviewer for their comments. Based on this Reviewer's suggestions, and also the ones made by Reviewer Xwfq, we will add more experiments where we compare the performance of FedLin to FedAvg, FedProx, Scaffold, and FedSplit (in the absence of systems heterogeneity and compression).
>
> Reviewer: One contention I have with the work is that it ignores a number of works that first proposed and analyzed the speed-accuracy trade-off discussed in the abstract and in Section 2. In particular, it is worth noting that works such as (Malinovsky et al., 2020; Charles and Konecny, 2020; Fallah et al, 2020, and Pathak and Wainwright, 2020)  showed that various federated learning and meta-learning methods (including FedAvg, FedProx, and FOMAML) are actually optimizing surrogate objectives. (Charles and Konecny, 2021)
> also formalized the actual underlying trade-off (as Propositions 1 & 2 don't actually show a trade-off they simply show non-optimality of the methods). Moreover, Propositions 1 and 2 are special cases of Theorem 1 of (Charles and Konecny, 2021).
>
> Response: Firstly, we would like to thank the Reviewer for bringing the work (Charles and Konecny, 2021) to our attention. We were not aware of it at the time of our submission. Since Propositions 1 and 2 are indeed special cases of Theorem 1 in (Charles and Konecny, 2021), we will definitely point this out explicitly in the main body of our revised paper. In our revised manuscript, we will also make sure that we give due credit to each of the works the Reviewer has referred to above.
>
> Reviewer: The authors have some discussion of limitations of the work in Section 4. However, I would argue that the primary limitation is not addressed, and should at least be discussed. Currently, FedLin effectively requires a full participation among clients. It also requires a kind of two-stage communication process in each round, where clients train, communicate with the server, and then compute a full gradient over their data. It is worth noting that this does increase the system-level complexity of FedLin as compared to methods such as FedAvg. I do not think that contending with partial participation need be in the scope of this work (as the full participation case aligns well with cross-silo federated learning), but I do believe this limitation should be mentioned.
>
> Response: We concur with the Reviewer's assessment. In our discussion of limitations, we will specify in the revised manuscript that (i) FedLin essentially requires two passes of communication between the clients and the server as opposed to algorithms like FedAvg that require only one pass; and (ii) assumes full client participation. Although we believe that item (ii) can be resolved by adapting the existing partial client participation analyses to our setting, since we do not provide such an analysis, we agree that this limitation needs to be pointed out.

---

### Official Review · Reviewer_Xwfq · 2021-07-26

**Rating:** 7
**Confidence:** 4

**Summary:**

This paper proposes a new algorithm called *FedLin* which,

T.1) queries a full gradient oracle,

T.2) de-biases the local gradients using a correction term (with the last synchronized iterate in memory) to deal with objective heterogeneity,

T.3) error corrects to deal with compression at the client and the server, and

T.4) uses a client-specific learning rate to deal with the different number of local steps on each client.

Thus, FedLin is able to accommodate both objective and system heterogeneity along with sparse gradients and local steps. In doing so it is an early work that can deal with these multiple aspects of federated learning at once, albeit at the cost of an expensive gradient oracle.

The paper provides the following theoretical guarantees for FedLin,

R.1) matching upper and lower bounds in the strongly convex-smooth setting without compression,

R.2) upper bounds without compression in the convex-smooth setting and the non-convex setting,

R.3) an iterate sub-optimality recursion for the strongly convex-smooth setting while using a stochastic gradient oracle,

R.4) upper bounds for client and server compression in the strongly convex-smooth setting.

See the main review for comments on these results.


**Limitations And Societal Impact:**

See the main review.

**Main Review:**

I think overall the paper is clear and well-written. I have still only given it a score of 6, because there are aspects which can be improved, some of which are just presentation issues. If the following concerns can be handled, I am willing to improve my score.

**v/s Scaffold**

The upper bounds in R.1, R.2, and R.3 are not better than Scaffold [1] but they can additionally accommodate a different number of local steps. On the other hand, Scaffold uses a stochastic oracle and accommodates partial client sampling. Both these differences are not really big though. With a step size of $1/\tau_i$, it is not surprising that FedLin can control the client drift, and it is not difficult to introduce uniform partial client sampling either. Thus if it were only for the upper bounds in R.1-R.3 (which are compression-free), the results would have been incremental to Scaffold.

What is perhaps more interesting is the lower bound in R.1. After the Scaffold paper [1], one open question was if such a conservative client learning rate was actually necessary to compete with large mini-batch SGD (T rounds of batch size HM)? One could conjecture that it was a necessary evil to control the client drift, essentially giving us large mini-batch SGD like updates despite local steps! I.e., the obtained convergence rate can't depend on $TH$ but only $T$. This paper proves this conjecture for FedLin, giving lower bounds that depend only on $T$. This was a major limitation of Scaffold's results and analysis, and it is satisfying to see this lower bound.

For the sake of completeness and for comparison, I would encourage the authors to also provide a similar lower bound for Scaffold, perhaps in the appendix. I expect that Scaffold's lower bound might also match its upper bound. In that case, I am not sure if the results are very exciting in the uncompressed case. Do the authors believe that proving Thm. 1-4 is technically very different from Scaffold ([1] does a good exposition of its proof technique in its appendix)? Is it fundamentally different from choosing a small enough step-size to control the client drift? If there are additional technical challenges besides this, they should be highlighted, otherwise I maintain that the techniques are incrementally different.

**Sparsification results**

I believe these results are indeed novel. It would have been nice to have a theorem statement with both server and client level sparsification. Did the authors try this? What were the challenges in doing so?

**Presentation**

Based on my comments above, I think the authors should spend less space presenting and discussing Thm 1.-4 given how they are incremental over Scaffold. The paper should actually develope the lower bound in R.1 better, talk about its construction, and what it implies about other distributed algorithms like Scaffold. Perhaps Theorem 1 doesn't even need to be in the main paper, as Theorem 6. already generalizes it. It would be good to point out how the proof techniques in section 5. are different/similar to the ones in section 4, i.e., controlling the client drift? Finally, I would encourage the authors to talk a bit more about the novel proof technique developed in section 6.

It feels a bit weird that the authors claim that all of the results can be obtained for the stochastic setting, but still don't chose to present them (even in the appendix). Why was Theorem 4. not completely worked out, along side other theorems? Having a noise term in the guarantee doesn't obfuscate the analysis or the takeaways. For instance I could go back to the Scaffold paper, and send the noise term to zero to compare the results with this paper very easily. Infact, it is important to understand if stochasticity behaves differently in presence of heterogeneity/local steps. Won't the authors be required to make more assumptions (perhaps on the stochastic gradients) to even carry out the analysis (like [2], which is a related reference that should probably be compared to as well)? It is not clear to me. Either present the stochastic results, or be upfront about the issues in doing so. In practice, it is going to be prohibitive to use a full gradient oracle, and some mini-batch version of FedLin would be used. If so how would that compare to accelerated minibatch SGD [2] ?

**Experiments**

Continuing on the last issue, brings me to the (lack of) baselines in experiments. The algorithm essentially compares only to itself!

1) Why is Scaffold not present in any of the experiments and specially in the motivating figure 1?

2) In practice people will use a stochastic oracle. Atleast some (if not all) experiments should also show a stochastic version of FedLin, along with important baselines such as accelerated minibatch SGD, Scaffold, etc. I would expect that when the local steps are the same and the heterogeneity level is low, different trade-offs will start emerging, and algorithms like  large mini-batch SGD or FedAC [3] might dominate FedLin.

**Minor Comments**

1) It would be better to present Theorem 4 for function sub-optimality, instead of recursion on iterate sub-optimality. This is good for uniformity w.r.t. the other results as well as to make the comparison with earlier work more transparent.

# Updated Review

The authors have honestly addressed some of the limitations of their work. They have also promised to make important additions to the revised paper including experiments with other baselines and with a stochastic oracle, as well as a more detailed discussion of the baselines. I hope the authors will sincerely make these additions and will also improve upon their presentation as discussed. In light of this I am improving my score to a 7.

**References**

[1] Karimireddy, Sai Praneeth, et al. "Scaffold: Stochastic controlled averaging for federated learning." International Conference on Machine Learning. PMLR, 2020.

[2] Woodworth, Blake, Kumar Kshitij Patel, and Nathan Srebro. "Minibatch vs Local SGD for Heterogeneous Distributed Learning." Advances in Neural Information Processing Systems 33 (NeurIPS 2020) (2020).

[3] Yuan, Honglin, and Tengyu Ma. "Federated Accelerated Stochastic Gradient Descent." Advances in Neural Information Processing Systems 33 (2020).

**Time Spent Reviewing:**

10

---

> ### Author Response · Authors · 2021-08-09
> **Response to Reviewer Xwfq**
>
> We thank the Reviewer for their detailed and valuable feedback on our work. Below, we discuss each of the Reviewer's concerns,
> and explain how we plan to address them in the revised version of our manuscript.
>
> Reviewer: The upper bounds in R.1, R.2, and R.3 are not better than Scaffold [1] but they can additionally accommodate a different number of local steps.
>
> Response: Theorems 1-4 help demonstrate via simple, clean results that whatever rates can be achieved without systems heterogeneity, can also be achieved under the systems heterogeneity model in the recent work (Wang et al., NeuRIPS 20). Since varying client speeds is a practical concern in FL, we believe that these results are important.
>
> Reviewer: For the sake of completeness and for comparison, I would encourage the authors to also provide a similar lower bound for Scaffold, perhaps in the appendix.
>
> Response: This is a very pertinent comment. The main factor that contributes to the lower bound in Theorem 5 is the *staleness* of the gradient correction term.  Since the gradient correction term in FedLin is less stale than the control variates employed in Scaffold (lines 177-188 of our paper),  intuition dictates that problems arising due to staleness in FedLin should also manifest themselves in Scaffold. We will try our best to formally establish this fact and add it to the appendix as the Reviewer suggests; we believe this is quite doable.
>
> Based on the fixed-point property that FedLin satisfies, but Scaffold does not (lines 177-188), our hope was that we will be able to establish better convergence rates relative to Scaffold for the uncompressed setting.  We conjecture that the reason why our convergence rate is no better than that of Scaffold is because of the *arbitrary* objective heterogeneity assumption. Nonetheless, we believe that our approach provides a promising unified framework to tackle some of the key challenges in FL: objective and systems heterogeneity, and compression.
>
> Reviewer: Do the authors believe that proving Thm. 1-4 is technically very different from Scaffold ([1] does a good exposition of its proof technique in its appendix)? Is it fundamentally different from choosing a small enough step-size to control the client drift?
>
> Response: Our proofs of Theorems 1-4 are simpler relative to Scaffold. The reason for this is that the gradient correction term used in round t of FedLin only uses terms from the beginning of round t. Thus, when we develop a recursion relating iterates at the end of round t to those at the beginning of round t, we need not worry about quantities in round t-1. This is not the case with Scaffold as the control variates used in round t do involve terms from round t-1. The simplicity of our proof has two key benefits: (i) it helps build intuition and exposes some of the basic ideas at play; and (ii) prepares us to tackle the more involved settings when compression is considered.
>
> Although our analysis in the uncompressed case does bear similarities with that for Scaffold, it should, however, be noted that almost all proofs that we have come across for FL algorithms essentially share the same structure: (i) establishing a "one round progress" recursion, and (ii) bounding the client-drift that naturally shows up in such a recursion. Moreover,  choosing a small step size to control the client drift is also common to all FL algorithms, ours being no exception. For instance, if one takes a closer look at the analysis for FedAvg in the Scaffold paper, the client learning rate $\eta_l$ scales inversely with the number of local steps H, exactly as in both Scaffold and FedLin.
>
> Reviewer: (Sparsification results) I believe these results are indeed novel. It would have been nice to have a theorem statement with both server and client-level sparsification. Did the authors try this? What were the challenges in doing so?
>
> Response: We agree with the Reviewer's viewpoint that having a result, or at least a discussion on sparsification at both ends would be useful. We did not try and derive such a result for the following reason. The primary focus of our compression analysis was to identify key differences between up-link and down-link sparsification, and to investigate which of these mechanisms preserve linear convergence. Our analysis reveals that while server-side sparsification preserves linear convergence, sparsification at clients does not. Based on these insights, it seemed natural that with compression at both ends, FedLin would converge linearly to a ball around the true minimum, at best. Thus, even without any further analysis, the qualitative nature of convergence with sparsification at both ends was known to us.
>
> We do not foresee any major technical challenges in deriving a result involving sparsification at both ends; this should be a combination of our existing analyses. We will add a discussion on this topic in the revised paper.
>
> Response to Reviewer's comment on presentation: We thank the Reviewer for their valuable suggestions. In the revised paper, we will discuss the lower bound in Theorem 5 in more detail, and point out its implications for Scaffold. We will also provide intuition regarding our proof techniques in Sections 5 and 6. In a nutshell, the proofs of Theorems 7 and 8 are based on the construction of appropriate Lyapunov functions that account for both iterate sub-optimality and errors due to sparsification.  We will add the precise nature of our potential function in the revised manuscript.
>
> Response to Reviewer's comments on noise/stochasticity: As we discuss at the end of Section 2, even in the deterministic setting, there were unresolved questions regarding when one can achieve linear convergence in the presence of objective and systems heterogeneity and biased compression. Since the main focus of our work is to systematically answer these questions, our exposition primarily deals with the deterministic case.
>
> Nonetheless, as the Reviewer rightly suggests, understanding the performance of FedLin in a stochastic setting is indeed important. To this end, Theorem 4 highlights two key aspects. First, the recursion in Eq. (9) exactly resembles that for centralized SGD. Second, to arrive at such a recursion, we make the exact same assumptions on the stochastic gradients as are made in the analysis of centralized SGD - unbiased gradients with bounded variance, nothing more. Hopefully, this clarifies the Reviewer's queries regarding how stochasticity behaves with heterogeneity/local steps and the nature of assumptions on the stochastic gradients. Our result in Theorem 4 suggests that even in the presence of heterogeneity (both objective and systems) and local steps, one can achieve the exact same O(1/T) convergence rate as for centralized SGD.
>
> For clarity, instead of leaving the result of Theorem 4 as a recursion, we will work it out completely in the revised manuscript and explicitly show the O(1/T)  rate.
>
> Since establishing even the noiseless versions of Theorems 6-8 is quite non-trivial and involves messy calculations, we did not want to clutter the exposition further by considering stochasticity. That being said, we believe that the analysis of Theorem 4 provides all that is needed for an interested reader to work out the noisy versions of these results.
>
> Reviewer: In practice, it is going to be prohibitive to use a full gradient oracle, and some mini-batch version of FedLin would be used. If so how would that compare to accelerated minibatch SGD [2]?
>
> Response: In the revised paper, we will conduct experiments to compare FedLin with the accelerated minibatch SGD method in [2]. In Appendix L, we do conduct experiments for FedLin with a stochastic oracle. Our simulation results comply with the theory in Theorem 4: under a general stochastic oracle model, FedLin converges linearly to a ball around the true minimum, where the size of the ball depends on the variance of the noise model. We missed pointing this out in the main body of the paper and will make sure to discuss this in the revised manuscript.
>
> Reviewer: (i) Why is Scaffold not present in any of the experiments and specially in the motivating figure 1? (ii) In practice people will use a stochastic oracle. Atleast some (if not all) experiments should also show a stochastic version of FedLin, along with important baselines
> such as accelerated minibatch SGD, Scaffold, etc.
>
> Response to (i): We were unsure of what would be the right way to set up fair experimental comparisons with Scaffold since it neither accounts for systems heterogeneity nor compression, both of which are key features of our algorithm. For that matter, no FL algorithm we know of studies biased compression - thus, there are no natural baselines to compare with here. Moreover, other than FedNova (to which we do compare), no other FL algorithm simultaneously considers both objective and systems heterogeneity.
>
> When there is no sparsification or systems heterogeneity, the convergence rates for Scaffold and FedLin coincide. Thus, for this setting, we did not expect to see any major differences between the two algorithms that can be explained by our current theory.  Based on the Reviewer's recommendation (and also the recommendation of Reviewer 2DHS), we will make sure to compare FedLin with Scaffold and report our observations in the revised paper.
>
> In Section II (that contains Fig. 1), we wanted to motivate our work by conveying that even in a deterministic setting, there exist unresolved technical challenges in FL. We used the recently proposed FedNova algorithm [21] to demonstrate this fact since in [21] the authors argue that normalized weighted aggregation can tackle the objective inconsistency problem. We wanted to highlight that this is not the case.
>
> Response to (ii): As we mentioned earlier, we do conduct experiments on FedLin with a stochastic oracle model in Appendix L. We will also compare FedLin with the baselines suggested by the Reviewer.

---

### Comment · Area_Chair_PTXh · 2021-09-02
**Clarifying relation to FedSVRG**

Dear authors,

Please excuse this question almost at the end of the discussion phase. However, could you please explain to the committee the differences between FedLin and FedSVRG as in [[Konecny et al, Federated Optimization: Distributed Machine Learning for On-Device Intelligence, 2016](https://arxiv.org/pdf/1610.02527.pdf)] - your citation [1] ?

While I see that the paper contains numerous additional technical contributions on the analysis side, the claims regarding the novelty of the algorithmic framework could perhaps be phrased more cautiously in regard to [1].

-- AC

---

> ### Author Response · Authors · 2021-09-02
> **Clarifications pertaining to FedSVRG**
>
> Dear AC,
>
> Thank you for pointing us to this specific algorithm. We took a close look at Algorithm 4 (Federated SVRG) in reference [1]. In what follows, we discuss the similarities and differences of FedLin with FedSVRG. We note here that (as far as we could tell), the FedSVRG algorithm in [1] does not come with any formal guarantees of convergence. In contrast, key contributions of our paper include a detailed technical analysis for FedLin. We discuss this point later.
>
> **Algorithmic Similarities and Differences**
>
> - One major difference of FedSVRG with FedLin is that it does not consider
> the effect of gradient compression/sparsification. Our local update rule, in contrast, accommodates
> inexact gradient updates which significantly complicates the analysis. Nonetheless, our work provides a general analysis framework for studying gradient-tracking/variance reduction schemes with inexact gradient correction terms. Since gradient-tracking/variance reduction are two very popular techniques in the optimization and machine learning community, we believe that such an analysis will be useful beyond the specific FL context we study here.
>
> - Unlike our work, system heterogeneity does not seem to be explicitly considered in [1]. That being said, in Algorithm 4 of Ref [1], each client runs a number of local computations equal to the number of data points on its device, and the step-size is adjusted accordingly. We find this connection very interesting. Specifically, our choice of learning rate stems from a tight client-drift analysis. In [1], although no such analysis is provided to motivate the choice of learning rate, it is mentioned that the learning rate is chosen in a manner to ensure that all clients roughly make the same amount of progress.
>
> - In the absence of compression effects, the update rule in line 8 of FedSVRG does indeed include a similar gradient-correction term like FedLin (assuming that full gradients can be accessed at each node). However, the former has certain additional diagonal scaling matrices (denoted by $S_k$) that do not appear in FedLin. Moreover, unlike FedLin, the aggregation step (line 11) in FedSVRG uses a pre-conditioning matrix $A$. At this point, we are not sure what differences in the convergence rates arise because of these scaling matrices.
>
> We were unaware of these connections to FedSVRG, and are grateful to the AC for bringing them up. In our revised manuscript, we will make sure to discuss these points in detail, and explain our contributions accordingly.
>
> **Differences in terms of Analysis**
>
> We note that no convergence analysis is provided for FedSVRG in [1]. In contrast, our work provides a detailed technical analysis that systematically covers tight linear convergence rate guarantees; guarantees for the convex and non-convex setting; guarantees under noise for a general stochastic oracle model; and also a comprehensive analysis of the effects of both up-link and down-link compression/sparsification. In particular, while our paper provides lower bounds for FedLin, no such lower bounds are reported in papers that propose competitive algorithms like FedProx, Scaffold, FedNova, and FedSplit. It would be interesting to explore the connections between the theoretical analyses developed in our paper for FedLin and the empirical observations made in [1] for FedSVRG.
>
> Overall, we would like to once again thank the AC for highlighting the connection of FedLin to FedSVRG. We will make sure to discuss all the above relevant points in our revised manuscript.

---

### Decision · Program_Chairs · 2021-09-27

**Decision:**

Accept (Poster)

**Comment:**

This paper introduces the FedLin algorithm (which could be seen as an adaptation of FedSVRG) for cross-silo federated learning (without client sampling) and derives rigorous complexity guarantees.

The reviewers commended the theoretical results, in particular the simple convergence proof of FedLin, and corresponding (algorithm specific) lower bound.
Partial communication compression was studied as additional contribution. Yet, this aspect was assessed more critically by the reviewers, as  compression of the parameters is not supported.

The reviewers are of the opinion that their concerns were adequately addressed by the author's response (and that the promised changes by the authors will be implemented in the final version).

Additionally, I strongly encourage the authors to also include:
- a discussion of the relation to 'FedSVRG' [[Konecny et al, Federated Optimization: Distributed Machine Learning for On-Device Intelligence, 2016]](https://arxiv.org/pdf/1610.02527.pdf) and possibly additional literature related to 'federated SVRG' variants. In this regard, the claims on novelty should be phrased more carefully. (In particular also more carefully than in the author's response, as e.g. the mentioned algorithmic differences seem quite small and partially irrelevant (i.e. omitting a discussion of heterogenity for an algorithm that does not depend on heterogenity does not seem to be a limitation, etc.).
- a discussion of the 'client sampling' aspect would be appreciated. The limitation of FedLin to do a pass over all clients (and requiring twice as much communication as FedAvg) seems quite limiting in practice.